# Sharper Characterization of the Global Maximizers in Bilinear Programming with Applications to Asynchronous Gradient Descent

## Abstract

We study the bilinear program that arises when tuning the stepsizes in asynchronous gradient descent (AGD). Notably, we prove a necessity theorem: every global maximizer lies at an extreme point of the feasible region, strengthening the classical sufficiency guarantee for linear objectives on compact sets. Exploiting this structure, we recast the continuous problem as a discrete search over the vertices of the hyper-cube and design a solver that performs a biased random walk among them. Over all the tested benchmarks, including the *Cyclic Staircase* benchmark, our solver reaches global optimality up to $1000\times$ faster than Gurobi 11 while using orders of magnitude fewer evaluations.

This structural result allows us to prove near-optimal stepsize scheme for the recently proposed Ringmaster AGD algorithm and a provable factor–2 approximation on the error to find an $\varepsilon$–stationary point. Together, our results provide both a sharper theoretical characterization and a practical solver for nonconvex bilinear programs emerging in distributed learning.

## 1 Introduction

Efficient optimization lies at the heart of modern AI applications. As neural networks scale toward trillion-parameter models (Rajbhandari et al., 2020), training must be distributed across hundreds or even thousands of compute nodes (Llama Team, 2024; Microsoft, 2024; OpenAI, 2024; Gemini Team, 2025). While Minibatch-SGD (Cotter et al., 2011; Dekel et al., 2012; Takac et al., 2013) is one of the most commonly used distributed training strategies, it suffers from a fundamental bottleneck: each worker must wait for the slowest node to finish its computations, leading to significant under-utilization of resources (Goyal et al., 2017; Bottou et al., 2018).

It is therefore not just natural, but practically necessary, to allow workers to proceed asynchronously. Asynchronous gradient-descent (AGD) methods (Recht et al., 2011) enable workers to read potentially stale model parameters and submit gradients without locks or coordination, dramatically improving resource utilization and throughput. However, this flexibility comes at a price: the presence of delays and stale updates makes the selection of an appropriate sequence of step sizes both crucial and challenging. A poorly chosen step-size schedule can easily lead to divergence or drastically slow convergence, while a carefully designed one can unlock the full potential of large-scale distributed training. Understanding and addressing this challenge is therefore indispensable for the practical success of asynchronous optimization in modern AI systems.

Like all gradient-descent methods, a crucial design choice in AGD is the stepsize policy, which must offset the extra variance introduced by delayed gradients. To our surprise, in many works the stepsizes are engineered based on prior intuitions on the behavior of the optimization method and lack rigorous justifications. While in general these hand crafted stepsizes does not hurt the convergence rate, they might lead to suboptimal *hidden constant* which in practice, e.g., when training large machine learning models, can be detrimental, especially in decentralized and federated learning (Dean et al., 2012; McMahan et al., 2017; Kairouz et al., 2021). Investigating for optimal stepsizes in AGD and compare them to known methods is therefore a crucial step, beyond the theoretical convergence rates, to understand how one algorithm compare to the other in practical scenarios.

## 1.1 OUR CONTRIBUTIONS

The contributions of the present work span from advances in bilinear programming theory and its implications to the design of asynchronous optimization methods, with a particular focus on providing a deeper understanding of the optimal choice of the stepsizes.

♠ **An Optimization Problem for Choosing The Stepsizes.** We show that selecting improved stepsizes for asynchronous gradient descent (AGD) can be cast as an optimization problem with a linear objective and bilinear constraints.

♣ **A Sharper Characterization of the Global Maximizers.** Starting from our stepsize problem, and beyond the existence of an optimal solution, we provide a sharper characterization of the optimal solutions of a whole family of bilinear programs by establishing a *necessity* theorem: every global maximizer is necessarily extremal, thereby tightening the classical result.

♦ **A Simple yet Powerful Heuristic to Solve the Optimization Problem.** Leveraging our extremality guarantee, we show how a simple randomized heuristic, searching over the vertices of the feasible region, can already be very efficient in practice and we empirically compare this heuristic to the general-purpose solver Gurobi.

Together, these contributions yield both a refined theoretical understanding and a practical heuristic for nonconvex bilinear programs, particularly those with separable or low-dimensional nonconvex components, such as problems with one constraint per coordinate of the ambient space. This framework is not limited to AGD, and can be naturally extended to inform the design of other distributed learning methods.

## 2 RELATED WORKS[†]

### 2.1 ASYNCHRONOUS GRADIENT DESCENT (AGD)

Asynchronous optimization can be dated back to the 1970-80s (Baudet, 1978; Tsitsiklis et al., 1986; Bertsekas & Tsitsiklis, 1989) and regains interest with the seminal work of Recht et al. (2011). While subsequent works have focused on the stochastic variant of AGD, i.e., ASGD (Agarwal & Duchi, 2011; Chaturapruek et al., 2015; Lian et al., 2015; Feyzmahdavian et al., 2016; Sra et al., 2016; Dutta et al., 2018; Nguyen et al., 2018; Arjevani et al., 2020; Stich & Karimireddy, 2020), it is only recently that tight convergence analysis of ASGD and optimal algorithms have been derived (Koloskova et al., 2022; Mishchenko et al., 2022; Feyzmahdavian & Johansson, 2023) culminating in Ringmaster ASGD (Maranjyan et al., 2025) with provable optimal time complexity. In Zhang et al. (2016); Mishchenko et al. (2022) delay-adaptive stepsizes are used where the learning rate is divided by the delay while Koloskova et al. (2022); Maranjyan et al. (2025) use a threshold to penalize/discard stale gradients. Surprisingly, the delay threshold used in Ringmaster ASGD does not depend on the compute times nor on the delays and it is an open question whether one can improve this threshold.

### 2.2 BILINEAR PROGRAM (BLP)

BLPs are a class of nonlinear optimization problems in which the objective function or constraints are bilinear, i.e., involve products of disjoint pairs of variables, leading to intrinsic non-convexity and computational hardness (Al-Khayyal, 1992). Even for seemingly simple linear objectives and bilinear constraints, the feasible region can have complex geometry (Horst & Hoang, 1996). BLPs arise in diverse applications from pooling (Misener & FLOUDAS, 2009) and packing (Locatelli & Raber, 2002) to network design (Davarnia et al., 2017) and economic equilibrium (Mathiesen, 1985), motivating a range of algorithmic solutions. Approaches for solving BLPs include convex relaxations such as McCormick envelopes (McCormick, 1976), mixed-integer programming reformulations (Adams & Sherali, 1993), and advanced cutting plane or disjunctive algorithms (Saxena et al., 2011; Fampa & Lee, 2021; Rahimian & Mehrotra, 2024) for global solution strategies. Despite these advances, exact solution and efficient computation for large-scale BLPs remain significant research challenges (Rahimian & Mehrotra, 2024).

---

† We refer the reader to Appendix B for further references.

# 3 GLOBAL MAXIMIZERS IN BILINEAR PROGRAMS

In this section, we introduce and study in depth a class of bilinear programs that is essential for our later analysis of AGD.

## 3.1 THE OPTIMIZATION PROBLEM

The quadratically constrained program we are interested is the following maximization problem:

$$
\begin{aligned}
(\mathscr{P}_d): \quad & \text{maximize } \langle \Lambda \mid \mathbf{a} \rangle = \sum_{k=1}^{d} a_k \lambda_k \\
& \text{over} \qquad (\lambda_1, \ldots, \lambda_d) \in [0, 1]^d \\
& \text{subject to } 0 \le \lambda_i \left( 1 + \sum_{j=1}^{d} M_{i,j} \lambda_j \right) \le 1 \text{ for } i = 1, 2, \ldots, d;
\end{aligned}
\tag{1}
$$

where $d > 0$ is the dimension, $\Lambda = (\lambda_1, \ldots, \lambda_d)^\top$ are the variables of the problem, $\mathbf{a} = (a_1, \ldots, a_d)^\top \in \mathbb{R}^d$ is a constant vector such that for all $i \in [d]$, $a_i \ne 0$ and $M$ is a $d \times d$ matrix with non-negative entries. It is worth noting that the bilinear constraints of $(\mathscr{P}_d)$ can be re-written in the following "matrix-form" inequality

$$
0 \le \Lambda + \Lambda \odot (M\Lambda) \le 1,
\tag{2}
$$

where $\odot$ denotes the Hadamard product, i.e., element-wise multiplication[1] and the inequalities from (2) are considered coordinate-wise. Additionally, notice that problem $(\mathscr{P}_d)$ is scale-invariant in $\mathbf{a}$, that is, if we scale the vector $\mathbf{a}$ in the objective function by some positive scalar then the set of solution is unchanged.

Throughout this work, while we mainly focus on the general case where $M$ has non-negative entries, we also highlight in Appendix F and in the experiments reported in Section 6 an important special case of problem (1) where $M$ is a strictly upper triangular matrix (with non-negative entries), that is, $M_{i,j} = 0$ for every $1 \le j \le i \le d$ and $M_{i,j} \ge 0$ for all $1 \le i, j \le d$ so that the constraints in (2) simplify to

$$
\lambda_i \left( 1 + \sum_{j=i+1}^{d} M_{i,j} \lambda_j \right) \le 1, \quad i = 1, 2, \ldots, d.
\tag{3}
$$

Observe that in this particular case, the problem reduces to a BLP, as it involves only products of distinct variables. This scenario naturally arises in the state-of-the-art analysis of asynchronous gradient descent (AGD) as outlined in Section 5 and more thoroughly in Appendix G.

## 3.2 GEOMETRY OF THE FEASIBLE REGION

**The feasible region.** We start by defining the feasible region of $(\mathscr{P}_d)$.

**Definition 3.1.** The feasible region $\mathscr{F}$ of problem $(\mathscr{P}_d)$ is

$$
\mathscr{F} := \left\{ \Lambda \in [0, 1]^d : 0 \le \Lambda + \Lambda \odot (M\Lambda) \le 1 \right\},
\tag{4}
$$

where $M$ is a $d \times d$ matrix with non-negative entries.

In Figure 1 we give visualizations of the feasible region $\mathscr{F}$ for several matrices $M$ in dimension 3.

In Appendix D.2 we provide several lemmas to better understand the geometrical properties of $\mathscr{F}$ and the constraints from (4). In Lemmas D.1 and D.2 we study how the constraints shape the region $\mathscr{F}$. Then, in Definitions D.4, D.5 and D.8 we partition $\mathscr{F}$ in different components based on the which constraints are tight or not and study the properties of these components in Lemmas D.6 and D.7.

---

[1]For two matrices $A$ and $B$ from $\mathbb{R}^{d \times n}$, the Hadamard product of $A$ by $B$, denoted by $A \odot B$ is the matrix $C$ whose entry $(i, j) \in [d] \times [n]$ is given by $C_{i,j} = A_{i,j} \times B_{i,j}$.

**Some intuitions on the feasible region and problem** $(\mathscr{P}_d)$**.** To provide additional intuition, beyond the formal lemmas in the appendix, we give some insights on the geometry of the feasible region $\mathscr{F}$ and how it enforces that every maximizer of $(\mathscr{P}_d)$ must be an extreme point.

First, note that the set $\mathscr{F}$ can be obtained as the unit hypercube $[0,1]^d$ to which we remove the *convex* subsets defined by the constraints

$$\lambda_i \left(1 + (M\Lambda)_i\right) \geq 1, \quad i = 1, 2, \ldots, d,$$

where $\lambda_1, \ldots, \lambda_d \geq 0$. The resulting shape, which is $\mathscr{F}$, is connected, compact, and its faces are either "flat" (which happens when one of the $(\lambda_i)_{i \in [d]}$ is zero, and the face is aligned with one of the axis) or "concave"[2]: this can be seen in Figure 1: in both figures we have 4 "flat" faces and 2 "concave" faces. These "concave" faces belong to the hypersurface $\lambda_i(1 + (M\Lambda)_i) = 1$ for some $i \in [d]$, and importantly *none* of these hypersurfaces is an affine hyperplane. Therefore, given a cost vector $\mathbf{a} \in \mathbb{R}^d$, the levels sets of the objective function $\langle \Lambda \mid \mathbf{a} \rangle$, which are hyperplanes orthogonal to $\mathbf{a}$, never coincide with or are contained in any of the "concave" faces of $\mathscr{F}$. Additionally, it is useful geometrically to interpret the maximization of the objective $\langle \Lambda \mid \mathbf{a} \rangle$ as *sliding a hyperplane, oriented according to* $\mathbf{a}$*, in the direction of increasing* $\langle \Lambda \mid \mathbf{a} \rangle$. With this geometric perspective, the concavity of the faces of $\mathscr{F}$ ensures that, as the hyperplane is translated in the direction of increase, whenever it intersects a face $F$ of $\mathscr{F}$, there is always a portion of the boundary of $F$ that lies above the hyperplane. Consequently, as long as the intersection with $\mathscr{F}$ is non empty, there is always one extreme point above this sliding hyperplane. An illustration of such a hyperplane crossing $\mathscr{F}$ is displayed in Figure 1a. This geometric intuition, which conceals certain subtleties, is precisely what the necessity theorem formalizes.

Moreover, it is worth mentioning that in problem (1), the cost vector $\mathbf{a} \in \mathbb{R}^d$ is assumed to have non-zero entries. This condition is actually essential and it ensures that none of the "flat" faces of $\mathscr{F}$ are entirely contained in any of the level set of the objective $\langle \Lambda \mid \mathbf{a} \rangle$; otherwise, our main result would not hold.

## 3.3 THE SUFFICIENCY RESULT

In this section, we recall a general result which implies that the problem $(\mathscr{P}_d)$ in (1) admits at least one optimal solution that is an extreme point of the feasible region. Before stating our results, we recall the two common definitions of an *extreme point* for general (e.g., non-convex) subsets of $\mathbb{R}^d$. One (Definition 3.2) is more wide spread in the literature than the other (Definition 3.3). We refer the reader to Appendix C.2 for further discussions on this point.

**Definition 3.2** (Extreme Point). Let $S \subseteq \mathbb{R}^d$ be a non-empty subset, a point $x \in S$ is said to be an *extreme point* of $S$ if, for any $a, b \in S$ with $a \neq b$, the point $x$ does not lie in the interior of the segment $[a, b]$, that is, $x \notin (a, b)$. The set of extreme points of $S$ is denoted by $\text{Extr}\, S$.

**Definition 3.3** (Extreme Point: a Relaxed Variant). Let $S \subseteq \mathbb{R}^d$ be a non-empty subset, a point $x \in S$ is said to be an *extreme point in the "relaxed" sense* of $S$ if, for any $a, b \in S$ with $a \neq b$ such that $[a, b] \subset S$ the point $x$ does not lie in the interior of the segment $[a, b]$, that is, $x \notin (a, b)$. The set of extreme points of $S$ in the sense of this relaxed definition is denoted by $\text{Extr}_{\mathcal{R}}\, S$.

Clearly we have $\text{Extr}\, S \subseteq \text{Extr}_{\mathcal{R}}\, S$ for any subset $S \subseteq \mathbb{R}^d$. This inclusion can be tight in some specific cases, for instance, when $S$ is a convex set[3] we have $\text{Extr}\, S = \text{Extr}_{\mathcal{R}}\, S$.

We now consider the general optimization problem:

$$(\mathscr{P}_{\text{cpt}}^{\text{lin}}): \quad \text{maximize} \quad \langle \mathbf{x} \mid \mathbf{c} \rangle$$
$$\text{over} \quad \mathbf{x} \in K, \tag{5}$$

where $\mathbf{c} \in \mathbb{R}^d \setminus \{0\}$ is a constant non-zero vector and $K \subseteq \mathbb{R}^d$ a non-empty and compact subset[4]. Nonetheless we can still say something about some of the global maximizers of prob-

---

[2]Roughly speaking, these faces can be intuitively visualized as surface of concave lens.

[3]So as to make the paper self-contained, we recall some basic notions of convexity (convex sets, convex functions...) in Appendix C.1.

[4]Here we do not impose anything special on the geometry of the compact set $K$, e.g., convexity or the fact that $K$ is described by linear inequalities. So $K$ can be an arbitrary compact and non-empty subset of $\mathbb{R}^d$, notably $K$ is not necessarily convex.

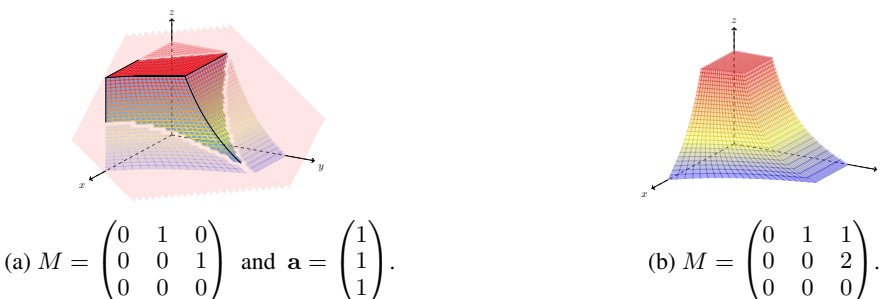

(a) $M = \begin{pmatrix} 0 & 1 & 0 \\ 0 & 0 & 1 \\ 0 & 0 & 0 \end{pmatrix}$ and $\mathbf{a} = \begin{pmatrix} 1 \\ 1 \\ 1 \end{pmatrix}$.  (b) $M = \begin{pmatrix} 0 & 1 & 1 \\ 0 & 0 & 2 \\ 0 & 0 & 0 \end{pmatrix}$.

Figure 1: Illustration of the feasible region for two instances of problem (1) in $d = 3$.

lem (5) as stated in the next result and proved in Appendix E.1. For convenience, we denote by $X^* := \arg\max_{\mathbf{x} \in K} f(\mathbf{x}) := \langle \mathbf{x} \mid \mathbf{c} \rangle$ the set of global maximizers of (5).

**Theorem 3.4** (Maximization of Linear Forms over Non-empty Compact Sets). *There exists an optimal solution of problem $(\mathscr{P}_{\mathrm{cpt}}^{\mathrm{lin}})$ in (5) which is also an extreme point of $K$, i.e., $\mathrm{Extr}\, K \cap X^* \neq \varnothing$.*

Actually Theorem 3.4 above is a special case of Theorem 3.1 from Chen et al. (2021) but since we only focus here on the particular case where the objective is linear, we can prove Theorem 3.4 more directly. We refer the reader to Appendix E.1.[5]

### 3.4 Some Key Lemmas

In this part, we establish two key results concerning the system of inequalities defined by the $d$ constraints in problem $(\mathscr{P}_d)$ in (1). In the first result (Lemma 3.5), we prove that one can control the value of each coordinate of the column vector $\Lambda + \Lambda \odot (M\Lambda)$. That is, given some weights $\mathbf{w} = (w_1, \ldots, w_d)^\top \in [0, 1]^d$, the system of $d$ equations $\Lambda + \Lambda \odot (M\Lambda) = \mathbf{w}$, is always solvable and we prove that this system admits a unique solution $\Lambda^{(\mathbf{w})}$. In the second result (Lemma 3.6) we study the regularity of this unique solution as the weights vector $\mathbf{w}$ varies in $[0, 1]^d$.

**Lemma 3.5** (A Linear-Quadratic System; Proof in Appendix E.2). *Let $d \in \mathbb{N}$ be a positive integer, $M \in \mathbb{R}^{d \times d}$ a matrix with non-negative entries and $W = (w_1, \ldots, w_d)^\top \in \mathbb{R}^d$ a $d$-dimensional column vector with non-negative entries. Then, the system*

$$\Lambda + \Lambda \odot (M\Lambda) = W, \tag{6}$$

*has a unique solution $\Lambda = (\lambda_1, \ldots, \lambda_d)^\top \in \mathbb{R}^d$ with non-negative entries and for any $i \in [d]$ we have $\lambda_i = 0$ if, and only if $w_i = 0$.*

The proof of Lemma 3.5 is deferred to Appendix E.2. It uses the notion of $P$-matrix and crucially relies the *Gale–Nikaidô* theorem. This theorem is a powerful tool which provides a link between $P$-matrices and the injectivity of functions defined from $\mathbb{R}^d$ to $\mathbb{R}^d$. The reader can refer to Appendix C.5 for more details about $P$-matrices.

Counter-examples to the existence and uniqueness of solution(s) to (6) are discussed in Appendix E.2.

**Lemma 3.6** (Regularity of the Solution of (6)). *Let $d \in \mathbb{N}$ be a positive integer and $M \in \mathbb{R}^{d \times d}$ a matrix with non-negative entries. For any $d$-dimensional column vector $\mathbf{w} = (w_1, \ldots, w_d)^\top \in \mathbb{R}^d$ with non-negative entries, let $\Lambda^{(\mathbf{w})} = (\lambda_1^{(\mathbf{w})}, \ldots, \lambda_d^{(\mathbf{w})})^\top$ be the unique solution of the equation*

$$\Lambda + \Lambda \odot (M\Lambda) = \mathbf{w}, \tag{7}$$

*then, the map $\Psi : [0, 1]^d \to \mathscr{F}$ defined for $\mathbf{w} \in [0, 1]^d$ by*

$$\Psi(\mathbf{w}) := \Lambda^{(\mathbf{w})} = \left( \lambda_1^{(\mathbf{w})}, \ldots, \lambda_d^{(\mathbf{w})} \right)^\top,$$

*where $\mathscr{F} := \left\{ \Lambda \in [0, 1]^d : 0 \leq \Lambda + \Lambda \odot (M\Lambda) \leq 1 \right\}$, is a $\mathcal{C}^\infty$–diffeomorphism[6].*

---

[5]Our argument is inspired by the solution in (https://math.stackexchange.com/users/232/qiaochu yuan).

[6]The notion of diffeomorphism is recalled in Definition C.1.

# 4 MAIN RESULTS

## 4.1 CHARACTERIZING THE EXTREME POINTS OF $\mathscr{F}$

We start this section by studying the extremal points[7] of the feasible set $\mathscr{F}$. More precisely, we prove that the set of extreme points of $\mathscr{F}$ can be characterized as the set of vertices $\{0,1\}^d$ of the hypercube $[0,1]^d$ mapped by the diffeomorphism $\Psi$ defined in Lemma 3.6.

The next two theorems characterize the extreme points of the feasible region $\mathscr{F}$, either in the general setting (Theorem 4.1) or when the matrix $M$ is assumed to be strictly upper triangular (Theorem 4.2). Their proof can be found respectively in Appendix E.3 and in Appendix F.1.

**Theorem 4.1** (Extreme Points of $\mathscr{F}$ in the Relaxed Sense). *For the feasible region $\mathscr{F}$ of problem $(\mathscr{P}_d)$, we have*

$$\mathrm{Extr}_{\mathcal{R}}\,\mathscr{F} = \left\{ \Psi(w)\,:\, w \in \{0,1\}^d \right\}, \tag{8}$$

*that is, the extreme points of $\mathscr{F}$ (in the relaxed sense) are exactly the vertices of the hypercube $[0,1]^d$ mapped by the diffeomorphism $\Psi$.*

In the particular case where the matrix $M$ is strictly upper triangular, we can strengthen this result with the set $\mathrm{Extr}\,\mathscr{F}$.

**Theorem 4.2** (Extreme Points of $\mathscr{F}$ in the Strictly Upper Triangular Case). *For the feasible region $\mathscr{F}$ of the problem $(\mathscr{P}_d)$ in the particular case where the matrix $M$ is strictly upper triangular with non-negative entries, we have*

$$\mathrm{Extr}\,\mathscr{F} = \left\{ \Psi(w)\,:\, w \in \{0,1\}^d \right\}, \tag{9}$$

*that is, the extreme points of $\mathscr{F}$ are exactly the vertices of the hypercube $[0,1]^d$ mapped by the diffeomorphism $\Psi$.*

*Remark* 4.3. As a consequence of the above two theorems, when the matrix $M$ is strictly upper triangular the feasible region $\mathscr{F}$ of problem $(\mathscr{P}_d)$ satisfies $\mathrm{Extr}\,\mathscr{F} = \mathrm{Extr}_{\mathcal{R}}\,\mathscr{F}$.

## 4.2 EVERY OPTIMAL SOLUTION IS EXTREMAL

We now state our main theorem which complements the "sufficiency" result from Section 3.3 and provides a sharper characterization of the global maximizers of problem $(\mathscr{P}_d)$. Indeed, while the later Theorem 3.4 asserts that there exists *at least* an extreme point of $\mathscr{F}$ which is an optimal solution to $(\mathscr{P}_d)$, our result strengthen this claim and states that *every* optimal solution to the problem $(\mathscr{P}_d)$ is necessarily an extreme point of $\mathscr{F}$ and hence, reduces the search space from the whole domain $\mathscr{F}$ to only its extremal points.

**Theorem 4.4** (Global Maximizers of Problem $(\mathscr{P}_d)$; Proof in Appendix E.4). *The set $X^*$ of the global maximizers of problem $(\mathscr{P}_d)$ as defined in (1) satisfies*

$$X^* \subseteq \left\{ \Psi(w)\,:\, w \in \{0,1\}^d \right\},$$

*that is, the global maximizers of $(\mathscr{P}_d)$ must be some points $p$ of the feasible region $\mathscr{F}$ which are mapped (through the bijection $\Psi^{-1}$) to the vertices of the unit hypercube $[0,1]^d$.*

More specifically Theorem 4.4 allows us to drastically simplify the original problem $(\mathscr{P}_d)$ by restricting the constrained set to a finite set of points. This gives the following reformulation of $(\mathscr{P}_d)$:

$$(\mathscr{P}_d'): \quad \text{maximize} \ \langle \mathbf{a} \mid \Psi(w) \rangle$$
$$\text{over} \qquad w \in \{0,1\}^d. \tag{10}$$

The essence of our result, illustrated in Figure 2, is that the inverse map $\Psi^{-1}$ carries the complicated feasible set $\mathscr{F}$ onto the familiar hypercube $[0,1]^d$. By Theorems 4.1 and 4.4, every global maximizer of the original problem lies at a vertex of $\mathscr{F}$. Hence it suffices to evaluate the objective only on the $2^d$ vertices in $\{0,1\}^d$ using $\Psi$ to pull them back to the corresponding points in the original space. This

---

[7]The definition of an extreme point is recalled in Section 3.3.

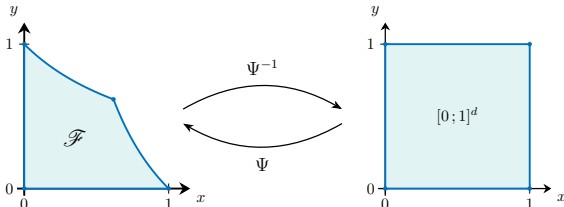

Figure 2: Flattening the Nonconvex Feasible Set $\mathscr{F}$ via $\Psi$.

formulation as a discrete optimization problem suggests to use evolutionary algorithms in order to tackle (10). These algorithms are known to be particularly useful in such setting where only function calls are allowed. Based on this observation and on recent results in the field of randomized search algorithms (Lissovoi et al., 2023; Bendahi et al., 2025), we conceive a new randomized heuristic, the *MMAHH Solver*, tailored to problem ($\mathscr{P}'_d$) and compare it empirically with the well-established and general-purposes *Gurobi* solver (Gurobi Optimization, LLC, 2024) in Section 6.

Notes on the uniqueness of optimal solution(s) to the problem ($\mathscr{P}_d$) are provided in Appendix I.

## 5 APPLICATION TO ASYNCHRONOUS GD

We consider the following optimization problem

$$\min_{x \in \mathbb{R}^d} f(x), \tag{11}$$

where $f \colon \mathbb{R}^d \to \mathbb{R}$ is the objective to minimize. In the nonconvex setting, the goal is to find an $\varepsilon$–stationary point, i.e., a vector $x^*$ such that $\|\nabla f(x^*)\|^2 \leq \varepsilon$ (Nesterov & Polyak, 2006; Zhang et al., 2020). In practical scenarios, e.g., in machine learning, $f(x)$ denotes the loss of a model with weights $x$ on the training dataset.

### 5.1 PRESENTATION OF THE METHOD

Let us recall the well-known asynchronous GD (AGD) algorithm (Algorithm 1). For the sake of generality, we allow arbitrary non-negative stepsizes $\{\gamma_k\}_{k \geq 0}$ in the gradient descent step (line 8) contrary to the original version where the stepsizes are assumed to be constant. In the distributed framework under consideration, $n$ machines operate in parallel under the coordination of a central server. At the beginning of Algorithm 1, all workers start computing a stochastic gradient at a common initial point $x_0$ (line 6). Then the server enters a loop (assumed infinite for simplicity of the exposition and analysis) where it awaits and processes incoming gradient estimates from the workers as they complete their computations. At the beginning of the $k^{\text{th}}$ iteration of the **while** loop, a stochastic gradient $g_i^k$ is received from some worker $i \in [n]$ (line 7), and this gradient is applied to the sequence of iterates $\{x^k\}_{k \geq 0}$. We say the gradient $g_i^k$ is "accepted" by the server if $\gamma_k > 0$ otherwise, it is "discarded" ($\gamma_k = 0$) and $x^{k+1} = x^k - \gamma_k g_i^k = x^k$ so we do not move during $k^{\text{th}}$ loop. Additionally, in Algorithm 1, the delays $\{\delta^k\}_{k \geq 0}$ represent the total number of gradients the server receives between the time a worker starts computing and the time it sends its result. More precisely, if worker $i \in [n]$ sends a stochastic gradient to the server at iteration $k \geq 0$, then

$$\delta^k := k - \max\left\{r \in [1 .. k] \ : \ \mathcal{L}_W[r-1] = i\right\},$$

where $\mathcal{L}_W$ is an ordered list that records the history of gradient submissions by the workers. Specifically, for each iteration $k \geq 0$, the entry $\mathcal{L}_W[k] = i$ indicates that worker $i$ sent a stochastic gradient to the server at iteration $k$. This list allows us to determine, for any iteration $k$, when a particular worker $i$ last contributed a gradient, which is crucial for computing the corresponding delay $\delta^k$.

---

**Algorithm 1:** Asynchronous GD | **Procedure 1:** Workers' (infinite) loop
---|---

**Algorithm 1:** Asynchronous GD

1 **Initialization:**
2    $k \leftarrow 0$, the iteration counter
3    $x^0 \in \mathbb{R}^d$, the starting point
4    $\{\gamma_k\}_{k \geq 0}$, the stepsizes, $\gamma_k \geq 0$
5 Run **Procedure 1** in all workers
6 Send to all worker the point $x^0$
7 **while** true **do**
8    Wait until receiving $g_i^k := \nabla f\left(x^{k-\delta^k}\right)$ from worker $i$
     // Do one descent step.
9    $x^{k+1} \leftarrow x^k - \gamma_k g_i^k$
     // Reset the delay of worker $i$
10    Send to worker $i$ the point $x^{k+1}$
11    Update the iteration counter: $k \leftarrow k + 1$

**Procedure 1:** Workers' (infinite) loop

1 **while** true **do**
2    Wait until receiving $x^k \in \mathbb{R}^d$ from the server
     // May take some time.
3    Compute a full gradient $g \leftarrow \nabla f(x^k)$
4    Send $g$ to the server

Hence two natural questions arise: **(1)** what are the optimal "*gradient-independent*" stepsizes $\{\gamma_k^*\}_{k \geq 0}$ and **(2)** how do the hand crafted stepsizes compared to them? We investigate these two questions in the deterministic setting (i.e., no stochasticity) and, to the best of our knowledge, prove a first theoretical guarantee in this direction: AGD *with constant stepsizes and a tuned threshold[8] (to discard old gradients) leads to near-optimal theoretical performance.*

## 5.2 CONVERGENCE OF AGD IN THE NONCONVEX SETUP

We recall below the assumptions satisfied by the function $f$ from (11) and the stochastic gradients; these assumptions are standard in the analysis of SGD-type methods in the nonconvex setting (Ghadimi & Lan, 2013; Bottou et al., 2018).

**Assumption 5.1.** Function $f : \mathbb{R}^d \to \mathbb{R}$ is differentiable, and its gradients are $L$–Lipschitz continuous, i.e., $\|\nabla f(x) - \nabla f(y)\| \leq L \|x - y\|, \ \forall x, y \in \mathbb{R}^d$.

**Assumption 5.2.** There exist $f^{\inf} \in \mathbb{R}$ such that $f(x) \geq f^{\inf}$ for all $x \in \mathbb{R}^d$.

Based on Assumption 5.2, we define the initial sub-optimality $\Delta := f(x^0) - f^{\inf}$, where $x^0$ is the starting point of optimization method.

**Assumption 5.3.** The workers can compute *full* gradients, that is, when asked to compute a gradient of $f$ at $x \in \mathbb{R}^d$ they will reply, deterministically, $\nabla f(x)$ after some time.

**Main Result** We now state the convergence analysis of Algorithm 1: the proof builds on the state-of-the-art analysis of asynchronous methods (Mishchenko et al., 2022; Koloskova et al., 2022; Maranjyan et al., 2025; Tyurin & Sivtsov, 2025). As discussed in a subsequent paragraph, we further refine our analysis in Appendix G.9 and, as a byproduct of our general analysis, we recover with more transparency the convergence rate of Ringmaster ASGD (see Theorem G.14).

**Theorem 5.4** (Convergence Analysis of AGD). *Under Assumptions 5.1 to 5.3, for any integer $K \geq 0$ and any choice of non-negative stepsizes $\{\gamma_k\}_{k \geq 0}$ such that there exists $k \in [0..K]$ for which $\gamma_k > 0$, the iterates $\{x^k\}_{k \geq 0}$ of AGD (Algorithm 1) satisfy, with $\Gamma_K := \gamma_0 + \cdots + \gamma_K > 0$*

$$\frac{1}{\Gamma_K} \sum_{k=0}^{K} \gamma_k \left\| \nabla f\left(x^k\right) \right\|^2 \leq \frac{2\Delta}{\Gamma_K} + \underbrace{\frac{1}{\Gamma_K} \sum_{k=0}^{K} R_k \gamma_k \left\| \nabla f\left(x^{k-\delta^k}\right) \right\|^2}_{:= R(K)}, \tag{12}$$

*where $R_k := \gamma_k L + 2\gamma_k L^2 \sum_{j \in M_k} \gamma_j \, \delta^j - 1$ and $M_k := \left\{ j \in [0..K] : j - \delta^j \leq k \leq j - 1 \right\}$.*

**Link to the Optimization Problem** $(\mathscr{P}_d)$ According to the analysis done in Theorem 5.4, a natural approach to get rid of the $R(K)$ term in (12) is to ensure each $R_k \leq 0$, i.e.,

$$L\gamma_k + 2L^2 \gamma_k \sum_{j \in M_k} \gamma_j \, \delta^j - 1 \leq 0, \ \ k = 0, 1, \ldots, K \tag{13}$$

---

[8]Such an algorithm is considered in the work of Maranjyan et al. (2025) and the method is called Ringmaster ASGD.

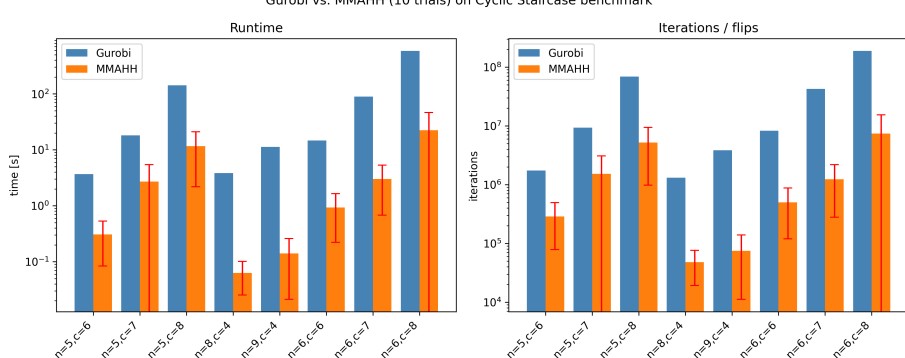

Figure 3: Comparison of solver runtime (left) and number of iterations (right) for Gurobi (blue) vs. MMAHH (orange) on the *Cyclic Staircase Benchmark*. For the MMAHH, means and standard deviations are taken over 10 runs.

and, if we let $M_{i,j} = 2\delta^j \mathbb{I}\{j \in M_i\}$ for all $i, j \in [0 .. K]$ then as $R(K) \leq 0$ by (13), and minimizing the right-hand side of (12) is equivalent to maximizing $\gamma_0 + \cdots + \gamma_K$ over

$$\mathscr{F} = \left\{ \Lambda \in [0,1]^{K+1} \, : \, 0 \leq L\Lambda + (L\Lambda) \odot (M^\delta[L\Lambda]) \leq 1 \right\},$$

where $\Lambda = (\gamma_0, \ldots, \gamma_K)$ and $M^\delta = (M_{i,j})_{i,j \in [0 .. K]}$ is the "matrix of delays" and we recover problem $(\mathscr{P}_d)$ with $\mathbf{a} = (1, \ldots, 1)^\top$ and $M = M^\delta$. Hence, optimal stepsizes in Algorithm 1 and satisfying (13) are obtained when solving this specific instance of $(\mathscr{P}_d)$.

**A Small Caveat** In Algorithm 1, the delay $\delta^k$ stays constant whether the gradient is accepted ($\gamma_k > 0$) or discarded ($\gamma_k = 0$): $\delta^k$ is only influenced by the workers' compute times and not how the gradients are selected. It seems much more natural (e.g., as in Ringmaster ASGD) for the delay to be the total number of *accepted* gradients, i.e., we define the *effective* delay $\widetilde{\delta}^k$ as

$$\widetilde{\delta}^k := \delta^k - \left| \left\{ j \in \left[ k - \delta^k .. k-1 \right] \, : \, \gamma_j = 0 \right\} \right| \leq \delta^k. \tag{14}$$

While Theorem 5.4 still holds with the delays $\{\widetilde{\delta}^k\}_{k \geq 0}$, (14) shows that the constraints (13) needs binary variables to be expressed and the optimization problem then becomes a *mixed-integer* nonlinear program. Nonetheless, we show in Appendix G.10 that we can still apply the main Theorem 4.4 and obtain the next result, proved in Appendix G.11. We refer to Appendix G for more details.

**Theorem 5.5** (Near Optimality of Ringmaster AGD)**.** *Under Assumptions 5.1, 5.2 and G.6, for any integer $K \geq 0$ the stepsizes $\{\gamma_k^{(R)}\}_{k \geq 0}$ of* Ringmaster AGD *(with a threshold[9] of $R = 1$) satisfy*

$$\sum_{k=0}^{K} \gamma_k^{(R)} \leq \sum_{k=0}^{K} \gamma_k^* \leq 2 \sum_{k=0}^{K} \gamma_k^{(R)},$$

*with $\{\gamma_k^*\}_{k \geq 0}$ the optimal stepsizes and $\gamma_k^{(R)} = \frac{1}{L}\mathbb{I}\left\{\widetilde{\delta}^k = 0\right\}$.*

In other word Theorem 5.5 asserts that once AGD, when ran with optimal stepsizes $\{\gamma_k^*\}_{k \geq 0}$, has found a $\varepsilon$–stationary point then Ringmaster AGD has provably found a $2\varepsilon$–stationary point. This proves that Ringmaster AGD achieve an approximation factor of 2.

## 6 EXPERIMENTAL RESULTS

**The** MMAHH **Solver.** The reformulation $(\mathscr{P}_d')$ of $(\mathscr{P}_d)$ in (10) reduces the original continuous optimization problem into a discrete one, suggesting the use of evolutionary algorithms. Based on

---

[9]Following the choice of Maranjyan et al. (2025), when $\sigma^2 = 0$ then $R = 1$.

this observation, we propose a new solver based on the recent Markov Move-Acceptance Hyper-Heuristic (MMAHH; Bendahi et al. (2025)). The MMAHH maintains a vector $x \in \{0, 1\}^d$ and flips one randomly chosen bit at each iteration to explore new candidates. Moreover, the MMAHH alternates between two search phases: ONLYIMPROVING (OI) where a move is accepted only if it improves the objective value, and ONLYWORSENING (OW) where a move is accepted only if it worsens the objective value. Two independent hyper-parameters $p$ and $q$ (the switching probabilities) are used to switch between the operators OI and OW. While there is no theoretically optimal values for $p$ and $q$, the choice $p = q = \mathcal{O}(1/(d \log d))$ seems to perform well in practice.

**Benchmarking Gurobi vs. MMAHH.** To benchmark its performance against a state-of-the-art solver, we compare the MMAHH to Gurobi 11 (Gurobi Optimization, LLC, 2024) on two families of instances: **(1)** the *Cyclic Staircase Benchmark* which corresponds to the case where workers periodically send a gradient to the server so that the list of worker's index $\mathcal{L}_W$ consists in repeating $[1, 2, \ldots, n]$ exactly $c$ times for some integers $n$ and $c$, e.g., with $n = 4$ and $c = 3$ the instance is $\mathcal{L}_W = [\underline{1, 2, 3, 4}, 1, 2, 3, 4, 1, 2, 3, 4]$, and **(2)** the *Stochastic Repetition Benchmark*, which consists of repeating a uniformly random sequence of length $n$ exactly $c$ times, allowing repetitions of workers. For $n = 4$ and $c = 3$ an instance can be $\mathcal{L}_W = [\underline{3, 4, 5, 4, 10}, 3, 4, 5, 4, 10, 3, 4, 5, 4, 10]$. Notice that for both benchmarks, the dimension of an instance with parameters $(n, c)$ is $d = nc$. Gurobi can solve the bilinear problem $(\mathscr{P}_d)$ via non-convex branch-and-bound and finds a *provable* global optima but at the cost of millions of simplex iterations and long runtimes. We run Gurobi once per instance and the MMAHH 10 independent trials to report the means and standard deviations for both wall-clock time and bit-flip counts. Across all tested instances $(n, c)$, MMAHH achieves better performance, reaching up to a $100\times$ speed-up in runtime while requiring up to $100\times$ less iterations on the *Cyclic Staircase Benchmark* (Figure 3). On the *Stochastic Repetition Benchmark*, MMAHH reaches speed-ups up to a $10^5\times$ factor in both runtime and number of iterations (see Appendix H).

**Landscape of the Discrete Function.** To give an idea of the landscape of the discrete function $\varphi(w) := \langle \mathbf{a} \mid \Psi(w) \rangle$ (for $w \in \{0, 1\}^d$) we optimize with the MMAHH solver, we represent $\varphi$ for $(n, c) = (5, 4)$ on the *Cyclic Staircase* and on the *Stochastic Repetition* benchmarks. We plot in Appendix H.3 the value of the $2^{30}$ bit-strings in $\{0, 1\}^{30}$. We group the points $w$ by their Hamming distance to the optimum $w^*$, more precisely, the $x$-axis corresponds to the quantity $30 - d_H(w, w^*)$, which is equal to 30 only for $w = w^*$ and to 0 only for $w = (w^*)^c$, where $(w^*)^c$ is the complementary bit-string of $w^*$, i.e., $(w^*)_i^c = 1 - w_i$ for all $i \in [d]$. The plots indicate that the discrete objective we optimize is not "monotonic across the layers" (see the definition in Appendix B.2), which unfortunately is outside the class of functions for which the theoretical work of Bendahi et al. (2025) applies. Nonetheless, we show that the MMAHH still achieves strong performance in practice on all these instances. This highlights a key advantage of hyper-heuristics: even when deployed outside their ideal theoretical framework (where guarantees hold) they can deliver excellent results, reflecting their inherently *heuristic* nature.

# 7 CONCLUSION

We presented a sharper characterization of the global maximizers in a class of bilinear programs arising naturally in the analysis of asynchronous gradient descent. Our main theoretical contribution shows that under general conditions, every global maximizer is extremal, reducing the search space from a continuous non-convex region to a finite set of structured vertices. This insight allows us to reformulate the original optimization problem into a discrete one over the vertices of unit hypercube, enabling the design of a randomized hyper-heuristic solver based on the recent MMAHH framework. Our experiments on the challenging *Cyclic Staircase* and *Stochastic Repetition* benchmarks demonstrate that a simple heuristic can already outperforms the commercial solver Gurobi by several orders of magnitude in both runtime and iteration count. These results highlight the practical and theoretical value of exploiting extremality in non-convex optimization and open the door to future work on applying combinatorial solvers and heuristics in non-convex settings.

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

CONTENTS

# A Notation

| Asymptotic | Meaning |
|---|---|
| $g = \mathrm{o}(f)$ (resp. $g = \omega(f)$) | When $g(n)/f(n) \xrightarrow[n\to+\infty]{} 0$ (resp. $+\infty$) |
| $g = \mathrm{O}(f)$ | There exists $C > 0$ such that $g(n) \leq Cf(n)$ for $n$ sufficiently large |
| $g = \Omega(f)$ | There exists $c > 0$ such that $g(n) \geq cf(n)$ for $n$ sufficiently large |
| $g = \Theta(f)$ | When both $g = \mathrm{O}(f)$ and $g = \Omega(f)$ |

| Sets and intervals | Meaning |
|---|---|
| $\mathbb{N}_0, \mathbb{N}$ | The set of non-negative (left) and positive (right) integers |
| $[a..b]$ $(a, b \in \mathbb{N}_0)$ | The set $[a, b] = \{a, a+1, \ldots, b-1, b\}$ |
| $[n]$ $(n \in \mathbb{N})$ | The set $[n] = [1, n] = \{1, 2, \ldots, n\}$ |

| Symbol | Meaning |
|---|---|
| $\mathbb{P}(\cdot), \mathbb{P}(\cdot|\cdot)$ | Probability and conditional probability |
| $\mathbb{E}[\cdot], \mathbb{E}[\cdot \mid \cdot]$ | Expectation and conditional expectation |

# B ADDITIONAL RELATED WORKS

## B.1 REVERSE-CONVEX PROGRAMMING (RCP)

RCP addresses global optimization over a convex feasible set with one or more reverse convex (complement of convex) constraints, resulting in highly nonconvex solution spaces. Classical theory provides foundational optimality and stability conditions, decomposition algorithms, and reduction approaches for RCPs Horst (1988); Tuy & Nguyen Duc (2000). Major algorithmic advances include cut-generating methods, polyhedral annexation, and intersection cut techniques for non-polyhedral settings (Towle & Luedtke, 2022; Yamada et al., 2000).

## B.2 HYPER-HEURISTICS

Hyper-heuristics, defined in Burke et al. (2013) as "*a search method or learning mechanism for selecting or generating heuristics to solve computational search problems*", emerged in early 2000s and quickly found numerous practical applications (Cowling et al., 2000; Ross et al., 2002; Chakhlevitch & Cowling, 2005; Garrido & Castro, 2009) notably to tackle NP-hard optimization tasks like scheduling, packing or routing problems (see the surveys Burke et al. (2003); Chakhlevitch & Cowling (2008); Burke et al. (2013; 2019)). While rigorous mathematical analysis of hyper-heuristics started only a decade ago (Lehre & Özcan, 2013), they have revealed intriguing results about their ability to solve optimization problems, notably on pseudo-Boolean functions $f \colon \{0,1\}^n \to \mathbb{R}$. Among them, selection hyper-heuristics (He et al., 2012; 2013; Alanazi & Lehre, 2014; Doerr et al., 2018; Lissovoi et al., 2019; 2020) and more recently the Move-Acceptance Hyper-Heuristic (MAHH) have gained attention for their remarkable efficiency in escaping local optima.

Based on this success, Bendahi et al. (2025) proposed an enhanced version of the MAHH: the Markov Move-Acceptance Hyper-Heuristic (MMAHH) with two enhancements that significantly improve the performance of the original MAHH across a broad range of functions. These two enhancements yields a significant runtime improvement and the authors derived a bound of $\mathcal{O}(n^{k+1} \log(n))$ on a wide class of functions: $\mathrm{SEQOPT}_k$.

We recall the next definitions from Bendahi et al. (2025) for clarity concerning the experiments.

**Definition B.1** (k-th Layer). Let $k \in [0..d]$ and $f \colon \{0,1\}^d \to \mathbb{R}$ such that $f$ admits a unique maximizer $x^* \in \{0,1\}^d$. The $k$-th layer $\mathcal{L}_k$ of $f$ is defined as:

$$\mathcal{L}_k := \{x \in \{0,1\}^n \mid d_H(x,x^*) = n - \|x\|_1 = k\}, \tag{15}$$

where we used $d_H(\cdot,\cdot)$ to denote the Hamming distance between two bit-strings. In other words, $\mathcal{L}_k$ is the set of all bit-strings at distance $k$ from the global maximum $x^*$ where the numbering starts at the global optimum, e.g., $\mathcal{L}_0 = \{x^*\}$, $\mathcal{L}_1$ are all bit-strings at Hamming distance $1$ from $x^*$, etc.

**Definition B.2** (Monotonicity across layers). Let $h \in [0..d-1]$ and $f \colon \{0,1\}^d \to \mathbb{R}$. We say that $f$ is increasing (resp. decreasing) between layers $\mathcal{L}_{h+1}$ and $\mathcal{L}_h$ if for any $y \in \mathcal{L}_{h+1}$ and any $x \in \mathcal{L}_h$ we have

$$f(y) < f(x) \text{ (resp. } f(y) > f(x)).$$

We denote this by $\mathcal{L}_{h+1} \overset{f}{\prec} \mathcal{L}_h$ (resp. $\mathcal{L}_{h+1} \overset{f}{\succ} \mathcal{L}_h$).

**Definition B.3** (The SEQOPT benchmark). Let $d \geq 2$ be an integer and $k \in [0..d-2]$. Let $d = d_0 > d_1 > d_2 > \cdots > d_k > d_{k+1} = 0$ be integers. We define $\mathrm{SEQOPT}_k(d_1,\ldots,d_k)$ to be the set of all functions $f \colon \{0,1\}^d \to \mathbb{R}$ such that $f$ has admits a unique maximizer $x^* \in \{0,1\}^d$ and for any $\ell \in [0..d]$,

  1. if $k - \ell$ is even then $f$ is increasing across $\mathcal{L}_{d_\ell},\ldots,\mathcal{L}_{d_{\ell+1}}$, i.e., $\mathcal{L}_{d_\ell} \overset{f}{\prec} \cdots \overset{f}{\prec} \mathcal{L}_{d_{\ell+1}}$,

  2. if $k - \ell$ is odd, $f$ is decreasing, i.e., it satisfies $\mathcal{L}_{d_\ell} \overset{f}{\succ} \cdots \overset{f}{\succ} \mathcal{L}_{d_{\ell+1}}$.

The union of these classes of functions, for fixed $k$, will be denoted by

$$\mathrm{SEQOPT}_k := \bigcup_{d > d_1 > \cdots > d_k > 0} \mathrm{SEQOPT}_k(d_1,\ldots,d_k).$$

## C  PRELIMINARIES AND USEFUL RESULTS

### C.1  DIFFEOMORPHISMS, CONVEX FUNCTIONS AND CONVEX SETS

**Definition C.1** ($\mathscr{C}^k$–Diffeomorphism). Let $U \subseteq \mathbb{R}^n$ and $V \subseteq \mathbb{R}^n$ be non-empty open sets. A map $f\colon U \to V$ is called a $\mathscr{C}^k$–*diffeomorphism* for integer $k > 0$ or $k = +\infty$ if

1. $f$ is bijective,

2. $f$ is of class $\mathscr{C}^k$ on $U$,

3. the inverse map $f^{-1}\colon V \to U$ and is $\mathscr{C}^k$ on $V$.

Equivalently, $f$ is a $\mathscr{C}^k$–diffeomorphism if $f$ is a bijection and both $f$ and $f^{-1}$ are $\mathscr{C}^k$ maps on their respective domains.

**Definition C.2** (Convex and Strictly Convex Function; Definitions 8.1 and 8.7 in Bauschke & Combettes (2017)). Let $C$ be a convex subset of $\mathbb{R}^d$, then the function $f\colon C \to \mathbb{R}$ is

* *convex* on $C$ if its epigraph

$$\mathrm{epi}(f) := \{(x, t) \in \mathbb{R}^d \times \mathbb{R} : t \geq f(x)\},$$

  is a convex subset of $\mathbb{R}^d \times \mathbb{R}$.

* *strictly convex*[10] on $C$ if for any $x, y \in C$ such that $x \neq y$ and for any $\lambda \in (0, 1)$ we have

$$f(\lambda x + (1 - \lambda y) < \lambda f(x) + (1 - \lambda) f(y).$$

**Lemma C.3** (Proposition 8.4 of Bauschke & Combettes (2017)). *Let $C$ be a convex subset of $\mathbb{R}^d$, then the function $f\colon C \to \mathbb{R}$ is convex on $C$ if for any $x, y \in C$ and any $\lambda \in (0, 1)$ we have*

$$f(\lambda x + (1 - \lambda)y) \leq \lambda f(x) + (1 - \lambda) f(y).$$

**Lemma C.4** (Composition of a Convex and a Linear Function). *Let $C$ be a convex subset of $\mathbb{R}^d$, $h\colon C \to \mathbb{R}$ be a linear function, $I \subseteq \mathbb{R}$ an open interval containing $h(C) \subseteq \mathbb{R}$ and let $g\colon I \to \mathbb{R}$ be a convex function then the map $f = g \circ h$ is convex on $C$.*

*Proof.* Note that the map $f = g \circ h\colon C \to \mathbb{R}$ is well-defined. Now, let $x, y \in C$ and let $\lambda \in (0, 1)$ then since $g$ is convex, by Lemma C.3 and by linearity of $h$ we have

$$
\begin{aligned}
f(\lambda x + (1 - \lambda)y) &= (g \circ h)(\lambda x + (1 - \lambda)y)\\
&\stackrel{\text{(a)}}{=} g(\lambda h(x) + (1 - \lambda)h(y))\\
&\stackrel{\text{Def. C.2}}{\leq} \lambda g(h(x)) + (1 - \lambda)g(h(y))\\
&= \lambda f(x) + (1 - \lambda)f(y),
\end{aligned}
$$

hence, $f$ is convex according to Lemma C.3. $\qquad\square$

Given any two points $a, b \in \mathbb{R}^d$, we denote by

$$[a, b] := \{ta + (1 - t)b : t \in [0, 1]\}, \tag{16}$$

the closed segment joining $a$ to $b$ and by

$$(a, b) := [a, b] \setminus \{a, b\} = \{tx + (1 - t)y : t \in (0, 1)\} \setminus \{a, b\}, \tag{17}$$

the *interior* of the segment $[a, b]$ or *open segment* from $a$ to $b$. Note that when $a = b$ we have both $[a, b] = \{a\}$ and $(a, b) = \varnothing$. More generally from (16) and (17) it follows

$$[a, b] \setminus (a, b) = \{a, b\}. \tag{18}$$

---

[10]To clarify, here the functions we consider always have a non-empty domain and they never take the value $\pm\infty$ hence, they are automatically *proper*. That is why we do not precise this in our definition, contrary to Bauschke & Combettes (2017).

## C.2 EXTREME POINTS

The notion of *extreme point* is often studied along with convex sets and convexity. Nonetheless, we can still extend the definition of extreme point from convex sets to, more generally, any subset of a linear space. In what follows we consider $S$ to be a non-empty subset of $\mathbb{R}^d$.

To the best of our knowledge, there are two ways to do this generalization and these approaches end up giving a slightly different meaning for what an "extreme point" is (actually, one definition is narrower than the other). In the literature, the most common approach is to define the concept of an *extremal set* of $S$ which we recall in Definition C.5. Besides, another option consists in defining a *support variety* of $S$ as stated in Definition C.6.

**Definition C.5** (Extremal Set; See Taylor & Lay (1980); Dunford & Schwartz (1988); Rudin (1991); Brezis (2010)). Let $S$ be a subset of a $\mathbb{R}^d$. A non-empty set $K \subseteq S$ is called an *extreme set* of $S$ if for any $x, y \in S$ and $t \in (0, 1)$ then $tx + (1 - t)y \in K$ if, and only if, $x \in K$ and $y \in K$.

Then following Definition C.5 an *extremal point* is defined as an extremal set which consists in just a single point.

**Definition C.6** (Support Variety; See Grothendieck (1973)). Let $S$ be a subset of a $\mathbb{R}^d$. A linear sub-variety $A$ (i.e., an affine subspace) of $\mathbb{R}^d$ is a *support variety* if $S \cap A \neq \varnothing$ and for every open segment $I \subseteq S$ whose interior meets $A$ then $I \subseteq A$.

Then, based on Definition C.5, an *extremal point* is defined as a (linear) support variety of dimension 0 (which is a single point).

We can see that the constraints which ensure a point $x \in S$ is extremal are more restrictive in Definition C.6 than in Definition C.5. More precisely, for a point $x \in S$ to be an extreme point, it must not be in the interior of any segment $[a, b] \subseteq S$ while in Definition C.5 it is only required that the endpoints $a$ end $b$ to be in $S$ and not whole segment $[a, b]$ anymore. Since it seems that the Definition C.5 has been more widely accepted and used in the literature, we then define *extreme points* following this definition.

Below we recall for clarity what we mean by an "extreme point" of a non-empty subset $S \subseteq \mathbb{R}^d$. This is the definition used throughout this paper, unless otherwise specified.

**Definition 3.2** (Extreme Point; Following Definition C.5). Let $S \subseteq \mathbb{R}^d$ be a non-empty subset, a point $x \in S$ is said to be an *extreme point* of $S$ if, for any $a, b \in S$ with $a \neq b$, the point $x$ does not lie in the interior of the segment $[a, b]$, that is, $x \notin (a, b)$.

The set of extreme points of $S$ is denoted by $\text{Extr } S$.

**Lemma C.7.** *The Definitions 3.2 and C.5 are equivalent.*

*Proof.* Let $S \subseteq \mathbb{R}^d$. Assume first $p \in S$ is an extreme point in the sense of Definition C.5. Given $x, y \in S$ we suppose for the sake of contradiction that $p \in (x, y)$, then necessarily $x \neq y$ (otherwise, if $x = y$ then $(x, y) = [x, x] \setminus x = \varnothing$ which is not possible) and by (17), there must exists $t \in (0, 1)$ such that $tx + (1 - t)y = p$ but then, since $p$ is an extreme point we must have $x = y = p$ which is a contradiction. Hence, we must have $p \notin (x, y)$.

Now, for the converse direction, let $p \in S$ to be an extreme point in the sense of Definition 3.2. Given $x, y \in S$ and any $t \in (0, 1)$, if $x = y = p$ then $tx + (1 - t)y = x = y = p$. For the other direction, if $tx + (1 - t)y = p$ then $p \in [x, y]$ and since $p$ is an extreme point, we have $p \notin (x, y)$ so

$$p \in [x, y] \setminus (x, y) \overset{(18)}{=} \{x, y\}.$$

Then, it remains to distinguish the cases $p = x$ or $p = y$. Without loss of generality, assume $p = x$ then from $tx + (1 - t)y = p$ we obtain $(1 - t)y = (1 - t)p$ thus $p = y$. Hence, $p = x = y$ which proves the equivalence of Definition C.5. $\square$

## C.3 CONVEX HULLS

Below, we recall both the definition of the convex hull and closed convex hull of a subset $S \subseteq \mathbb{R}^d$.

**Definition C.8** (Convex Hull and Closed Convex Hull). Let $S \subseteq \mathbb{R}^d$ then, the *convex hull* of $S$, denoted by $\text{Conv}\, S$ is defined as the smallest convex subset of $\mathbb{R}^d$ which contains $S$, alternatively,

$$\text{Conv}\, S := \bigcap_{\substack{C \subseteq \mathbb{R}^d,\, \text{convex} \\ S \subseteq C}} C.$$

The *closed convex hull* of $S$, denoted by $\overline{\text{Conv}}\, S$ is defined as the smallest closed convex subset of $\mathbb{R}^d$ which contains $S$, alternatively,

$$\overline{\text{Conv}}\, S := \bigcap_{\substack{C \subseteq \mathbb{R}^d,\, \text{closed and convex} \\ S \subseteq C}} C.$$

**Lemma C.9** (Closure of the Convex Hull of a Compact Set; Theorem 5.35 from Aliprantis & Border (2006)). *Let $S \subset \mathbb{R}^d$ be a compact set then, the closed convex hull of $S$, denoted by $\text{Conv}\, S$ is also a compact subset of $\mathbb{R}^d$.*

The next result is a special case of a partial "converse" of the Krein-Milman theorem formulated by Milman (1947). A general statement can be found in Phelps (2001) and in earlier works of Klee (1957; 1958). We state below the particular case of a compact subset of $\mathbb{R}^d$.

**Lemma C.10** (Lemma 3.4 of Chen et al. (2021)). *Let $S$ be a compact subset of $\mathbb{R}^d$ then*

$$\text{Extr}\left(\overline{\text{Conv}}\, S\right) \subseteq \text{Extr}\, S.$$

**Lemma C.11** (Extreme Points Always Exists on Non-empty Compact Sets). *Let $S$ be a non-empty compact subset of $\mathbb{R}^d$ then $\text{Extr}\, S \neq \varnothing$.*

*Proof.* Let $S \subseteq \mathbb{R}^d$ be a non-empty and compact set, consider the function $\|\cdot\|^2 : S \to \mathbb{R}$ then, as it is continuous over the compact $S$, the function $\|\cdot\|^2$ is bounded and it reaches its global maximum $M \in \mathbb{R}_+$, say, at some point $p \in S$. We now show that $p$ must be an extreme point of $S$. To do so, assume for the sake of contradiction that is it not the case so there exists $x, y \in S$ such that $p \in (x, y)$. Moreover, as $\|\cdot\|^2$ attains its global maximum at $p$ we must have $\|p\|^2 \geq \|x\|^2$ and $\|p\|^2 \geq \|y\|^2$. but, since $p \in (x, y)$ then, by definition (16) we have $p \neq x$ and $p \neq y$ and since the points $p$, $x$ and $y$ are aligned, there exists some vector $v \in \mathbb{R}^d \setminus \{0\}$ and scalars $t_x, t_y \in \mathbb{R}^*$ such that $t_x t_y < 0$[11] and

$$x = p + t_x v \quad \text{and} \quad y = p + t_y v.$$

Now, we distinguish two cases:

- if $\langle p \mid v \rangle = 0$ then expanding $\|x\|^2$ we obtain

$$\begin{aligned}
\|x\|^2 &= \|p + t_x v\|^2 \\
&= \|p\|^2 + 2t_x \langle p \mid v \rangle + t_x^2 \|v\|^2 \\
&= \|p\|^2 + t_x^2 \|v\|^2 \\
&> \|p\|^2,
\end{aligned} \tag{19}$$

  since by assumption we have $v \neq 0$ and the scalar $t_x \neq 0$ (because $p \neq 0$). We see that the inequality (19) is contradictory about the maximality of $\|\cdot\|^2$ on $S$.

- if $\langle p \mid v \rangle \neq 0$ then, without loss of generality we may assume $\langle p \mid v \rangle > 0$ and since $t_x t_y < 0$ then, one of them must be positive, say without loss of generality it is $t_x > 0$ and, expanding $\|x\|^2$ again gives

$$\|x\|^2 = \|p\|^2 + 2t_x \langle p \mid v \rangle + t_x^2 \|v\|^2 > \|p\|^2, \tag{20}$$

  because both quantities $2t_x \langle p \mid v \rangle$ and $t_x^2 \|v\|^2$ are positive. This is again a contradiction.

---

[11] Both scalars $t_x$ and $t_y$ are non-zero since $p \neq x$ and $p \neq y$. Moreover, they must have opposite sign since $p$ lies in the interior of the segment $[x, y]$, that is, $x$ and $y$ are on the opposite side of $p$ on the line $[x, y]$.

Thus we conclude that the point $p$ cannot lie in the interior of the segment $[x, y]$, and this holds true for any points $x, y \in S$ so according to Definition 3.2 $p$ must be an extreme point of $S$, i.e., $p \in \operatorname{Extr} S \neq \varnothing$. $\qquad\square$

## C.4 SUPPORT HYPERPLANES

We now recall some results concerning the support hyperplanes of a convex subset $C$ of $\mathbb{R}^d$.

**Definition C.12** (Supporting Hyperplane). Let $C \subseteq \mathbb{R}^d$ be a convex subset. We say that an (affine) hyperplane $H$ is a *supporting hyperplane* of $C$ at point $p \in \partial C$ if, and only if there exists some vector $a \in \mathbb{R}^d \setminus \{(0, \ldots, 0)\}$ such that

$$H = \left\{ x \in \mathbb{R}^d \, : \, \langle a \mid x \rangle = \langle a \mid p \rangle \right\}$$

and $\langle a \mid x \rangle \geq \langle a \mid p \rangle$ for all $x \in C$.

In other word, there exists an affine hyperplane which meets $p$ and for which the convex set $C$ is included in one of its two closed half-spaces:

$$H^+ := \left\{ x \in \mathbb{R}^d \, : \, \langle a \mid x \rangle \geq \langle a \mid p \rangle \right\}, \tag{21}$$

or

$$H^- := \left\{ x \in \mathbb{R}^d \, : \, \langle a \mid x \rangle \leq \langle a \mid p \rangle \right\}. \tag{22}$$

**Lemma C.13** (Supporting Hyperplane Theorem). *For any non-empty convex subset $C \subseteq \mathbb{R}^d$ and any $p \in \partial C$ there exists a supporting hyperplane of $C$ at point $p$.*

A refined version of the supporting hyperplane theorem above, for the case of convex subsets which are level-sets of convex functions, is provided below. Notably, it provides the uniqueness of the supporting hyperplane.

**Lemma C.14** (Theorem 3.1 of He & Xu (2013); Case of $H = \mathbb{R}^d$). *Let $\varphi \colon \mathbb{R}^d \to \mathbb{R}$ be a real-valued, continuous and convex function which is differentiable[12] on $\mathbb{R}^d$, then the level set*

$$C := \left\{ x \in \mathbb{R}^d \, : \, \varphi(x) \leq 0 \right\},$$

*is convex and for each point $p \in \partial C$ there exists a unique supporting hyperplane of $C$ at $p$. Moreover, this supporting hyperplane is given by*

$$H = \left\{ x \in \mathbb{R}^d \, : \, \langle \nabla \varphi(p) \mid x - p \rangle = 0 \right\}.$$

**Lemma C.15** (Intersection of a Family of Affine Hyperplanes). *Let $k \in [d]$ be an integer, $v_1, \ldots, v_k \in \mathbb{R}^d$ be vectors, $a_1, \ldots, a_k \in \mathbb{R}$ some scalars and $H_1, \ldots, H_k$ be the $k$ affine hyperplanes of $\mathbb{R}^d$ associated to the linear forms $(\langle v_i \mid \cdot \rangle)_{i \in [k]}$, that is, for any $i \in [k]$*

$$H_i := \left\{ x \in \mathbb{R}^d \, : \, \langle v_i \mid x \rangle = a_i \right\}.$$

*If $A := \bigcap_{i \in [k]} H_i \neq \varnothing$ then, $\dim A \geq d - k$.*

*Proof.* By assumption $\bigcap_{i \in [k]} H_i \neq \varnothing$ hence, the system

$$\langle v_i \mid x \rangle = a_i, \ \ i = 1, 2, \ldots, k, \tag{23}$$

consisting of $k$ equation has a solution $x_0 \in \mathbb{R}^d$. Then for all $i \in [k]$, if we subtract $\langle v_i \mid x_0 \rangle$ in the $i^{\text{th}}$ equation from the system (23), we obtain the equivalent system

$$\langle v_i \mid x - x_0 \rangle = 0, \ \ i = 1, 2, \ldots, k,$$

hence $(x - x_0) \in \{v_1, \ldots, v_k\}^{\perp}$ so $x - x_0$ belongs to the subspace of $\mathbb{R}^d$ orthogonal to each $(v_i)_{i \in [k]}$. Hence, we deduce that

$$A := \bigcap_{i \in [k]} H_i = x_0 + \{v_1, \ldots, v_k\}^{\perp},$$

which is a subspace of $\mathbb{R}^d$ whose dimension is

$$\dim A = \dim \left( \{v_1, \ldots, v_k\}^{\perp} \right) \geq d - k,$$

since the rank of the family $(v_1, \ldots, v_k)$ is at most $k$. This concludes the proof of the lemma. $\qquad\square$

---

[12]More precisely, *Gateaux differentiable* which means that $\varphi$ has a gradient at all point $x \in \mathbb{R}^d$.

## C.5 $P$-MATRICES AND INJECTIVITY

In this section, we present a very practical sufficient condition of injectivity of functions $f$ defined from $\mathbb{R}^d$ to $\mathbb{R}^d$. This condition is captured by the celebrated *Gale–Nikaidô theorem*, a cornerstone of global analysis and mathematical economics. Detailed proofs and broader context for this result can be found in Gale & Nikaido (1965) and Okuguchi (1978), as well as in later expositions within applied mathematics and dynamical systems (Banaji et al., 2007).

We start with some fundamental definitions commonly referenced in linear algebra and matrix theory:

**Definition C.16** (Minors and Principal Minors of a Matrix). A *minor* of a matrix $A \in \mathbb{R}^{d \times d}$ is the determinant of some square sub-matrix of $A$ obtained by removing one or more of its rows and columns. If $I$ and $J$ are (ordered) subsets of $[d]$ with $k$ elements (where $1 \leq k \leq d$), then we denote by $[A]_{I,J}$ the $k \times k$ minor of $A$ that corresponds to the intersection of the rows and columns of $A$ whose indices are taken in $I$ and in $J$ respectively.

When $I = J$, the minor $[A]_{I,I}$ is called a *principal minor*.

**Definition C.17** ((Positive) Dominant Diagonal). Let $A$ be a $\mathbb{R}^{d \times d}$ matrix, then $A$ has a *dominant diagonal* if, and only if there exists $d$ positive real numbers $\alpha_1, \ldots, \alpha_d > 0$ such that for all $i \in [n]$ the inequality

$$\alpha_i \left| A_{i,i} \right| > \sum_{\substack{j=1 \\ j \neq i}}^{d} \alpha_j \left| A_{i,j} \right|, \tag{24}$$

holds.

Additionally, if for all $i \in [n]$ we have $A_{i,i} > 0$, i.e., $A$ has positive diagonal entries then $A$ has a *positive dominant diagonal*.

**Definition C.18** ($P$-matrix). A real matrix $A \in \mathbb{R}^d$ is said to be a *P-matrix* if, and only if, all its principal minors are positive.

**Definition C.19** (Region and Closed Rectangular Region). A *region* is an connected set in $\mathbb{R}^d$, either without its boundary or together with its boundary.

A closed *rectangular region* is a subset of $\mathbb{R}^d$ of the form

$$\left\{ x \in \mathbb{R}^d \, : \, \forall i \in [d], p_i \leq x_i \leq q_i \right\},$$

where $-\infty \leq p_i < q_i \leq +\infty$ are numbers (possibly $\pm\infty$).

A key property relevant to our context is the following classical result:

**Lemma C.20** (Positive Dominant Diagonal Implies $P$-Matrix). *Let $A$ be a matrix in $\mathbb{R}^{d \times d}$ such that $A$ has a positive dominant diagonal, then $A$ is a P-matrix.*

The foundational theorem that links $P$-matrices to injectivity is as follows:

**Theorem C.21** (Gale–Nikaidô). *Let $\Omega$ be a closed rectangular region of $\mathbb{R}^d$. If $F \colon \Omega \to \mathbb{R}^d$ is a differentiable function such that its Jacobian matrix $\nabla F(x)$ is a P-matrix for all $x \in \Omega$, then $F$ is injective on $\Omega$, i.e., if $a, b \in \Omega$ are such that $F(a) = F(b)$ then necessarily, $a = b$.*

These results, originally developed in the seminal paper by Gale & Nikaido (1965), have extensive applications in nonlinear analysis, mathematical economics, chemical reaction networks, and beyond.

## C.6 FIXED-POINT THEOREMS

Fixed-point theorems are foundational tools in nonlinear analysis, optimization, game theory, and mathematical economics. These theorems assert that, under certain topological or algebraic conditions, a mapping admits a point that is mapped to itself. In particular, we focus here on the

classical *Brouwer* fixed-point theorem, which forms the backbone of many existence proofs in high-dimensional non-convex settings. Comprehensive treatments of this result can be found in standard texts such as Brouwer (1911); Border (1985); Granas & Dugundji (2003).

We state the central result in finite-dimensional topological fixed-point theory:

**Theorem C.22** (Brouwer Fixed-Point Theorem). *Let $D \subset \mathbb{R}^d$ be a non-empty, compact, convex subset. Then any continuous function $f : D \to D$ has at least one fixed-point in D, i.e., there exists $x^* \in D$ such that $f(x^*) = x^*$.*

## C.7 Implicit Functions Theorem

The Implicit Function Theorem (IFT) is a cornerstone result in multivariable calculus and nonlinear analysis. It gives conditions under which a system of equations implicitly defines one set of variables as functions of another. The theorem ensures local solvability and differentiability of these implicit functions under mild regularity conditions. This result underpins much of optimization theory, differential equations, and dynamical systems. For formal treatments, see Rudin (1976); Lang (1995); Krantz & Parks (2002).

**Theorem C.23** (Implicit Functions Theorem). *Let $F : \mathbb{R}^{n+m} \to \mathbb{R}^m$ be a continuously differentiable on an open set $U \subset \mathbb{R}^{n+m}$, and let $(x_0, y_0) \in U$ such thta $F(x_0, y_0) = 0$. Suppose the Jacobian matrix $\nabla_y F(x_0, y_0) \in \mathbb{R}^{m \times m}$ is invertible. Then there exist open neighborhoods $V \subset \mathbb{R}^n$ of $x_0$ and $W \subset \mathbb{R}^m$ of $y_0$, and a unique continuously differentiable function $g : V \to W$ such that:*

$$F(x, g(x)) = 0 \quad \text{for all } x \in V. \tag{25}$$

In essence, the theorem guarantees the local solvability of the system $F(x, y) = 0$ for $y$ in terms of $x$, assuming local nonsingularity of the Jacobian with respect to $y$.

*Remark C.24.* When $F$ is infinitely differentiable, i.e., $\mathcal{C}^\infty$, then the function $g$ in the previous theorem inherits the same regularity property.

## C.8 Useful Identities and Inequalities

For any vectors $x, y \in \mathbb{R}^d$, we have

$$2 \langle x \mid y \rangle = \|x\|^2 + \|y\|^2 - \|x - y\|^2. \tag{26}$$

**Lemma C.25** ($L$–Lipchitz Gradients Implies $L$–Smoothness (Nesterov, 2018, Lemma 1.2.3, p. 25)). *Let $f : \mathbb{R}^d \to \mathbb{R}$ be continuously differentiable such that $f$ has $L$–Lipchitz gradients then, for any $x, y \in \mathbb{R}^d$*

$$-L \|x - y\|^2 \le 2D_f(x, y) \le L \|x - y\|^2,$$

*where $D_f(x, y) := f(x) - f(y) - \langle \nabla f(y) \mid x - y \rangle$ is the Bregman divergence of $f$ at $x$ and $y$.*

**Lemma C.26** (Variance Decomposition). *For any random vector $X \in \mathbb{R}^d$ and any non-random vector $c \in \mathbb{R}^d$ we have*

$$\mathbb{E}\left[\|X - c\|^2\right] = \mathbb{E}\left[\|X - \mathbb{E}[X]\|^2\right] + \|\mathbb{E}[X] - c\|^2.$$

**Lemma C.27** (Tower Property of the Expectation). *For any random variables $X \in \mathbb{R}^d$ and $Y_1, \ldots, Y_n$ we have*

$$\mathbb{E}[\mathbb{E}[X \mid Y_1, \ldots, Y_n]] = \mathbb{E}[X].$$

**Lemma C.28** (Cauchy Schwarz's Inequality). *For any vectors $a, b \in \mathbb{R}^d$ we have*

$$\langle a \mid b \rangle \le |\langle a \mid b \rangle| \le \|a\| \|b\|.$$

**Lemma C.29** (Young's inequality (Norm Form)). *For any vectors $a, b \in \mathbb{R}^d$ and any scalar $\alpha > 0$ we have*

$$\|a + b\|^2 \le (1 + \alpha) \|x\|^2 + \left(1 + \frac{1}{\alpha}\right) \|y\|^2.$$

**Lemma C.30** (Bounded Variance of Pairwise Independent Stochastic Gradients). *Under Assumption G.5, let $x_1, \ldots, x_n \in \mathbb{R}^d$ be non-random vectors and $\alpha_1, \ldots, \alpha_n \in \mathbb{R}$ be scalars then for any pairwise independent random variables $\xi_1, \ldots, \xi_n \sim \mathcal{D}$ we have*

$$\mathbb{E}\left[\left\|\sum_{i=1}^n \alpha_i \left(\nabla f(x_i, \xi_i) - \nabla f(x_i)\right)\right\|^2\right] = \sum_{i=1}^n \alpha_i^2 \, \mathbb{E}\left[\|\nabla f(x_i, \xi_i) - \nabla f(x_i)\|^2\right] \leq \sigma^2 \sum_{i=1}^n \alpha_i^2. \tag{27}$$

*Proof.* Expanding the squared norm in left-hand side of (27) (for now, without taking the expectation in account) we get

$$\left\|\sum_{i=1}^n \alpha_i \left(\nabla f(x_i; \xi_i) - \nabla f(x_i)\right)\right\|^2$$
$$= \sum_{i=1}^n \alpha_i^2 \|\nabla f(x_i, \xi_i) - \nabla f(x_i)\|^2$$
$$+ \sum_{\substack{1 \leq i,j \leq n \\ i \neq j}} \alpha_i \alpha_j \left\langle \nabla f(x_i, \xi_i) - \nabla f(x_i) \mid \nabla f(x_j, \xi_j) - \nabla f(x_j) \right\rangle, \tag{28}$$

and for any $1 \leq i, j \leq n$ such that $i \neq j$ we have

$$\mathbb{E}\left[\langle \nabla f(x_i; \xi_i) - \nabla f(x_i) \mid \nabla f(x_j; \xi_j) - \nabla f(x_j) \rangle\right]$$
$$\overset{\text{(a)}}{=} \langle \mathbb{E}\left[f(x_i, \xi_i) - \nabla f(x_i)\right] \mid \mathbb{E}\left[\nabla f(x_j, \xi_j) - \nabla f(x_j)\right]\rangle$$
$$\overset{\text{Ass. G.6}}{=} 0,$$

where in (a) we use the pairwise independence of the stochastic gradients while in the second equality we rely on the unbiasedness of the stochastic gradients (Assumption G.5) to get rid of the above cross-product. Hence, taking the expectation in (28) gives

$$\mathbb{E}\left[\left\|\sum_{i=1}^n \alpha_i \left(\nabla f(x_i, \xi_i) - \nabla f(x_i)\right)\right\|^2\right] = \sum_{i=1}^n \alpha_i^2 \, \mathbb{E}\left[\|\nabla f(x_i, \xi_i) - \nabla f(x_i)\|^2\right] \overset{\text{Ass. G.5}}{\leq} \sigma^2 \sum_{i=1}^n \alpha_i^2,$$

as desired. $\qquad\square$

**Lemma C.31** (Jensen's Inequality). *Let $f \colon \mathbb{R}^d \to \mathbb{R}$ be a convex function then*

*1. (probabilistic form) for any random vector $X \in \mathbb{R}^d$ we have*
$$\mathbb{E}\left[f(X)\right] \geq f\left(\mathbb{E}\left[X\right]\right).$$

*2. (deterministic form) for any vectors $v_1, \ldots, v_n \in \mathbb{R}^d$ and scalars $\lambda_1, \ldots, \lambda_n \in \mathbb{R}$ we have*
$$\sum_{i=1}^n \lambda_i f(v_i) \geq f\left(\sum_{i=1}^n \lambda_i v_i\right),$$

*provided $\lambda_i \geq 0$ for all $i \in [n]$ and $\sum_{i=1}^n \lambda_i = 1$.*

**Lemma C.32.** *For any vectors $v_1, \ldots, v_n \in \mathbb{R}^d$ we have*
$$\left\|\sum_{i=1}^n v_i\right\|^2 \leq n \sum_{i=1}^n \|v_i\|^2.$$

*Proof.* The function $\|\cdot\|^2 : \mathbb{R}^d \to \mathbb{R}$ is $\mu$-strongly convex with $\mu = 2$ so is convex thus applying Jensen's inequality (Lemma C.31) with $\lambda_1 = \cdots = \lambda_n = \frac{1}{n}$ gives

$$\left\|\sum_{i=1}^n \frac{v_i}{n}\right\|^2 \leq \frac{1}{n} \sum_{i=1}^n \|v_i\|^2,$$

and multiplying both sides by $n^2$ gives the desired result. $\qquad\square$

*Remark* C.33. Note that we can obtain the following improved upper bound from Lemma C.32; for any vectors $v_1, \ldots, v_n \in \mathbb{R}^d$, let $\boldsymbol{v} = (v_1, \ldots, v_n)$ then, we have

$$\left\| \sum_{i=1}^{n} v_i \right\|^2 \leq |\mathrm{supp}\, \boldsymbol{v}| \cdot \sum_{i=1}^{n} \|v_i\|^2, \tag{29}$$

where $\mathrm{supp}\, \boldsymbol{v} := \{i \in [n] : v_i \neq 0\}$ is the set of non-zero vectors among $v_1, \ldots, v_n$.

**Lemma C.34** (Switching Two Nested Sums). *Let $S$ be a finite set (possible empty[13]) and for every $k \in S$, let $S(k)$ be another, eventually empty, finite set. For any $k \in S$ and any $j \in S(k)$ let $C_{k,j}$ be a real number then*

$$\sum_{k \in S} \sum_{j \in S(k)} C_{k,j} = \sum_{j \in S'} \sum_{k \in S'(j)} C_{k,j}, \tag{30}$$

*where $S'$ is a finite set such $\bigcup_{k \in S} S(k) \subseteq S'$ and*

$$S'(j) := \{k \in S : j \in S(k)\}.$$

*Proof.* First, note that since $S$ is finite and since each $S(k)$ for $k \in S$ is finite then $\bigcup_{k \in S} S(k)$ is also finite and a finite set $S'$ containing the union of the $\{S(k)\}_{k \in S}$ exists. Moreover, if there exists $j \in S' \setminus \bigcup_{k \in S} S(k)$ then by definition

$$S'(j) := \{k \in S : j \in S(k)\} = \varnothing,$$

so taking a bigger $S'$ doesn't affect the right-hand side of (30) hence, without loss of generality assume

$$S' = \bigcup_{k \in S} S(k).$$

Now let us define the sets

$$E := \{(k, j) \,:\, k \in S,\, j \in S(k)\},$$

and

$$E' := \{(j, k) \,:\, j \in S',\, k \in S'(j)\}$$

then the map $\phi \colon E \to E'$ is well-defined since for any $(k, j) \in E$ we have $j \in S(k) \subseteq S'$ and because $k \in S$ and $j \in S(k)$ then by definition of $S'(j)$ we also have $k \in S'(j)$ thus $(j, k) \in E'$. Moreover, the map $\phi$ is injective because, if $(j, k) = \phi(k, j) = \phi(k', j') = (j', k')$ for some $(k, j), (k', j') \in E$ then $j = j'$ and $k = k'$. Also, $\phi$ is surjective since, given $(j, k) \in E'$ we have $j \in S'$ and $k \in S'(j)$ by definition of $E'$, then as $k \in S'(j)$ we deduce that $k \in S$ and $j \in S(k)$ so $(k, j) \in E$ and $\phi(k, j) = (j, k)$ so $(j, k)$ admits an antecedent by $\phi$ in $E$. This shows that the map $\phi$ is bijective hence

$$\sum_{(k,j) \in E} C_{k,j} = \sum_{(j,k) \in \phi(E)} C_{k,j} = \sum_{(j,k) \in E'} C_{k,j},$$

thus, since

$$\sum_{k \in S} \sum_{j \in S(k)} C_{k,j} = \sum_{(k,j) \in E} C_{k,j},$$

and

$$\sum_{j \in S'} \sum_{k \in S'(j)} C_{k,j} = \sum_{(j,k) \in E'} C_{k,j},$$

we deduce that equality (30) holds. $\square$

---

[13] By convention any sum $\sum_{k \in \varnothing} \cdot$ over the empty set is equals to zero.

# D   TECHNICAL LEMMAS

## D.1   PRELIMINARY LEMMAS

**Lemma D.1** (A Convex Function on $\mathbb{R}^d$). *Let $d$ be a positive integer, $\boldsymbol{\alpha} = \{\alpha_j\}_{1 \leq j \leq d}$ be $d$ non-negative real numbers and $C = \mathcal{H}_{\boldsymbol{\alpha}}^+$ be the open half-space above the hyperplane $\mathcal{H}_{\boldsymbol{\alpha}} := \{\mathbf{x} = (x_1, \ldots, x_d) \in \mathbb{R}^d : \langle \mathbf{x} \mid \boldsymbol{\alpha} \rangle = -1\}$ so that $C$ is defined as*

$$C := \left\{ (x_1, \ldots, x_d) \in \mathbb{R}^d : \sum_{j=1}^d \alpha_j \, x_j > -1 \right\}. \tag{31}$$

*Then $C$ is a convex subset of $\mathbb{R}^d$ and the function $f : C \to \mathbb{R}_+^d$ defined as*

$$f : (x_1, \ldots, x_d) \mapsto \left( 1 + \sum_{j=1}^d \alpha_j x_j \right)^{-1}, \tag{32}$$

*is smooth, i.e., $f \in \mathcal{C}^\infty(C, \mathbb{R}_+^d)$[14] and convex[15] on $C$.*

*Proof of Lemma D.1.* We first show that $C$ is an open convex subset of $\mathbb{R}^d$. Note that for any $x, y \in C$ and any $t \in [0, 1]$, we have

$$\langle tx + (1-t)y \mid \boldsymbol{\alpha} \rangle = t \langle x \mid \boldsymbol{\alpha} \rangle + (1-t) \langle y \mid \boldsymbol{\alpha} \rangle > -1,$$

since both $\langle x \mid \boldsymbol{\alpha} \rangle - 1$ and $\langle y \mid \boldsymbol{\alpha} \rangle > -1$ and because $\max \{t, 1-t\} \geq \frac{1}{2} > 0$ so none of the two terms can simultaneously vanish due to the variable $t$; this proves that the closed segment $[x, y] \subseteq C$ hence $C$ is convex. Moreover, to prove $C$ is an open subset of $\mathbb{R}^d$, let $x \in C$ so we can define the positive real number

$$\varepsilon := \sum_{j=1}^d \alpha_j x_j + 1 > 0,$$

Now, we argue that the open ball $\mathrm{B}(x, r)$ where $r = \frac{\varepsilon}{(1+\|\boldsymbol{\alpha}\|_\infty)d} > 0$ is included in $C$. Here, we consider $\mathbb{R}^d$ equipped with its usual euclidean norm $\| \cdot \|_2$ and we denote by $\| \cdot \|_\infty$ the supremum norm, that is, for any $x = (x_1, \ldots, x_d) \in \mathbb{R}^d$ we have $\|x\|_\infty = \sup_{i \in [d]} |x_i|$. To do so, let $y = (y_1, \ldots, y_d) \in \mathrm{B}(x, r)$ and define $v = (v_1, \ldots, v_d) := y - x \in \mathrm{B}(0, r)$ then

$$\sum_{j=1}^d \alpha_j y_j = \sum_{j=1}^d \alpha_j (x_j + v_j) \tag{33}$$

$$= \sum_{j=1}^d \alpha_j x_j + \sum_{j=1}^d \alpha_j v_j$$

$$\overset{(a)}{\geq} \sum_{j=1}^d \alpha_j x_j - \|\boldsymbol{\alpha}\|_\infty \sum_{j=1}^d |v_j|$$

$$\overset{(b)}{\geq} \sum_{j=1}^d \alpha_j x_j - d\|\boldsymbol{\alpha}\|_\infty \|v\|_\infty$$

$$= -1 + \varepsilon - d\|\boldsymbol{\alpha}\|_\infty \|v\|_\infty,$$

where in (a) we lower bound the right sum as

$$\sum_{j=1}^d \alpha_j v_j \geq - \sum_{j=1}^d |\alpha_j| \, |v_j|,$$

---

[14]By this we mean that the function $f$ defined from $C \to \mathbb{R}_+^d$ is infinitely differentiable.

[15]For the sake of clarity and completeness, we included a definition of convexity in the appendix (see Definition C.2 along with the usual inequality characterizing convex functions $f : \mathbb{R}^d \to \mathbb{R}$ in Lemma C.3.

and then we lower bound again using the inequality $\|\boldsymbol{\alpha}\|_\infty \geq |\alpha_j|$ for all $j \in [d]$. In (b) we use the inequality $\|v\|_\infty \geq |v_j|$ for all $j \in [d]$ to lower bound the sum by $d\|v\|_\infty$. Now, since $v \in \mathrm{B}(0,r)$ we have

$$\|v\|_2 = \sqrt{\sum_{j=1}^{d} |v_j|^2} \geq \|v\|_\infty,$$

hence

$$\begin{aligned}
\varepsilon - d\|\boldsymbol{\alpha}\|_\infty\|v\|_\infty &\geq \varepsilon - d\|\boldsymbol{\alpha}\|_\infty\|v\|_2 \\
&\geq \varepsilon - \frac{\varepsilon}{(1 + \|\boldsymbol{\alpha}\|_\infty)d} \cdot d\|\boldsymbol{\alpha}\|_\infty \\
&= \varepsilon\left(1 - \frac{\|\boldsymbol{\alpha}\|_\infty}{1 + \|\boldsymbol{\alpha}\|_\infty}\right) > 0,
\end{aligned}$$

because $\|\boldsymbol{\alpha}\|_\infty \geq 0$ and thus the quantity in (33) is lower bounded by

$$\sum_{j=1}^{d} \alpha_j y_j \geq -1 + \varepsilon\left(1 - \frac{\|\boldsymbol{\alpha}\|_\infty}{1 + \|\boldsymbol{\alpha}\|_\infty}\right) > -1,$$

which implies that $y \in C$ and since this holds for any $y \in \mathrm{B}(x,r)$ then $\mathrm{B}(x,r) \subseteq C$ as desired.

Now, for the other part of the lemma, note that the function $f\colon C \to \mathbb{R}_+$ is well-defined and smooth, that is, $\mathcal{C}^\infty$ on its domain. Note that for any $(x_1, \ldots, x_d) \in C$, the function $f$ can be rewritten as

$$f(x_1, \ldots, x_d) = g(h(x_1, \ldots, x_d)),$$

where $g\colon \mathbb{R}_+^* \to \mathbb{R}_+^*$ is the inverse function, that is, $g\colon x \mapsto \frac{1}{x}$ and $h\colon C \to (0, +\infty)$ is the linear functional

$$h\colon (x_1, \ldots, x_d) \mapsto 1 + \sum_{j=1}^{d} \alpha_j x_j. \tag{34}$$

Using this, for any $(x_1, \ldots, x_d) \in C$ and thanks to the non-negativity of the coefficients $\{\alpha_j\}_{1 \leq j \leq d}$ and the definition of $C$, the following inequality holds:

$$h(x_1, \ldots, x_d) = 1 + \sum_{j=1}^{d} \alpha_j x_j > 0,$$

hence $f$ is well-defined on its domain since $h$ takes its values in $(0, +\infty)$. Moreover as the function $g$ is strictly decreasing over $\mathbb{R}_+^*$, we obtain $0 < f(x_1, \ldots, x_d) < +\infty$[16]. Additionally, because both $h$ and $g$ are $\mathcal{C}^\infty$ functions respectively from $C \to (0, +\infty)$ and from $\mathbb{R}_+^* \to \mathbb{R}_+^*$ then their composition $f = g \circ h$ is also a $\mathcal{C}^\infty$ function from $C \to \mathbb{R}_+^*$.

Now, we show that $f$ is convex on its domain. From (34), we see that $h$ is linear in $x_1, \ldots, x_d$ from $C \to (0, +\infty)$, and since $g\colon x \mapsto \frac{1}{x}$ is strictly convex[17] on $(0, +\infty)$ then it is convex and we can conclude using Lemma C.4 that the composition

$$f = g \circ h,$$

is a convex function from $C \to (0, +\infty)$.

This completes the proof of the lemma. $\qquad\square$

---

[16]Thus $f$ is a proper function ($f$ never takes the value $+\infty$ on its domain).

[17]It suffice to compute the first and second derivative of $g$. Since $g\colon x \mapsto \frac{1}{x}$ is $\mathcal{C}^\infty$ these derivatives are well-defined and for any real number $x > 0$

$$g'(x) = -\frac{1}{x^2} \quad \text{and} \quad g''(x) = \frac{2}{x^3},$$

thus $g'(x) < 0$ and $g''(x) > 0$ on $(0, +\infty)$ thus $g$ is strictly decreasing and strictly convex over its domain.

In particular Lemma D.1 above shows that the epigraph of $f$ is convex. We give further properties of $f$ in the next Lemma D.2 where we provide some results about its epigraph and on the hypersurface $\mathcal{S}$ induced by the graph of $f$.

**Lemma D.2** (Properties of the Hypersurface $\mathcal{S}$ and the Epigraph epi $f$). *Let $C$ as defined in (31) be the domain of the function $f$ defined in (32) and let $\boldsymbol{\alpha} = (\alpha_1, \ldots, \alpha_d)$ be non-negative real numbers. Assume $\alpha_k = 0$ and let $g_k \colon C_k \to (0, +\infty)$ be the function*

$$g_k \colon \breve{x}^{(k)} := (x_1, \ldots, x_{k-1}, x_{k+1}, \ldots, x_d) \mapsto \left( 1 + \sum_{j \in [1..K] \setminus \{k\}} \alpha_j x_j \right)^{-1},$$

*where $C_k := \left\{ \breve{x}^{(k)} \in \mathbb{R}^{d-1} \ : \ \sum_{j \in [1..K] \setminus \{k\}} \alpha_j x_j > -1 \right\}$. Then*

1. *epi $g_k$ is a $d$-dimensional closed convex subset of $\mathbb{R}^d$ where*

$$\mathrm{epi}\, g_k := \left\{ (x_1, \ldots, x_n) \in C \ : \ x_k \geq \left( 1 + \sum_{j=1}^{d} \alpha_j x_j \right)^{-1} \right\},$$

2. *given $x \in \mathcal{S}_k := \partial(\mathrm{epi}\, g_k)^{18}$ then, for any vector $v = (v_1, \ldots, v_d) \in \mathbb{R}^d \setminus \{(0, \ldots, 0)\}$ such that*

$$v_k = 0 \quad \text{and} \quad \sum_{j=1}^{d} \alpha_j v_j = \langle \boldsymbol{\alpha}^\top \mid v \rangle = 0, \tag{35}$$

   *the parametric line $(\ell) \colon x + tv$ belongs to $\mathcal{S}_k$. Conversely, if for some $\varepsilon > 0$ and vector $v \in \mathbb{R}^d \setminus \{(0, \ldots, 0)\}$ the segment $[x - \varepsilon v, x + \varepsilon v]$ is included in $\mathcal{S}_k$ then the whole line $(\ell) \colon x + tv$ for $t \in \mathbb{R}$ is also included in $\mathcal{S}_k$ and $v$ is of the form (35),*

3. *let $J := \{j \in [d] \ : \ j \neq k \text{ and } \alpha_j = 0\}$, then*

   - *either $J = [d] \setminus \{k\}$, that is, all the coefficients $\alpha_j$ for $j \in [d] \setminus \{k\}$ are zero, in which case $\mathcal{S}_k$ is the $(d-1)$-dimensional affine hyperplane $A$ defined as*

$$A = \left\{ (x_1, \ldots, x_d) \in \mathbb{R}^d \ : \ x_k = 1 \right\},$$

   - *otherwise, there exists at least one $j \in [d]$ with $j \neq k$ such that $\alpha_j \neq 0$, and for every $p \in \mathcal{S}_k$ there exists a unique affine subspace $A$ of $\mathbb{R}^d$ of dimension $d - 2$ which meets $p$ and is included in the hypersurface $\mathcal{S}_k$, that is, such that $p \in A$ and $A \subseteq S_k$.*

     *Moreover, if we decompose the affine subspace $A$ as $A = p + E$ where $E$ is parallel to $A$ and pass through the origin, then the canonical basis vectors $(e_j)_{j \in J}$ all belong to $E$,*

4. *for any point $p \in \mathcal{S}_k := \partial(\mathrm{epi}\, g_k)$, there exists a unique supporting hyperplane $H_k(p)$ of epi $g_k$ at $p$ and this affine hyperplane $H_k(p)$ contains the affine subspace $A$ described above (property 3).*

*Proof of Lemma D.2.* We establish these claims one after the other.

1. First, note that since $\alpha_k = 0$ then the function $g_k$ is well-defined since its value does not depend on $x_k$. Then, up to a permutation of the coordinates, we see that we can apply Lemma D.1 to $g_k$ hence, the function $g_k \colon C_k \to (0, +\infty)$ is convex. According

---

[18]It should be understand here that the hypersurface $\mathcal{S}_k$ is the set

$$\mathcal{S}_k := \left\{ (x_1, \ldots, x_d) \in C \ : \ x_k = \left( 1 + \sum_{j=1}^{d} \alpha_j x_j \right)^{-1} \right\}.$$

to Definition C.2 this means that the epigraph of $g_k$ is a convex subset of $\mathbb{R}^d$. Moreover, this epigraph is

$$
\operatorname{epi} g_k = \left\{ (x_1, \ldots, x_n) \in C \; : \; x_k \geq \left( 1 + \sum_{j=1}^d \alpha_j x_j \right)^{-1} \right\},
$$

which is a closed subset of $\mathbb{R}^d$. Effectively, if $\left( (x_i^{(\ell)})_{i \in [d]} \right)_{\ell \geq 0}$ is a sequence of points of $\operatorname{epi} g_k$ which converges (say, in $\ell_2$-norm) to the point $(x_i^{(\infty)})_{i \in [d]} \in \mathbb{R}^d$ then, for any integer $\ell \geq 0$

$$
x_k^{(\ell)} \geq \left( 1 + \sum_{j=1}^d \alpha_j x_j^{(\ell)} \right)^{-1},
$$

and taking the limits $\ell \to +\infty$ leads to

$$
x_k^{(\infty)} \geq \left( 1 + \sum_{j=1}^d \alpha_j x_j^{(\infty)} \right)^{-1},
$$

since the inverse function is continuous on $\mathbb{R}_+^*$. Hence $(x_i^{(\infty)})_{i \in [d]} \in \operatorname{epi} g_k$ so is closed.

To show that $\operatorname{epi} g_k$ is a $d$-dimensional convex subset of $\mathbb{R}^d$ is suffices to show that it contains an non-empty open-ball (say, for the $\ell_2$-norm). First, note that the function $g_k$ is continuous over $C_k$ and since $C_k$ is an open convex subset of $\mathbb{R}^d$ as proved in Lemma D.1 then (since $\mathbb{R}^d$ is a *metric space*), there exists some $r > 0$ and some point $\breve{x}^{(k)} = (x_1, \ldots, x_{k-1}, x_{k+1}, \ldots, x_d) \in C_k$ such that the (non-empty) *closed* ball $\overline{\mathrm{B}}(\breve{x}^{(k)}, r) \subseteq C_k$. Now, since we are in a finite dimensional space, the we can apply Riesz theorem (Rynne & Youngson, 2008) so that the closed ball $\overline{\mathrm{B}}(\breve{x}^{(k)}, r)$ is a compact subset of $C_k$. As the function $g_k$ is continuous then, it is upper bounded on the ball $\overline{\mathrm{B}}(\breve{x}^{(k)}, r)$ by some constant $M \geq 0$. Then, let $x_k \geq M + r$, we deduce that the open ball

$$
\mathrm{B}(x, r) \subseteq \operatorname{epi} g_k,
$$

where $x = (x_1, \ldots, x_k)$. Effectively, for any $y = (y_1, \ldots, y_d) \in \mathrm{B}(x, r)$, we have $\breve{y}^{(k)} \in \overline{\mathrm{B}}(\breve{x}^{(k)}, r)$ and

$$
y_k > x_k - r \geq M \geq \max_{z \in \overline{\mathrm{B}}(\breve{x}^{(k)}, r)} g_k(z) \geq \left( 1 + \sum_{j=1}^d \alpha_k y_j \right)^{-1},
$$

which proves the desired result.

2. We will first prove the second part of the statement (the "converse" direction) namely, that every vector $v \in \mathbb{R}^d \setminus \{(0, \ldots, 0)\}$ for which $x + tv \in \mathcal{S}_k$ for all $t \in (-\varepsilon, \varepsilon)$ where $\varepsilon > 0$ is fixed is of the form (35) and, in this case, the whole line for $t \in \mathbb{R}$ is included in the hypersurface $\mathcal{S}_k$. Hence, let $x \in \mathcal{S}_k$ and assume there exists some non-zero $v = (v_1, \ldots, v_d) \in \mathbb{R}^d$ and $\varepsilon > 0$ such that for every $t \in (-\varepsilon, \varepsilon)$ we have $x + tv \in \mathcal{S}_k$. This means

$$
x_k + tv_k = \left( 1 + \sum_{j=1}^d \alpha_j (x_j + tv_j) \right)^{-1},
$$

that is

$$(x_k + tv_k) \left( 1 + \sum_{j=1}^{d} \alpha_j (x_j + tv_j) \right)$$

$$= x_k \left( 1 + \sum_{j=1}^{d} \alpha_j x_j \right) + t \left[ v_k \left( 1 + \sum_{j=1}^{d} \alpha_k x_j \right) + x_k \sum_{j=1}^{d} \alpha_j v_j \right] + t^2 v_k \sum_{j=1}^{d} \alpha_j v_j$$

$$\overset{(a)}{=} 1 + t \left[ v_k \left( 1 + \sum_{j=1}^{d} \alpha_k x_j \right) + x_k \sum_{j=1}^{d} \alpha_j v_j \right] + t^2 v_k \sum_{j=1}^{d} \alpha_j v_j$$

$$= 1,$$

where in (a) we use the fact that $x \in \mathcal{S}_k$, in particular, $x_k > 0$. Hence, simplifying the above computation gives

$$t \left[ v_k \left( 1 + \sum_{j=1}^{d} \alpha_k x_j \right) + x_k \sum_{j=1}^{d} \alpha_j v_j + tv_k \sum_{j=1}^{d} \alpha_j v_j \right] = 0, \tag{36}$$

and since this equality holds for all $t \in (-\varepsilon, \varepsilon)$ hence, the right factor in (36) vanishes infinitely many times in $(-\varepsilon, \varepsilon) \setminus \{0\} \neq \varnothing$ hence, it must vanish everywhere thus, its coefficients must be zero, i.e.

$$v_k \left( 1 + \sum_{j=1}^{d} \alpha_k x_j \right) + x_k \sum_{j=1}^{d} \alpha_j v_j = 0,$$

and

$$v_k \sum_{j=1}^{d} \alpha_j v_j = 0.$$

Thus, either $v_0 = 0$ which implies

$$x_k \sum_{j=1}^{d} \alpha_j v_j = 0,$$

but since $x \in \mathcal{S}_k$ then $x_k > 0$ hence $\sum_{j=1}^{d} \alpha_j v_j = 0$. Otherwise, if $\sum_{j=1}^{d} \alpha_j v_j = 0$ then

$$v_k (1 + 0) + 0 = v_k = 0,$$

thus we obtain the claimed conditions

$$v_k = 0 \quad \text{and} \quad \sum_{j=1}^{d} \alpha_j v_j = 0.$$

Conversely, let $x \in \mathcal{S}_k$ and let $v = (v_1, \ldots, v_d) \in \mathbb{R}^d \setminus \{(0, \ldots, 0)\}$ such that

$$v_k = 0 \quad \text{and} \quad \sum_{j=1}^{d} \alpha_j v_j = 0,$$

then, for any $t \in \mathbb{R}$, we have

$$x_k + tv_k = x_k = \left( 1 + \sum_{j=1}^{d} \alpha_j x_j \right)^{-1} = \left( 1 + \sum_{j=1}^{d} \alpha_j (x_j + tv_j) \right),$$

hence, the whole parametric line $(\ell)\colon x + tv$ belongs to the hypersurface $\mathcal{S}_k$ and this achieves the proof of the statement.

3. Recall the definition of the set $J := \{j \in [d] : j \neq k \text{ and } \alpha_j = 0\}$, we distinguish two cases:

   - if $J = [d] \setminus \{k\}$ then $\boldsymbol{\alpha} = (0, \ldots, 0) \in \mathbb{R}^d$ thus the function $g_k \colon C_k \to (0, +\infty)$ is constant equal to one thus, the hypersurface $\mathcal{S}_k = \partial(\text{epi } g_k)$ is by definition the set
   
   $$\mathcal{S}_k := \left\{(x_1, \ldots, x_d) \in \mathbb{R}^d : x_k = 1\right\},$$
   
   which is a non-trivial hyperplane of $\mathbb{R}^d$. This proves the first claim,
   - otherwise, assume there exists some $j \neq k$ in $[d]$ such that $\alpha_j \neq 0$ and let $p \in \mathcal{S}_k$. First, any affine subspace $A$ of $\mathbb{R}^d$ which meets $p$ is of the form $A = p + \text{Vect}_{\mathbb{R}}\left((v^{(1)}, \ldots, v^{(\ell)})\right)$ for some integer $\ell \geq 1$ and (possibly zero) vectors $(v_1, \ldots, v_\ell) \in \mathbb{R}^d$. Then, note that according to the previous statement (property 2), if $A$ is included in $\mathcal{S}_k$ then the lines $(\ell_i) \colon p + tv^{(i)}$ for any $i \in [\ell]$ should be included in $\mathcal{S}_k$ hence, are of the form (35), that is
   
   $$v_k^{(i)} = 0 \text{ and } \left\langle \boldsymbol{\alpha}^\top \mid v^{(i)} \right\rangle = 0,$$
   
   hence $v^{(i)} \in \{\boldsymbol{\alpha}^\top\}^\perp$, the orthogonal subspace to the line $\text{Vect}_{\mathbb{R}}(\boldsymbol{\alpha}^\top)$. Moreover, since $\boldsymbol{\alpha} \neq (0, \ldots, 0)$ and $\alpha_k = 0$ then $\{\boldsymbol{\alpha}^\top\}^\perp$ is a subspace of $\mathbb{R}^d$ of dimension $d - 1$ containing $e_k$, the $k$-th basis vector. Hence, we deduce that
   
   $$v^{(i)} \in \{\boldsymbol{\alpha}^\top\}^\perp \cap \{e_k\}^\perp,$$
   
   which is a subspace of dimension $d-2$ of $\mathbb{R}^d$ because $\alpha_k = 0$ hence the family $(\boldsymbol{\alpha}, e_k)$ has rank 2. Hence, any affine subspace which meets $p$ and is included in $\mathcal{S}_k$ satisfies
   
   $$A \subseteq p + \{\boldsymbol{\alpha}^\top\}^\perp \cap \{e_k\}^\perp.$$
   
   Conversely, the affine subspace $p + \{\boldsymbol{\alpha}^\top\}^\perp \cap \{e_k\}^\perp$ meets $p$ and is also included in $\mathcal{S}_k$ since for any $v = (v_1, \ldots, v_d) \in \{\boldsymbol{\alpha}^\top\}^\perp \cap \{e_k\}^\perp$ we have
   
   $$\langle v \mid e_k \rangle = v_k = 0 \text{ and } \langle \boldsymbol{\alpha}^\top \mid v \rangle = 0,$$
   
   thus by property 2 above, the line $(\ell) \colon p + tv, t \in \mathbb{R}$ belongs to $\mathcal{S}_k$.
   This proves that there exists a unique maximal affine subspace $A$ which meets $p$ and which is included in $\mathcal{S}_k$. This affine subspace is $A = p + \{\boldsymbol{\alpha}^\top, e_k\}^\perp$ and has dimension $d - 2$. Additionally, for any $j \in J$, both $j \neq k$ and $\alpha_j = 0$ thus, since $\langle e_j \mid e_k \rangle = 0$ and $\langle \boldsymbol{\alpha}^\top \mid e_j \rangle = 0$ thus
   
   $$e_j \in \{\boldsymbol{\alpha}^\top\}^\perp \cap \{e_k\}^\perp,$$
   
   which shows that $e_j \in (A - p)$ hence, the basis vector $(e_j)_{j \in J}$ all belong to $(A - p)$ and the claim follows.

4. Note, by definition of $\text{epi } g_k$ we have

   $$\text{epi } g_k = \left\{(x_1, \ldots, x_k) \in \mathbb{R}^d : 0 \geq -x_k + \left(1 + \sum_{j=1}^d \alpha_j x_j\right)^{-1}\right\},$$

   and let

   $$\varphi(x_1, \ldots, x_n) := -x_k + \left(1 + \sum_{j=1}^d \alpha_j x_j\right)^{-1},$$

   then $\text{epi } g_k$ is a level set of $\varphi \colon \mathbb{R}^d \to \mathbb{R}$ and, since $\varphi$ is real-valued, continuous and differentiable over $\mathbb{R}^d$, applying Lemma C.14 gives, for any point $p = (p_1, \ldots, p_k) \in \mathcal{S}_k$ there exists a unique supporting hyperplane $H_k(p)$ of $\text{epi } g_k$ at point $p$. Moreover, we know that this supporting hyperplane is defined as

   $$H_k(p) := \left\{x \in \mathbb{R}^d : \langle \nabla\varphi(p) \mid x \rangle = \langle \nabla\varphi(p) \mid p \rangle\right\},$$

   hence, based on the previous property (and notably the set $J$), we distinguish two cases:

- if $J = [d] \setminus \{k\}$ then we proved that $\mathcal{S}_k$ is the affine hyperplane

$$\mathcal{S}_k = \left\{ (x_1, \ldots, x_d) \in \mathbb{R}^d \,:\, x_k = 1 \right\},$$

and in this case, the affine subspace $A$ and the supporting hyperplane $H_k(p)$ are the same, for any $p \in \mathcal{S}_k$, which follows from the fact that we have $\varphi \colon (x_1, \ldots, x_d) \mapsto 1 - x_k$ so $\nabla\varphi(p) = -e_k$ and hence, for any $x \in \mathbb{R}^d$, $\langle \nabla\varphi(p) \mid x \rangle = \langle \nabla\varphi(p) \mid p \rangle$ is, and only if

$$x_k = p_k = 1.$$

- now, assume $J$ contains some $j \neq k$ such that $\alpha_j \neq 0$. Recall that we proved the largest affine subspace $A$ which meets $p$ and which is included in $\mathcal{S}_k$ to be

$$A = p + \{\boldsymbol{\alpha}^\top, e_k\}^\perp,$$

and, since

$$\nabla\varphi(p) = \begin{pmatrix} \alpha_1/C(p) \\ \vdots \\ \alpha_{k-1}/C(p) \\ -1 \\ \alpha_{k+1}/C(p) \\ \vdots \\ \alpha_d/C(p) \end{pmatrix} = \frac{1}{C(p)}\boldsymbol{\alpha} - e_k,$$

where $C(p) := \left(1 + \sum_{j=1}^{d} \alpha_j p_j\right)^2$ then, for any vector $v \in \{\boldsymbol{\alpha}^\top, e_k\}^\perp$ we have both

$$\langle \boldsymbol{\alpha}^\top \mid v \rangle = 0 \quad \text{and} \quad \langle e_k \mid v \rangle,$$

which gives, by linearity of the cross-product

$$\langle \nabla\varphi(p) \mid v \rangle = \frac{1}{C(p)} \langle \boldsymbol{\alpha}^\top \mid v \rangle - \langle e_k \mid v \rangle = 0,$$

thus $v \in H_k(p)$. This shows that $A \subseteq H_k(p)$ but these affine subspaces are not equal since $\dim A = d - 2 < d - 1 = \dim H_k(p)$.

This achieves the proof of property 4.

$\square$

## D.2 The Geometry of the Feasible Region $\mathscr{F}$

Now, let us study the geometrical aspects of the feasible region $\mathscr{F}$ whose definition is recalled below for clarity.

**Definition D.3.** The feasible region $\mathscr{F}$ of problem $(\mathscr{P}_d)$ is the set

$$\mathscr{F} := \left\{ \Lambda \in [0, 1]^d : 0 \leq \Lambda + \Lambda \odot (M\Lambda) \leq 1 \ \text{ for all } k \in [d] \right\}, \tag{37}$$

where $M$ is a $d \times d$ matrix with non-negative entries.

Moreover, so as to handle the expression appearing in the above definition, we define, for any $\boldsymbol{\lambda} = (\lambda_1, \ldots, \lambda_d) \in [0, 1]^d$ and any $k \in [d]$ the quadratic function associated to the $k$-th constraint,

$$\rho_k(\boldsymbol{\lambda}) := \lambda_k \left(1 + \sum_{j=1}^{d} M_{k,j}\lambda_j\right).$$

We now start to study the geometrical aspect of the feasible region $\mathscr{F}$.

**Definition D.4** (Components of the Region $\mathscr{F}$). For any element $I = (i_1, \ldots, i_d) \in \{-1, 0, 1\}^d$, we define

$$
\mathcal{C}_I := \left\{ \boldsymbol{\lambda} = (\lambda_1, \ldots, \lambda_d) \in \mathscr{F} \; : \; \text{for all } k \in [d], \; \begin{cases} \text{if } i_k \in \{0, 1\}, & \text{then } \rho_k(\boldsymbol{\lambda}) = i_k \\ \text{if } i_k = -1, & \text{then } 0 < \rho_k(\boldsymbol{\lambda}) < 1 \end{cases} \right\},
$$

the *component* of $\mathscr{F}$ associated to the *constraints index* $I$.

**Definition D.5** (Interior Region, Extreme Points, Edges and Faces of $\mathscr{F}$). For the feasible region (37), given $I = (i_1, \ldots, i_d) \in \{-1, 0, 1\}^d$ then,

- if $I = (-1, \ldots, -1)$, we call the component $R_{\mathscr{F}} := \mathcal{C}_{(-1, \ldots, -1)}$ the *interior region* of $\mathscr{F}$,

- if $I \in \{0, 1\}^d$, the component $\mathcal{E}_I := \mathcal{C}_I$ is called an *extreme point*[19] of the domain $\mathscr{F}$,

- if there exists a unique $k \in [d]$ such that $i_k = -1$ then the component $E_I := \mathcal{C}_I$ is called an *edge* of $\mathscr{F}$. The set of all $I \in \{-1, 0, 1\}^d$ such that $\mathcal{C}_I$ is an edge of $\mathscr{F}$ is denoted by $E_{\mathscr{F}}$, that is

$$
E_{\mathscr{F}} := \left\{ (i_1, \ldots, i_d) \in \{-1, 0, 1\}^d \; : \; \text{there exists a unique } k \in [d], \text{ such that } i_k = -1 \right\}.
$$

- otherwise, if there exists $1 \leq k, \ell \leq d$ with $k \neq \ell$ such that $i_k = -1$ and $i_\ell \in \{0, 1\}$ then the component $F_I := \mathcal{C}_I$ is called a *face* of $\mathscr{F}$. The set of all $I \in \{-1, 0, 1\}^d$ such that $\mathcal{C}_I$ is a face of $\mathscr{F}$ is denoted by $F_{\mathscr{F}}$, that is

$$
F_{\mathscr{F}} := \{-1, 0, 1\}^d \setminus \left( \{-1\}^d \cup \{0, 1\}^d \cup E_{\mathscr{F}} \right).
$$

Let $k \in [d]$, recall that the constraint of the feasible region $\mathscr{F}$ associated to $\lambda_k$ as defined in (37) is given by

$$
0 \leq \lambda_k \left( 1 + \sum_{j=1}^d M_{k,j} \lambda_j \right) \leq 1,
$$

that is, $(\lambda_1, \ldots, \lambda_d)$ belongs to the quadrant $\mathbb{R}_+^d$ of non-negative real numbers, intersected with the *hypograph* of the function $g_k \colon \mathbb{R}_+^{d-1} \to \mathbb{R}$ defined as

$$
g_k \colon \breve{x}^{(k)} := (x_1, \ldots, x_{k-1}, x_{k+1}, \ldots, x_d) \mapsto \left( 1 + \sum_{j=1}^d M_{k,j} x_j \right)^{-1}, \tag{38}
$$

i.e.,

$$
(\lambda_1, \ldots, \lambda_d) \in \mathbb{R}_+^d \cap \left\{ (x_1, \ldots, x_d) \in \mathbb{R}^d \; : \; x_k \leq g_k\big(\breve{x}^{(k)}\big) \right\}.
$$

These are the same functions as introduced and studied in Lemma D.2 but specialized with the coefficients of the strictly upper triangular matrix $\mathcal{M}$. Moreover, so as to ease the statement of future results, we introduce the very similar function

$$
g_k^\varepsilon \colon \breve{x}^{(k)} := (x_1, \ldots, x_{k-1}, x_{k+1}, \ldots, x_d) \mapsto \varepsilon \left( 1 + \sum_{j=1}^d M_{k,j} x_j \right)^{-1}, \tag{39}
$$

where $\varepsilon \in \{0, 1\}$. Again, the function $g_k^\varepsilon$ is still convex and its epigraph is thus a $d$-dimensional convex subset of $\mathbb{R}^d$ according to Lemma D.2 (property 1). Additionally, if $\varepsilon = 0$ then $\partial(\operatorname{epi} g_k^\varepsilon)$ is simply the hyperplane orthogonal to basis vector $e_k$.

---

[19]It is not clear at this moment if the nomenclature of "extreme point" for these objects is meaningful. The definition of extreme point is provided in Definition 3.2 and it is shown in Lemma E.4 that indeed, the $(e_I)_{I \in \{0,1\}^d}$ are extreme points of the feasible region $\mathscr{F}$.

For clarity, we recall below the *epigraph* and *hypograph* of the function $g_k^\varepsilon$ which are defined as

$$\operatorname{epi} g_k^\varepsilon := \left\{ (x_1, \ldots, x_d) \in \mathbb{R}^d \ : \ x_k \geq g_k^\varepsilon\big(\breve{x}^{(k)}\big) \right\},$$

and

$$\operatorname{hypo} g_k^\varepsilon := \left\{ (x_1, \ldots, x_d) \in \mathbb{R}^d \ : \ x_k \leq g_k^\varepsilon\big(\breve{x}^{(k)}\big) \right\}.$$

Moreover, their *exterior* are the respective sets

$$\operatorname{ext}(\operatorname{epi} g_k^\varepsilon) := \left\{ (x_1, \ldots, x_d) \in \mathbb{R}^d \ : \ x_k < g_k^\varepsilon\big(\breve{x}^{(k)}\big) \right\},$$

and

$$\operatorname{ext}(\operatorname{hypo} g_k^\varepsilon) := \left\{ (x_1, \ldots, x_d) \in \mathbb{R}^d \ : \ x_k > g_k^\varepsilon\big(\breve{x}^{(k)}\big) \right\}.$$

Additionally, we define the closed half-space induced by the supporting hyperplane $H_k^\varepsilon(p)$ of $\operatorname{epi} g_k^\varepsilon$[20] at point $p \in \partial(\operatorname{epi} g_k^\varepsilon)$ and directed toward the feasible region $\mathscr{F}$ as

$$H_k^{\varepsilon,+}(p) := \left\{ x = (x_1, \ldots, x_d) \in \mathbb{R}^d \ : \ \begin{cases} x_k \geq 0, & \text{if } \varepsilon = 0, \\ \langle \nabla \varphi_k(p) \mid x - p \rangle \geq 0, & \text{if } \varepsilon = 1. \end{cases} \right\}, \qquad (40)$$

where $\varphi_k \colon C \to \mathbb{R}$ is defined as

$$\varphi_k \colon (x_1, \ldots, x_d) \mapsto -x_k + \left( 1 + \sum_{j=1}^d \alpha_j x_j \right)^{-1},$$

and $C$ is the convex set defined in Lemma D.1, i.e., in (31) (for the special case $\alpha_k = 0$). Notably, $(\operatorname{epi} g_k^1)$ is convex (see Lemma D.2, property 1) and $(\operatorname{hypo} g_k^0)$ is also convex since its the hypersurface $x = g_k^0(\breve{x}^{(k)})$ is an hyperplane of $\mathbb{R}^d$ so, the convexity of these two sets implies both

$$H_k^{1,+}(p) \cap \operatorname{int}(\operatorname{epi} g_k^1) = \varnothing,$$

and

$$H_k^{0,+}(p) \cap \operatorname{int}(\operatorname{hypo} g_k^0) = H_k^{0,+}(p) \cap \operatorname{ext}(\operatorname{epi} g_k^0) = \varnothing. \qquad (41)$$

**Lemma D.6** (Properties of the feasible region $\mathscr{F}$)**.** *The feasible region $\mathscr{F}$ as defined in definition 3.1*

1. *is diffeomorphic to the unit hypercube $[0, 1]^d$,*

2. *is a compact (closed and bounded subset of $\mathbb{R}^d$) and non-empty subset of $[0, 1]^d$. Moreover, it contains the zero vector $(0, \ldots, 0)^\top \in \mathscr{F}$,*

3. *has a non-empty interior,*

4. *is convex if, and only if $M_{k,j} = 0$ for all $1 \leq k, j \leq d$ iff $(1, \ldots, 1)^\top \in \mathscr{F}$.*

*Proof.* We establish these claims one after the other.

1. According to lemma 3.6, we know there exists a $\mathcal{C}^\infty$–diffeomorphism $\Psi \colon [0, 1]^d \to \mathscr{F}$ hence the feasible region $\mathscr{F}$ is diffeomorphic to the unit hypercube $[0, 1]^d$.

2. By definition of the feasible region $\mathscr{F}$, we know that $\mathscr{F} \subseteq [0, 1]^d$ so $\mathscr{F}$ is bounded. Moreover, the zero vector $(0, \ldots, 0)^\top$ is in $\mathscr{F}$ since putting $\lambda_0 = \cdots = \lambda_d = 0$ leads to

$$0 \leq 0 = \lambda_k \left( 1 + \sum_{j=k+1}^d M_{k,j} \lambda_j \right) \leq 1,$$

for all $k \in [d]$ and all constraints are satisfied so $\mathscr{F} \neq \varnothing$. Finally, $\mathscr{F}$ is also a closed subset of $\mathbb{R}^d$ because it is diffeomorphic to the unit (closed) hypercube $[0, 1]^d$ and since diffeomorphisms preserve open and closed sets then $\mathscr{F}$ is also closed thus, it is a compact subset of $\mathbb{R}^d$.

---

[20]Note that $\partial(\operatorname{epi} g_k^\varepsilon) = \partial(\operatorname{hypo} g_k^\varepsilon)$ so the boundary does not change if we take the epigraph of the hypograph of $g_k^\varepsilon$.

3. Here, as the map $\Psi \colon [0\,,1]^d \to \mathscr{F}$ is a homeomorphism (notably, $\Psi^{-1}$ is continuous), we have, where $\operatorname{int} A$ denotes the interior of a set $A$,

$$\operatorname{int} \mathscr{F} = \operatorname{int} \Psi([0\,,1]^d) = \Psi(\operatorname{int} [0\,,1]^d) = \Psi((0\,,1)^d) \neq \varnothing, \tag{42}$$

since $(0\,,1)^d \neq \varnothing$. Hence, the feasible region $\mathscr{F}$ has non-empty interior (and its interior is even diffeomorphic to the open unit hypercube $(0\,,1)^d$).

4. We first show the second equivalence, that is, $M_{k,j} = 0$ for all $1 \leq k,j \leq d$ iff $(1,\ldots,1)^\top \in \mathscr{F}$. Assume first that $M_{k,j} = 0$ for all $1 \leq k,j \leq d$ then, the inequality constraints in problem (1) reduce to

$$0 \leq \lambda_k \leq 1, \tag{43}$$

for all $k \in [d]$ thus $0 \leq \lambda_k \leq 1$ and since there is now no inter-dependency anymore between the stepsizes $\{\lambda_k\}_{k \in [d]}$ we deduce that the feasible region is simply $\mathscr{F} = [0\,,1]^d$ so it is convex and contains the vector $(1,\ldots,1)^\top$. Conversely, if $\mathscr{F}$ contains the vector $(1,\ldots,1)^\top$ then, it means this point satisfies all the constraints thus

$$0 \leq \left( 1 + \sum_{j=k+1}^{d} M_{k,j} \right) = 1 + \sum_{j=k+1}^{d} M_{k,j} \leq 1,$$

which is impossible, except in the case where $M_{k,j} = 0$ for all $j \in [k+1\,..\,d]$ that is, the upper triangular matrix $M = 0$ is the zero matrix.

Now, for the first equivalence, we already proved the converse, that is, if $M$ is the zero matrix then $\mathscr{F} = [0\,,1]^d$ so the feasible region is convex. So, let us assume $\mathscr{F}$ is convex and, for the sake of contradiction, suppose the strictly upper triangular matrix $M$ is non-zero, hence, there exists an integer $0 \leq k < j_0 \leq d$ such that $M_{k,j_0} \neq 0$. Necessarily, $k < d$ since $M$ is strictly upper triangular so without loss of generality, let us take $k \in [d-1]$ to be the largest integer such that for some $j \in [k+1\,..\,d]$ the coefficient $M_{k,j} \neq 0$. Then, for all $k' \in [k+1\,..\,d]$ we must have $M_{k',j} = 0$ for all $j \in [k'+1\,..\,d]$ so the variables $\lambda_{k+1},\ldots,\lambda_d$ all satisfy inequalities (43), i.e., we have the freedom to choose them inside $[0\,,1]$ and then we can always found values for the other variables $\lambda_1,\ldots,\lambda_k$ (notably, zero as it is always possible to choose this value) so as to ensure the point $(\lambda_0,\ldots,\lambda_d)$ is still feasible. That being said, note that the two points

$$\{0\}^k \times \left\{ \frac{1}{1+s_k} \right\} \times \{1\}^{d-k} \in \mathscr{F} \quad \text{and} \quad \{0\}^k \times \{1\} \times \{0\}^{d-k} \in \mathscr{F},$$

where $s_k := \sum_{j=k+1}^{d} M_{k,j} > 0$ since $M_{k,j_0} > 0$ by assumption. Effectively, for both points we only need to check the constraint associated to stepsize $\gamma_k$ which for the first one gives

$$0 \leq \frac{1}{1+s_k} \left( 1 + \sum_{j=k+1}^{d} M_{k,j} \right) = \frac{1}{1+s_k}(1+s_k) = 1 \leq 1,$$

while for the second one we have

$$0 \leq (1+0) = 1 \leq 1.$$

Note that the above two points are not *ill-defined* since $d - k > 0$. Now, as $\mathscr{F}$ is assumed convex then for any $t \in [0\,,1]$ we must have

$$t \left( \{0\}^k \times \left\{ \frac{1}{1+s_k} \right\} \times \{1\}^{d-k} \right) + (1-t) \left( \{0\}^k \times \{1\} \times \{0\}^{d-k} \right)$$

$$= \{0\}^k \times \left\{ \frac{t}{1+s_k} + (1-t) \right\} \times \{t\}^{d-k} \in \mathscr{F}.$$

Then, this implies that the points $\left\{ \{0\}^k \times \left\{ \frac{t}{1+s_k} + (1-t) \right\} \times \{t\}^{d-k} \right\}_{t \in [0,1]}$ all lie in the feasible region so, in particular, they satisfy the constraint associated to $\gamma_k$ that is

$$0 \le L \cdot \left[ \frac{t}{1+s_k} + (1-t) \right] \left( 1 + \sum_{j=k+1}^{d} t \, M_{k,j} \right) = \left[ \frac{t}{1+s_k} + 1 - t \right] \left( 1 + t \sum_{j=k+1}^{d} M_{k,j} \right) \le 1, \tag{44}$$

and, rewriting the left inequality in (44) using $s_k := \sum_{j=k+1}^{d} M_{k,j} > 0$ gives

$$\frac{t}{1+s_k} + \frac{t^2 s_k}{1+s_k} + 1 - t + t(1-t)s_k \le 1,$$

i.e.,

$$
\begin{aligned}
0 &\ge \frac{t}{1+s_k} + \frac{t^2 s_k}{1+s_k} - t + t(1-t)s_k \\
&= t \left( \frac{1}{1+s_k} + \frac{t s_k}{1+s_k} - 1 + (1-t)s_k \right) \\
&\overset{(a)}{=} t \left( \frac{1}{1+s_k} + t - \frac{t}{1+s_k} - 1 + (1-t)s_k \right) \\
&\overset{(b)}{=} t(1-t) \left( \frac{1}{1+s_k} - 1 + s_k \right) \\
&= t(1-t) \frac{1 - (1+s_k) + s_k(1+s_k)}{1+s_k} \\
&= t(1-t) \frac{s_k^2}{1+s_k} \\
&> 0, \tag{45}
\end{aligned}
$$

for any choice of $t \in (0,1)$ since $s_k > 0$. In the above, in (a) we split $\frac{t s_k}{1+s_k}$ as

$$\frac{t s_k}{1+s_k} = \frac{t(1+s_k-1)}{1+s_k} = t - \frac{t}{1+s_k},$$

while in (b) we factor out by $(1-t)$. But positivity in (45) violates the aforementioned constraint associated to $\lambda_k$ hence, we conclude that all entries of the upper triangular matrix $M$ are zero. This achieves the desired equivalence.

$\square$

We now give some properties satisfied by the components of the feasible set $\mathscr{F}$.

**Lemma D.7** (A partition of $\mathscr{F}$). *The components $(\mathcal{C}_I)_{I \in \{-1,0,1\}^d}$ of the feasible region $\mathscr{F}$ satisfy*

1. *they form a partition of $\mathscr{F}$, i.e., they are all non-empty and their union is $\mathscr{F}$,*

2. *for any $I \in \{0,1\}^d$, the extreme point $\mathcal{E}_I$ contains only a single feasible point,*

3. *the interior region $R_{\mathscr{F}}$ is exactly the interior of $\mathscr{F}$, that is $R_{\mathscr{F}} = \operatorname{int} \mathscr{F}$.*

4. *for any $I = (i_1, \ldots, i_d) \in \{-1, 0, 1\}^d$, we have*

$$\mathcal{C}_I \subseteq \mathbb{R}_+^d \cap \left( \bigcap_{\substack{j=1 \\ i_j \in \{0,1\}}}^{d} \partial(\operatorname{epi} g_j^{i_j}) \right) \cap \left( \bigcap_{\substack{j=1 \\ i_j = -1}}^{d} \left[ \operatorname{ext}(\operatorname{epi} g_j^1) \cap \operatorname{ext}(\operatorname{hypo} g_j^0) \right] \right).$$

5. *each component $\mathcal{C}_I$ for $I = (i_1, \ldots, i_d) \in \{-1, 0, 1\}^d$ is a bounded sub-manifold of $\mathbb{R}^d$ of dimension*

$$\dim(\mathcal{C}_I) = |\{k \in [d] \ : \ i_k = -1\}|,$$

   *e.g., if $d = 3$ then the faces of $\mathscr{F}$ are either 2-dimensional surfaces and the edges are 1-dimensional curves.*

*Proof of Lemma D.7.* We establish these claims one after the other.

1. Let $I = (i_1, \ldots, i_d) \in \{-1, 0, 1\}^d$, we define the weights vector $\mathbf{w} = (w_1, \ldots, w_d) \in [0, 1]^d$ as follows, for any $k \in [d]$

$$w_k = \begin{cases} i_k, & \text{if } i_k \in \{0, 1\}; \\ \frac{1}{2}, & \text{if } i_k = -1; \end{cases}$$

   then, according to lemma 3.5, the system of equations

$$\lambda_k \left( 1 + \sum_{j=k+1}^{d} M_{k,j} \lambda_j \right) = w_k,$$

   for all $k \in [d]$ admits a unique solution $\Lambda^{(\mathbf{w})} = \left( \lambda_1^{(\mathbf{w})}, \ldots, \lambda_d^{(\mathbf{w})} \right)$ and this solution is such that for any $k \in [d]$

$$\rho_k \left( \Lambda^{(\mathbf{w})} \right) = \begin{cases} i_k, & \text{if } w_k = i_k \in \{0, 1\}; \\ \frac{1}{2}, & \text{if } i_k = -1; \end{cases}$$

   thus $\Lambda^{(\mathbf{w})} \in \mathcal{C}_I \neq \varnothing$. More precisely, with the same $I = (i_1, \ldots, i_d)$ as above, we define for any $k \in [d]$ the set

$$S_k^{(I)} = \begin{cases} \{i_k\}, & \text{if } i_k \in \{0, 1\}; \\ (0, 1), & \text{if } i_k = -1; \end{cases}$$

   then, according to the definition D.4 of the component $\mathcal{C}_I$, we have by construction that $S^{(I)} := S_1^{(I)} \times \cdots \times S_d^{(I)} \neq \varnothing$ and

$$\mathcal{C}_I = \Psi \left( S_1^{(I)} \times \cdots \times S_d^{(I)} \right),$$

   where the map $\Psi \colon [0, 1]^d \to \mathscr{F}$ has been defined in lemma 3.6. Additionally, note that the sets $\{0\}$, $(0, 1)$ and $\{1\}$ are pairwise disjoint hence, for any two distinct $I \neq I'$ in $\{-1, 0, 1\}^d$ the elements $I$ and $I'$ differ at least by one coordinate thud

$$S^{(I)} \cap S^{(I')} = \varnothing,$$

   hence the sets $\left\{ S^{(I)} \right\}_{I \in \{-1,0,1\}^d}$ are pairwise disjoint and non-empty. Moreover, their disjoint union is

$$\bigsqcup_{I \in \{-1,0,1\}^d} S^{(I)} = \prod_{i=1}^{d} (\{0\} \cup (0, 1) \cup \{1\}) = [0, 1]^d,$$

   thus the sets $\left\{ S^{(I)} \right\}_{I \in \{-1,0,1\}^d}$ constitute a partition of the closed unit cube $[0, 1]^d$ and transferring them through the bijective map $\Psi \colon [0, 1]^d \to \mathscr{F}$ (the bijectivity being proved in lemma 3.5) leads to the fact the set sets $\{\mathcal{C}_I\}_{I \in \{-1,0,1\}^d}$ are pairwise disjoint (and even non-empty) and moreover,

$$\mathscr{F} = \Psi([0, 1]^d) = \Psi \left( \bigsqcup_{I \in \{-1,0,1\}^d} S^{(I)} \right) \overset{(a)}{=} \bigsqcup_{I \in \{-1,0,1\}^d} \Psi \left( S^{(I)} \right) = \bigsqcup_{I \in \{-1,0,1\}^d} \mathcal{C}_I,$$

   which shows that the $\{\mathcal{C}_I\}_{I \in \{-1,0,1\}^d}$ form a partition of the feasible region $\mathscr{F}$. Note that in (a) we use the fact that $\Psi$ is injective so that it preserves the disjoint union property.

2. Assume $I = (i_1, \ldots, i_d) \in \{0, 1\}^d$ then, for any $\boldsymbol{\lambda} \in \mathcal{C}_I$, since the value of each of the expressions $\{\rho_k(\boldsymbol{\lambda})\}_{k \in [d]}$ have been fixed (to either 0 or 1) then using lemma 3.5 we conclude that there exists a unique solution to the system of equations

$$\lambda_k \left( 1 + \sum_{j=k+1}^{d} M_{k,j} \lambda_j \right) = i_k,$$

for all $k \in [d]$. Hence, the set $\mathcal{C}_I$ is reduce to a single point, as claimed.

3. Using lemma 3.5 and what we have done in the first paragraph above, since the interior region is defined as $R_{\mathscr{F}} := C_{(-1, \ldots, -1)}$ then, we have

$$R_{\mathcal{F}} = \Psi((0, 1)^d),$$

and, using what we have proved from lemma D.6, more particularly from equation (42) gives

$$R_{\mathscr{F}} = \Psi((0, 1)^d) = \operatorname{int} \mathscr{F},$$

as desired.

4. Let $I \in \{-1, 0, 1\}^d$ then by definition D.4 we know that $\mathcal{C}_i \subseteq \mathbb{R}_+^d$. Now, let $k \in [d]$ then we distinguish two cases

   - if $i_k = -1$ then for any $\boldsymbol{\lambda} = (\lambda_1, \ldots, \lambda_d) \in \mathcal{C}_I$, we know that $0 < \rho_k(\boldsymbol{\lambda}) < 1$ so notably

     $$\lambda_k \left( 1 + \sum_{j=1}^{d} M_{k,j} \lambda_j \right) < 1,$$

     hence $\boldsymbol{\lambda} \in \operatorname{ext}(\operatorname{epi} g_k^1)$ and by the way

     $$0 < \lambda_k \left( 1 + \sum_{j=1}^{d} M_{k,j} \lambda_j \right),$$

     thus $\boldsymbol{\lambda} \in \operatorname{ext}(\operatorname{hypo} g_k^0)$,
   - if $i_k \in \{0, 1\}$ this means that for any $\boldsymbol{\lambda} = (\lambda_1, \ldots, \lambda_d) \in \mathcal{C}_I$ we have $\rho_k(\boldsymbol{\lambda}) \in \{0, 1\}$ hence: if $i_k = 1$ we should have $\rho_k(\boldsymbol{\lambda}) = 1$ thus $\boldsymbol{\lambda} \in \partial(\operatorname{epi} g_k^1)$, otherwise if $i_k = 0$ then we must have $\lambda_k = 0$ so $\boldsymbol{\lambda} \in \partial(\operatorname{epi} g_k^0) = \{(x_1, \ldots, x_d) \in \mathbb{R}^d : x_k = 0\}$.

   Thus, it follows that we have the inclusion

   $$\mathcal{C}_I \subseteq \mathbb{R}_+^d \cap \left( \bigcap_{\substack{j=1 \\ i_j \in \{0,1\}}}^{d} \partial(\operatorname{epi} g_j^{i_j}) \right) \cap \left( \bigcap_{\substack{j=1 \\ i_j = -1}}^{d} \left[ \operatorname{ext}(\operatorname{epi} g_j^1) \cap \operatorname{ext}(\operatorname{hypo} g_j^0) \right] \right),$$

   as desired.

5. Notice from lemma 3.6 that the component $\mathcal{C}_I$ where $I = (i_1, \ldots, i_d) \in \{-1, 0, 1\}^d$ is diffeomorphic (via $\Psi$) to the cartesian product

   $$S^{(I)} := S_1^{(I)} \times \cdots \times S_d^{(I)},$$

   where for any $k \in [d]$, we defined $S_k^{(I)} := \begin{cases} \{i_k\}, & \text{if } i_k \in \{0, 1\}, \\ (0, 1), & \text{if } i_k = -1. \end{cases}$ and, since $S^{(I)}$ is a bounded sub-manifold of $\mathbb{R}^d$ of dimension $\ell = |\{k \in [d] : i_k = -1\}|$, we deduce that $\mathcal{C}_I$ is also a bounded sub-manifold of $\mathbb{R}^d$ of dimension $\ell$ which proves the desired assertion

$\square$

**Definition D.8** (Degrees of Freedom of a Component). Given $I = (i_1, \ldots, i_d) \in \{-1, 0, 1\}^d$, the degrees of freedom of component $\mathcal{C}_I$ is denoted by

$$\deg(I) := \{j \in [d] \ : \ i_j = -1\}.$$

According to Lemma D.7, given a constraint index $I \in \{-1, 0, 1\}^d$, we have

$$|\deg(I)| = \dim(\mathcal{C}_I),$$

hence, faces are components of dimension at least 2 (with two degree of freedom), while edges are those of dimension 1 and have only one degree of freedom and extreme points have dimension 0 and degree 0.

**Lemma D.9** (Characterizing the Feasible Region $\mathscr{F}$). *We have*

$$\mathscr{F} = \mathbb{R}_+^d \setminus \bigcup_{i=1}^d \operatorname{int}\left(\operatorname{epi} g_i^1\right),$$

*where for any $i \in [d]$, $\operatorname{int}\left(\operatorname{epi} g_i^1\right)$ represents the interior of the epigraph of $g_i$.*

*Proof.* Note that for any $i \in [d]$, if we let $\mathbf{x} := (x, \ldots, x_d)$ and $\boldsymbol{\lambda} := (\lambda_1, \ldots, \lambda_d)$ then we have

$$\operatorname{int}(\operatorname{epi} g_i) = \left\{ \mathbf{x} \in \mathbb{R}^d \ : \ x_i > \left(1 + \sum_{j=k+1}^d M_{i,j} x_j\right)^{-1} \right\}.$$

Hence, by definition of $\mathscr{F}$ from (37) it follows

$$\mathscr{F} \overset{(37)}{:=} \left\{ \boldsymbol{\lambda} \in [0, 1]^d \ : \ 0 \le \lambda_i \left(1 + \sum_{j=i+1}^d M_{i,j} \lambda_j\right) \le 1 \ \text{ for all } i \in [d] \right\}$$

$$\overset{(a)}{=} \bigcap_{i=1}^d \left\{ \boldsymbol{\lambda} \in \mathbb{R}_+^d \ : \ \lambda_i \le \left(1 + \sum_{j=i+1}^d M_{i,j} \lambda_j\right)^{-1} \right\}$$

$$= \mathbb{R}_+^d \setminus \bigcup_{i=1}^d \left\{ \boldsymbol{\lambda} \in \mathbb{R}_+^d \ : \ \lambda_i > \left(1 + \sum_{j=i+1}^d M_{i,j} \lambda_j\right)^{-1} \right\}$$

$$= \mathbb{R}_+^d \setminus \bigcup_{i=1}^d \operatorname{int}(\operatorname{epi} g_i).$$

This proves the desired equality. Note that in (a) we use the non-negativity of the entries of the matrix $\mathcal{M}$ and that each of the $\lambda_1, \ldots, \lambda_d$ is also non-negative, which implies

$$0 \le \lambda_i \left(1 + \sum_{j=i+1}^d M_{i,j} \lambda_j\right),$$

for all $i \in [d]$, i.e., there is no need to force the $(\lambda_i)_{i \in [d]}$ to be less than one since the constraints already imply this inequality thanks to the non-negativity of the entries of the matrix $\mathcal{M}$ and of the $(\lambda_i)_{i \in [d]}$. $\qquad\square$

### D.3 SOME TECHNICAL LEMMAS

**Lemma D.10.** *For any $p \in \mathscr{F}$, let $w = (w_1, \ldots, w_d) = \Psi^{-1}(p) \in [0, 1]^d$ and for $i \in [d]$, let $H_i^{w_i}(p)$ be the supporting hyperplane of $\operatorname{epi} g_i^{w_i}$ at $p$, then*

$$A = \bigcap_{\substack{i=1 \\ w_i \in \{0,1\}}}^d H_i^{w_i}(p),$$

*is an affine subspace of $\mathbb{R}^d$ of dimension $\dim A \ge d - |\{i \in [d] \ : \ w_i \in \{0, 1\}\}|$.*

*Proof.* By definition of supporting hyperplane from Definition C.12, we know that

$$p \in \bigcap_{\substack{i=1 \\ w_i \in \{0,1\}}}^{d} H_i^{w_i}(p) \neq \varnothing,$$

hence, applying Lemma C.15 we have that the dimension of the intersection of all these $k = |\{i \in [d] : w_i \in \{0,1\}\}|$ affine hyperplanes $H_i^{w_i}(p)$ for $i \in [d]$ with $w_i \in \{0,1\}$ is at least $d - k = |\{i \in [d] : w_i = -1\}| = \deg(w)$ as claimed. $\qquad\square$

**Lemma D.11** (No Large Affine Subspaces Except *Flat* Ones)**.** *Let $I = (i_1, \ldots, i_d) \in \{-1, 0, 1\}^d$ and denote by $S := \{j \in [d] : i_j = -1\} \subseteq [d]$. Assume there exists some affine subspace $A$ of $\mathbb{R}^d$ of dimension $|S| = \deg(I)$ (the degrees of freedom of $\mathcal{C}_I$) such that*

$$A \subseteq \bigcap_{\substack{i=1 \\ w_i \in \{0,1\}}}^{d} \partial(\text{epi } g_i^{w_i}), \tag{46}$$

*then, $A = p + \text{Vect}_{\mathbb{R}} ((e_i)_{i \in S}))^{21}$ for any point $p \in A$.*

*Proof.* Recall from Lemma D.7 that the components of $\mathscr{F}$ are all non-empty so is the component $\mathcal{C}_I$ where $I$ is constraint index defined in the statement. We distinguish two cases:

- if $\deg(I) = 0$ then $A$ is an affine subspace of $\mathbb{R}^d$ of dimension 0 so is just a single point $p \in \mathbb{R}^d$ and as $S = \varnothing$ then $A = \{p\}$ and the claims follows,

- now assume $\deg(I) > 0$ then the intersection in (46) is non-empty. Let $v = (v_1, \ldots, v_d) \in (A - p)$ be a non-zero vector, where $p \in A$ then using Lemma D.2 (property 2) since the line $\text{Vect}_{\mathbb{R}} (v)$ is included in $A$ so in every $\partial(\text{epi } g_i^{w_i})$ for $i \in [d]$ with $w_i \in \{0,1\}$ then we must have
$$v_i = 0 \quad \text{and} \quad \langle M_{i,\cdot}{}^\top \mid v \rangle = 0,$$
for all $i \in [d] \setminus S$ such that $w_i = 1$. Otherwise, those $i \in [d] \setminus S$ for which $w_i = 0$, since $\partial(\text{epi } g_i^{w_i})$ is the hyperplane $\{(x_1, \ldots, x_d) \in \mathbb{R}^d : x_i = 0\} = \{e_i\}^\perp$ and the line $\text{Vect}_{\mathbb{R}} (v)$ belongs to this hyperplane then $\langle v \mid e_i \rangle = 0$, i.e., $v_i = 0$ too. Hence,
$$v \in \left\{ (e_i)_{i \in [d] \setminus S} \right\}^\perp = \underset{\mathbb{R}}{\text{Vect}} ((e_i)_{i \in S}),$$
thus $v \in \text{Vect}_{\mathbb{R}} ((e_i)_{i \in S})$ so $(A - p) \subseteq \text{Vect}_{\mathbb{R}} ((e_i)_{i \in S})$ and because $\dim A = |S| = \dim (\text{Vect}_{\mathbb{R}} ((e_i)_{i \in S}))$ then we must have equality in the previous inclusion that is
$$A = p + \underset{\mathbb{R}}{\text{Vect}} ((e_i)_{i \in S}),$$
and the assertion follows.

$\qquad\square$

**Lemma D.12** (A Technical Lemma)**.** *For the feasible region of problem $(\mathscr{P}_d)$, for any $w = (w_1, \ldots, w_d) \in [0, 1]^d \setminus \{0, 1\}^d$ ($w$ is not a vertex of the unit hypercube) let $x = \Psi(w) \in \mathscr{F}$, there exists $\rho > 0$ such that for any $y \in \text{B}(x, \rho)$, if*

$$y \in \bigcap_{\substack{i=1 \\ w_i \in \{0,1\}}}^{d} H_i^{w_i,+}(x),$$

*then $y \in \mathscr{F}$. Moreover, we can choose the radius $\rho > 0$ so that if $y \in \mathcal{C}_I$ for some $I = (i_1, \ldots, i_d) \in \{-1, 0, 1\}^d$ we have for all $j \in [d]$, if $0 < w_j < 1$ then $i_j = -1$.*

---

[21]Here, $(e_1, \ldots, e_d)$ denotes the canonical basis of $\mathbb{R}^d$ with $e_i = (0, \ldots, 0, 1, 0, \ldots, 0)^\top$ containing a 1 in its $i$-th coordinate and 0 elsewhere.

*Proof.* Assume for the sake of contradiction that the property does not hold then, there must exists some $w = (w_1, \ldots, w_d) \in [0,1]^d$ and $x = \Psi(w) \in \mathscr{F}$ such that for all radius $\rho > 0$, there exists some $y_\rho \in \mathrm{B}(x, \rho)$ such that

$$
y_\rho \in \bigcap_{\substack{i=1 \\ w_i \in \{0,1\}}}^{d} H_i^{w_i,+}(x) \ \text{ but } \ y_\rho \notin \mathscr{F}.
$$

First, let us show that for $\rho > 0$ small enough, we have $y_\rho \in \mathbb{R}_+^d$. Let $i \in [d]$, we distinguish three cases based on the value of $x_i$:

- if $x_i = 0$ and since $1 + \sum_{j=1}^{d} M_{i,j} x_j > 0$ then we must have $w_i = 0$[22] and the corresponding closed half-space is $H_i^{w_i,+}(x) = \left\{ (z_1, \ldots, z_d) \in \mathbb{R}^d : z_i \geq 0 \right\}$ so as $y_\rho \in H_i^{w_i,+}(x)$ then $[y_\rho]_i \geq 0$ and taking $\rho < 1$ is enough to ensure $[y_\rho]_i \leq 1$.

- otherwise, if $x_i >$ then take $m := \min_{\substack{i \in [d] \\ x_i > 0}} x_i$ then, it is enough to choose the radius $0 < \rho < \frac{m}{2}$ so as to ensure that the $y_\rho \in \mathrm{B}(x, \rho)$ will be such that $[y_\rho]_i > x_i - \rho > 0$ for all $i \in [d]$ with $x_i > 0$.

Hence, for all $\rho$ small enough we have $y_\rho \in \mathbb{R}^d+$.

Then using Lemma D.9 since

$$
\mathscr{F} = \mathbb{R}_+^d \setminus \bigcup_{i=1}^{d} \mathrm{int} \left( \mathrm{epi}\, g_i^1 \right),
$$

and $y_\rho \notin \mathscr{F}$, but $y_\rho \in \mathbb{R}_+^d$ by the above paragraph, then we must have $y_\rho \in \bigcup_{i=1}^{d} \mathrm{int} \left( \mathrm{epi}\, g_i^1 \right)$.

Now, since $y_\rho$ belongs to the intersection of the closed half-spaces $\bigcap_{\substack{i=1 \\ w_i \in \{0,1\}}}^{d} H_i^{w_i,+}(x)$ (the half-spaces *containing* $\mathscr{F}$, not the convex epigraph) and since by (41) we have $H_i^{w_i,+}(p) \cap \mathrm{int} \left( \mathrm{epi}\, g_i^{w_i} \right) = \varnothing$ for all $i \in [d]$ such that $w_i = 1$ so $y_\rho \notin \mathrm{int} \left( \mathrm{epi}\, g_i^1 \right)$ for all $i \in [d]$ such that $w_i = 1$. Moreover, for all $i \in [d]$ such that $w_i = 0$ we know by Lemma D.7 (property 4)

$$
x \in \partial(\mathrm{epi}\, g_i^0) = \left\{ (z_1, \ldots, z_d) \in \mathbb{R}^d : z_i = 0 \right\},
$$

so $x_i = 0$. Additionally, as the epigraph of $g_i^1$ is

$$
\mathrm{epi}\, g_i^1 = \left\{ (z_1, \ldots, z_d) \in \mathbb{R}^d : z_i \geq \left( 1 + \sum_{j=1}^{d} M_{i,j} z_j \right)^{-1} \right\},
$$

and the function $z \mapsto \left( 1 + \sum_{j=1}^{d} M_{i,j} z_j \right)^{-1}$ being continuous and positive all over the compact set $[0,2]^d$ then it must reach its global minimum somewhere on the unit hypercube, hence, there exists some $m_i > 0$ such that for all $(z_1, \ldots, z_d) \in [0,2]^d$ we have

$$
\left( 1 + \sum_{j=1}^{d} M_{i,j} z_j \right)^{-1} \geq m_i > 0.
$$

---

[22]Because by definition of $w$ and $x = \Psi(w)$, we have

$$
w_i = x_i \left( 1 + \sum_{j=1}^{d} M_{i,j} x_j \right),
$$

so if $x_i = 0$ then immediately we obtain $w_i = 0$.

By consequence, for all radius $\rho > 0$ small enough (say for instance $\rho \le \min_{\substack{i \in [d] \\ w_i = 0}} \frac{m_i}{2}$ and $\rho < 1$) then the open ball $\mathrm{B}(x, \rho)$ intersected with non-negative quadrant $\mathbb{R}_+^d$ is disjoint with the epigraph $\mathrm{epi}\, g_i^1$ for all $i \in [d]$ such that $w_i = 0$, because since $x \in [0, 1]^d$ then $\mathrm{B}(x, \rho) \cap \mathbb{R}_+^d \subseteq [0, 2]^d$ (as we take $\rho < 1$) thus,

$$\underbrace{\left(\mathbb{R}_+^d \cap \mathrm{B}(x, \rho)\right)}_{\ne \varnothing} \cap \left(\bigcup_{\substack{i \in [d] \\ w_i = 0}}^d \left(\mathrm{epi}\, g_i^1\right)\right) = \varnothing,$$

hence, for all radius $\rho > 0$ such that $\rho \le \min_{\substack{i \in [d] \\ w_i = 0}} \frac{m_i}{2}$ and $\rho < 1$, as $y_\rho \in \mathrm{B}(x, \rho) \cap \mathbb{R}_+^d$ then

$$y_\rho \notin \bigcup_{\substack{i \in [d] \\ w_i = 0}}^d \left(\mathrm{epi}\, g_i^1\right) \text{ thus } y_\rho \notin \bigcup_{\substack{i \in [d] \\ w_i = 0}}^d \mathrm{int}\left(\mathrm{epi}\, g_i^1\right).$$

From the above two paragraphs, we deduced that $y_\rho \notin \bigcup_{\substack{i=1 \\ w_i \in \{0,1\}}}^d \mathrm{int}\left(\mathrm{epi}\, g_i^1\right)$ so we must have

$$y_\rho \in \bigcup_{\substack{i=1 \\ 0 < w_i < 1}}^d \mathrm{int}\left(\mathrm{epi}\, g_i^1\right),$$

for all small enough radius $0 < \rho < \rho_0$.

Next, as asserted in the statement, the set $S = \{i \in [d] : 0 < w_i < 1\}$ is non-empty then, since the set $(0, \rho_0)$ has infinite cardinality but $1 \le |S| < +\infty$ we deduce that there must exists a $i_0 \in S$ and some sequences $(\rho_k)_{k \ge 1}$ such that for all $k \ge 1$, we have

$$0 < \rho_k < \rho_0 \text{ and } \rho_k \xrightarrow[k \to +\infty]{} 0 \text{ and } y_{\rho_k} \in \mathrm{int}\left(\mathrm{epi}\, g_{i_0}^1\right).$$

Since the sequence of radius $(\rho_k)_{k \ge 1}$ converges to $0$ then $y_{\rho_k} \xrightarrow[k \to +\infty]{} x$ thus

$$x \in \left(\mathrm{epi}\, g_{i_0}\right) \cap \mathscr{F},$$

hence by Lemma D.9 we obtain $x \in \partial(\mathrm{epi}\, g_{i_0})$ but this is a contradiction since $w_{i_0} \in (0, 1)$, i.e.,

$$x_{i_0} < \left(1 + \sum_{j=1} M_{i_0, j} x_j\right)^{-1}.$$

Finally, this proves that there must exists some radius $\rho > 0$ such that for any $y \in \mathrm{B}(x, \rho)$, if

$$y \in \bigcap_{\substack{i=1 \\ w_i \in \{0,1\}}}^d H_i^{w_i, +}(x),$$

then $y \in \mathscr{F}$. Moreover, using the set $S$ defined earlier, let $\varepsilon := \min_{i \in S} \min\{w_i, 1 - w_i\} > 0$. The quantity $\varepsilon$ is positive by definition and using the diffeomorphism $\Psi$ then $\Psi([0, 1]^d \cap \mathrm{B}(w, \frac{\varepsilon}{2}))$ is an open subset of $\mathscr{F}$ so there exists some radius $r > 0$, and without loss of generality we may take $r < \rho$, such that

$$\mathrm{B}(x, r) \cap \mathscr{F} \subseteq \Psi\left([0, 1]^d \cap \mathrm{B}\left(w, \frac{\varepsilon}{2}\right)\right),$$

so for any $y \in \mathrm{B}(x, r) \cap \mathscr{F}$, then $w' = (w_1', \ldots, w_d') = \Psi^{-1}(y) \in \mathrm{B}(w, \frac{\varepsilon}{2})$ thus for any $i \in S$,

$$0 < w_i - \frac{\varepsilon}{2} \le w_i' \le w_i + \frac{\varepsilon}{2} < 1.$$

hence the point $y \in \mathscr{F}$ keeps at least the same degrees of freedom than the point $x$ had.

This completes the proof of the lemma. $\qquad\square$

# E OMITTED PROOFS

## E.1 PROOF OF THEOREM 3.4

For completeness, we recall below the problem $(\mathscr{P}_{\mathrm{cpt}}^{\mathrm{lin}})$ as defined in the main paper in (5):

$$(\mathscr{P}_{\mathrm{cpt}}^{\mathrm{lin}}): \quad \text{maximize} \ \langle \mathbf{x} \mid \mathbf{c} \rangle \tag{47}$$
$$\text{over} \quad \mathbf{x} \in K.$$

**Theorem 3.4** (Maximization of a Linear Form over a Non-empty Compact Sets). *There exists an optimal solution of problem $(\mathscr{P}_{\mathrm{cpt}}^{\mathrm{lin}})$ in (47) which is also an extreme point of $K$, i.e.,*

$$\mathrm{Extr}\, K \cap X^* \neq \varnothing.$$

*Proof.* Let $K \subseteq \mathbb{R}^d$ be a non-empty and compact set and $f \colon \mathbb{R}^d \to \mathbb{R}$ be a linear form. Note that when $d = 0$, the space $\mathbb{R}^0$ is reduced to the single point $\{0\}$ and since $K \neq \varnothing$ then $K = \{0\}$ which is an extreme point according to Definition 3.2 (the set $K$ does not contain non-trivial segment). Thus we deduce that $\arg\max_{\mathbf{x} \in K} f(\mathbf{x}) = K = \{0\}$ for any linear form $f$ and the main claim follows.

Now, assume $d \geq 1$ then, either $f$ is constant, i.e., $f$ is always zero then $X^* = \mathbb{R}^d$ and since $\mathrm{Extr}\, K \neq \varnothing$ according to Lemma C.11 we obtain that $\mathrm{Extr}\, K \cap X^* = \mathrm{Extr}\, K \neq \varnothing$. Otherwise, when $f$ is a non-zero linear form, as we are in a finite dimensional space the linear form $f$ is continuous over the compact $K$ so we know that $f$ is bounded and that it reaches its global maximum $M \in \mathbb{R}$ somewhere over $K$. Moreover, since $f$ is non-constant then $H_{d-1} := f^{-1}(M)$ is a hyperplane of $\mathbb{R}^d$ and the set $K' := f^{-1}(M) \cap K$ is a compact subset of $H_{d-1}$ which is $(d-1)$-dimensional subspace of $\mathbb{R}^k$. Hence, up to a (linear) change of coordinates to transform linearly $H_{d-1}$ into $\mathbb{R}^{d-1}$ (and this preserves the alignments), we can apply Lemma C.11 to the compact subset $K'$ of $H_{d-1}$ and this show that $\mathrm{Extr}\, K' \neq \varnothing$. So let $p \in \mathrm{Extr}\, K' \subseteq K$ be such an extreme point, we now show that $p$ is also an extreme point of $K$. For the sake of contradiction, assume $p \notin \mathrm{Extr}\, K$ so there exists $x, y \in K$ such that $p \in (x, y)$ hence $x \neq y$ and there exists some scalar $t \in (0, 1)$ such that $p = tx + (1-t)y$. Moreover, since $p \in f^{-1}(M)$ this means that $f(p) = M$ so $f$ attains its global maximum on $K$ at least at $p$ from where $f(p) \geq f(x)$ and $f(p) \geq f(y)$, and since $f$ is linear

$$f(p) = tf(x) + (1-t)f(y) \overset{(a)}{\leq} \max\{f(x), f(y)\} \overset{(b)}{\leq} f(p), \tag{48}$$

where (a) follows from both non-negativity of $t$ and inequalities $f(p) \geq f(x)$ and $f(p) \geq f(y)$. Looking at the sequence of inequalities (48) we must have equality everywhere, notably in (a), that is to say, we must have $M = f(p) = f(x) = f(y)$ since otherwise as $t \in (0, 1)$, if $f(x) \neq f(y)$ or $\max\{f(x), f(y)\} < f(p)$ we cannot have equality in (a) for the former and in (b) for the later. This shows that $x, y \in f^{-1}(M) \cap K = K'$ thus we would have $p \in (x, y)$ in $K'$ too which means that $p$ would not be an extreme point of $K'$, but this is a contradiction. Hence $p$ must also be an extreme point of $K$ thus

$$\mathrm{Extr}\, K \cap X^* = \mathrm{Extr}\, K \cap \left( f^{-1}(M) \cap K \right) \neq \varnothing,$$

and we are done. $\square$

*Remark* E.1. We note that Theorem 3.2 (sufficiency) is a classical result: it can also be proved using the compactness of the feasible set together with standard results, e.g., Barvinok (2002).

## E.2 OMITTED PROOFS IN SECTION 3.4

**Lemma 3.5** (A Linear-Quadratic System). *Let $d \in \mathbb{N}$ be a positive integer, $M \in \mathbb{R}^{d \times d}$ a matrix with non-negative entries and $W = (w_1, \ldots, w_d)^\top \in \mathbb{R}^d$ a d-dimensional column vector with non-negative entries. Then, the system*

$$\Lambda + \Lambda \odot (M\Lambda) = W, \tag{49}$$

*has a unique solution $\Lambda = (\lambda_1, \ldots, \lambda_d)^\top \in \mathbb{R}^d$ with non-negative entries and for any $i \in [d]$ we have $\lambda_i = 0$ if, and only if $w_i = 0$.*

*Proof.* First, we prove the existence of a solution for the system (49). Notice that $\Lambda \in \mathbb{R}_+^d$ is solution to our linear quadratic system (49) if and only if

$$\forall i \in [d], \lambda_i = \frac{w_i}{1 + (M\Lambda)_i}, \tag{50}$$

which can be written as follows:

$$G_W(\Lambda) = \Lambda, \tag{51}$$

i.e., $\Lambda$ is a fixed point of $G_W$, where $G_W : \mathbb{R}_+^d \to \mathbb{R}_+^d$ is defined by $G_W(\Lambda)_i := \frac{w_i}{1+(M\Lambda)_i}$. Since we search for solutions $\Lambda \in \mathbb{R}_+^d$ (i.e., with non-negative entries), and $M$ has non-negative entries, we have from Equation (50) that if $\Lambda$ is a solution of the system, then necessarily $\lambda_i \leq w_i$ for all $i \in [d]$. Hence if $\Lambda \in \mathbb{R}_+^d$ is solution of (49), then $\Lambda \in K := [0, w_1] \times \cdots \times [0, w_d]$. Besides, $G_W$ has only values in this set $K = [0, w_1] \times \cdots \times [0, w_d]$. Since $G_W : K \to K$ is continuous and $K = [0, w_1] \times \cdots \times [0, w_d]$ is a non-empty compact convex subset of $\mathbb{R}^d$, then the *Brouwer's fixed point theorem* [23] gives the existence of a fixed point of $G_W$, and hence the existence of a solution to (49).

Now, to prove the uniqueness of the solution on $\mathbb{R}_+^d$, we will prove that the function $h \colon \mathbb{R}_+^d \to \mathbb{R}^d$ by:

$$h \colon x \mapsto (h_i(x) := x_i \left(1 + (Mx)_i\right))_{i \in [d]}, \tag{52}$$

is injective on $\mathbb{R}_+^d$. This implies the uniqueness of the solution, since $\Lambda \in \mathbb{R}_+^d$ is solution of (49) if and only if $h(\Lambda) = W$.

$h$ is a differentiable map from the *closed rectangular region*[24] $\mathbb{R}_+^d$ to $\mathbb{R}^d$, and its Jacobian is given by:

$$\nabla h(x)_{i,j} = \frac{\partial h_i}{\partial x_j}(x) = \begin{cases} M_{i,j}x_i, & \text{if } j \neq i \\ 1 + 2M_{i,i}x_i + \sum_{k \neq i} M_{i,k}x_k, & \text{if } j = i \end{cases}, \tag{53}$$

for all $i, j \in [d]$ and $x \in \mathbb{R}_+^d$.

Let $x \in \mathbb{R}_+^d$. We have for all $i \in [d]$:

$$\nabla h(x)_{i,i} = 1 + 2M_{i,i}x_i + \sum_{k \neq i} M_{i,k}x_k > 0, \tag{54}$$

so $\nabla h(x)$ has positive diagonal entries. We will use Lemma C.20 to prove that $\nabla h(x)$ is a $P$-matrix. To do so, we need to construct positive numbers $a_1, \ldots, a_d > 0$ such that for all $i \in [d]$:

$$a_i \left|\nabla h(x)_{i,i}\right| > \sum_{\substack{j=1 \\ j \neq i}}^{d} a_j \left|\nabla h(x)_{i,j}\right|, \tag{55}$$

which is equivalent, since $x$ and $M$ have non-negative coefficients, to:

$$a_i \left(1 + 2M_{i,i}x_i + \sum_{j \neq i} M_{i,j}x_j\right) > \sum_{j \neq i} M_{i,j}a_jx_i,$$

that is

$$g_x^i(a) := a_i + 2M_{i,i}x_ia_i + \sum_{j \neq i} M_{i,j}\left(a_ix_j - a_jx_i\right) > 0, \tag{56}$$

where $a$ denotes the vector $(a_1, \ldots, a_d)^\top$.

We prove that there exists $\varepsilon > 0$ such that the choice $a_i^\varepsilon := x_i + \varepsilon > 0$ satisfies the condition (56). With this choice, we have:

$$g_x^i(a^\varepsilon) = (x_i + \varepsilon)(1 + 2M_{i,i}x_i) + \sum_{j \neq i} M_{i,j}\left[(x_i + \varepsilon)x_j - (x_j + \varepsilon)x_i\right]$$

$$= (x_i + \varepsilon)(1 + 2M_{i,i}x_i) + \varepsilon\left(\sum_{j \neq i} M_{i,j}(x_j - x_i)\right). \tag{57}$$

---

[23]See Ben-El-Mechaieh & Mechaiekh (2022) for an elementary proof.

[24]A definition can be found in Appendix C.5.

If $x_i = 0$, we have

$$g_x^i(a^\varepsilon) = \varepsilon \left( 1 + 2M_{i,i}x_i + \sum_{j \neq i} M_{i,j}x_j \right) > 0,$$

for any $\varepsilon > 0$. Otherwise, $x_i > 0$ and we have

$$g_x^i(a^\varepsilon) = (x_i + \varepsilon)(1 + 2M_{i,i}x_i) + \varepsilon \left( \sum_{j \neq i} M_{i,j}(x_j - x_i) \right) \xrightarrow[\varepsilon \to 0]{} x_i(1 + 2M_{i,i}x_i) > 0, \quad (58)$$

since this limit is positive, there exists some $\varepsilon_i > 0$ such that for any $0 < \varepsilon < \varepsilon_i$, $g_x^i(a^\varepsilon) > 0$. Define $\varepsilon_0 := \min\{1, \min\{\varepsilon_i : i \text{ such that } x_i > 0\}\}$. Hence, the vector $a^{\varepsilon_0}$ satisfies $g_x^i(a^{\varepsilon_0}) > 0$ for all $i \in [d]$. In other words, $\nabla h(x)$ is *positive dominant diagonal* and hence it is a $P$-matrix by Lemma C.20. Since this holds for every $x \in \mathbb{R}_+^d$, Theorem C.21 gives that $h$ is an injective map, which implies the uniqueness of the solution to (49). This concludes our proof. $\square$

**On Some Counter-examples when $M$ has Negative Entries:** in the following two remarks, we provide counter-examples to the existence and uniqueness of solutions to (49) when the matrix $M$ has negative entries.

*Remark* E.2. The assumption on the non-negativity of the entries of the matrix $M$ in Lemma 3.5 cannot be relaxed, i.e., we cannot simply assume $M$ to be matrix in $\mathbb{R}^{d \times d}$. A simple counter-example can be constructed even when $d = 2$. For instance, consider the matrix $M$ and the vector $\mathbf{w}$ given by

$$M = \begin{pmatrix} 0 & -1 \\ 0 & 0 \end{pmatrix}, \quad \mathbf{w} = \begin{pmatrix} 1 \\ 1 \end{pmatrix}, \quad (59)$$

in which case the system $\Lambda + \Lambda \odot (M\Lambda) = \mathbf{w}$ can be written as

$$\begin{pmatrix} \lambda_1 \\ \lambda_2 \end{pmatrix} + \begin{pmatrix} \lambda_1 \\ \lambda_2 \end{pmatrix} \odot \begin{pmatrix} -\lambda_2 \\ 0 \end{pmatrix} = \begin{pmatrix} 1 \\ 1 \end{pmatrix}, \quad (60)$$

which is equivalent to

$$\begin{cases} \lambda_1 - \lambda_1\lambda_2 &= 1 \\ \lambda_2 &= 1 \end{cases}, \quad (61)$$

but the system (61) clearly does not admit any solution since the first equation reduces to $0 = 1$, which is absurd.

*Remark* E.3. We can also construct another counter-example to the uniqueness of the solutions to the system in $\mathbb{R}_+^d$ when we authorize the matrix $M$ to have negative entries, even with $d = 2$. For that, consider the matrix $M$ and the vector $\mathbf{w}$ given by

$$M = \begin{pmatrix} -1 & 1 \\ 0 & 0 \end{pmatrix}, \quad \mathbf{w} = \begin{pmatrix} 2 \\ 2 \end{pmatrix}, \quad (62)$$

in which case, the system is equivalent to:

$$\begin{cases} \lambda_1(1 - \lambda_1 + \lambda_2) &= 2 \\ \lambda_2 &= 1 \end{cases}, \quad (63)$$

and the first equation becomes $\lambda_1^2 - 3\lambda_1 + 2 = 0$ which has two solutions, namely 1 and 2, hence the system two solutions in $\mathbb{R}_+^d$:

$$\Lambda_1^* = \begin{pmatrix} 1 \\ 2 \end{pmatrix} \text{ and } \Lambda_2^* = \begin{pmatrix} 2 \\ 2 \end{pmatrix}. \quad (64)$$

**Lemma 3.6** (Regularity of the Solution of (6))**.** *Let $d \in \mathbb{N}$ be a positive integer and $M \in \mathbb{R}^{d \times d}$ a matrix with non-negative entries. For any $d$-dimensional column vector $\mathbf{w} = (w_1, \ldots, w_d)^\top \in \mathbb{R}^d$ with non-negative entries, let $\Lambda^{(\mathbf{w})} = (\lambda_1^{(\mathbf{w})}, \ldots, \lambda_d^{(\mathbf{w})})^\top$ be the unique solution of the equation*

$$\Lambda + \Lambda \odot (M\Lambda) = \mathbf{w}, \quad (65)$$

*then, the map* $\Psi \colon [0\,,1]^d \to \mathscr{F}$ *defined for* $\mathbf{w} \in [0\,,1]^d$ *by*

$$\Psi(\mathbf{w}) := \Lambda^{(\mathbf{w})} = \left(\lambda_1^{(\mathbf{w})}, \ldots, \lambda_d^{(\mathbf{w})}\right)^\top,$$

*where*

$$\mathscr{F} := \left\{\Lambda \in [0\,,1]^d \,:\, 0 \le \Lambda + \Lambda \odot (M\Lambda) \le 1\right\}, \tag{66}$$

*is a* $\mathcal{C}^\infty$*–diffeomorphism.*

*Proof of Lemma 3.6.* Note that the set $\mathscr{F}$ corresponds to all *feasible* points, that is, all points $\Lambda = (\lambda_1, \ldots, \lambda_d) \in [0\,,1]^d$ such that the inequalities

$$0 \le \lambda_k \left(1 + \sum_{j=k+1}^d M_{k,j}\lambda_j\right) \le 1, \tag{67}$$

hold for any $k \in [d]$. Hence, by Lemma 3.5 uniqueness of the solution $\Lambda^{(\mathbf{w})}$ for provided weights $\mathbf{w} \in [0\,,1]^d$ implies that the map $\Psi \colon [0\,,1]^d \to \mathscr{F}$ is bijective.

We consider the function $F : \mathbb{R}^d \times \mathbb{R}^d \to \mathbb{R}^d$ defined by:

$$F : (\mathbf{w}, \Lambda) \mapsto \Lambda + \Lambda \odot (M\Lambda) - \mathbf{w}. \tag{68}$$

$F$ is clearly $\mathcal{C}^\infty$ on $\mathbb{R}^d$ (all its components are polynomial in the entries). Now, consider an arbitrary $\mathbf{w}_0 \in [0\,,1]^d$, and let $\Lambda^{(\mathbf{w}_0)}$ be the unique solution to the system for that $\mathbf{w}_0$. We consider the point $(\mathbf{w}_0, \Lambda^{(\mathbf{w}_0)}) \in \mathbb{R}^d \times \mathbb{R}^d$. We have:

$$\nabla_{\boldsymbol{\Lambda}} F(\mathbf{w}_0, \Lambda^{(\mathbf{w}_0)}) = \nabla h(\Lambda^{(\mathbf{w}_0)}), \tag{69}$$

where $h$ is defined as in the proof of Lemma 3.5. We already proved that for every $x \in \mathbb{R}_+^d$, $\nabla h(x)$ is a $P$-matrix and hence it is invertible. Using the *implicit function theorem* (Theorem 11.4 in Loomis & Sternberg (2014)), there exists an open set $U \subset \mathbb{R}^d$ containing $\mathbf{w}_0$ such that there exists a unique function $g : U \to \mathbb{R}^d$ in $\mathcal{C}^\infty(\mathbb{R}^d, \mathbb{R}^d)$ such that $g(\mathbf{w}_0) = \Lambda^{(\mathbf{w}_0)}$ and $F(\mathbf{w}, g(\mathbf{w})) = 0$, for all $\mathbf{w} \in \mathbb{R}^d$. Note that $F(\mathbf{w}, g(\mathbf{w})) = 0$ if and only if

$$g(\mathbf{w}) + g(\mathbf{w}) \odot (Mg(\mathbf{w})) = \mathbf{w}, \tag{70}$$

that is, if and only if $g(\mathbf{w})$ is a solution of the system (65). By the uniqueness of the solution to the system for $\mathbf{w} \in [0\,,1]^d$, we have $\Psi = g$ on $U \cap [0\,,1]^d$. Since $g$ is $\mathcal{C}^\infty$ on $U \cap [0\,,1]^d$, then $\Psi$ is $\mathcal{C}^\infty$ on this intersection, and given that $\mathbf{w}_0 \in U \cap [0\,,1]^d$, we conclude that $\Psi$ is $\mathcal{C}^\infty$ in $\mathbf{w}_0$.

It only remains to prove that $\Psi^{-1}$ is $\mathcal{C}^\infty$ on $\mathscr{F}$, but given that $\Psi^{-1} = h$ (which was defined previously) and $h$ has all its components polynomial in the entries, it follows that it (and thus $\Psi^{-1}$) is $\mathcal{C}^\infty$ on $\mathscr{F}$. This concludes the proof. □

### E.3 Omitted Proofs of Section 4.1

**Theorem 4.1** (Extreme points of $\mathscr{F}$ in the Relaxed Sense)**.** *For the feasible region $\mathscr{F}$ of problem $(\mathscr{P}_d)$, we have*

$$\mathrm{Extr}_{\mathcal{R}}\, \mathscr{F} = \left\{\Psi(w) \,:\, w \in \{0,1\}^d\right\}, \tag{71}$$

*that is, the extreme points of $\mathscr{F}$ (in the relaxed sense) are exactly the vertices of the hypercube $[0\,,1]^d$ mapped by the diffeomorphism $\Psi$.*

*Proof of Theorem 4.1.* In order to prove the above Theorem 4.2, we first prove the next two lemmas.

**Lemma E.4.** *Given the feasible region $\mathscr{F}$, we have the inclusion*

$$\left\{\Psi(w) \,:\, w \in \{0,1\}^d\right\} \subseteq \mathrm{Extr}_{\mathcal{R}}\, \mathscr{F}. \tag{72}$$

*Proof of Lemma E.4.* Let $w = (w_1, \ldots, w_d) \in \{0, 1\}^d$ be a vertex of the hypercube $[0, 1]^d$, and assume, to reach a contradiction, that $\Psi(w) \in \mathscr{F}$ is not an extreme point (in the relaxed sense), i.e., $\Psi(w) \notin \text{Extr}_{\mathcal{R}} \mathscr{F}$. Then there exists $x$ and $y$ in $\mathscr{F}$ such that $x \neq y$, $[x, y] \subset \mathscr{F}$ and $p := \Psi(w) \in (x, y)$, i.e., there exists $0 < \theta < 1$ such that $p = \theta x + (1 - \theta)y$. Setting $d := x - y \neq 0$, we have $p \pm td \in \mathscr{F}$ for every $t \in [0, \varepsilon_0]$, for some $\varepsilon_0 > 0$.

We define the following sets of indices:

$$S := \{i \in [d] \ : \ w_i = 1\}, \tag{73}$$

$$Z := \{i \in [d] \ : \ w_i = 0\} = [d] \setminus S. \tag{74}$$

By definition of $\Psi$, we have for all $i \in [d]$:

$$p_i (1 + (Mp)_i) = w_i, \tag{75}$$

hence

$$\begin{cases} p_i = 0 & \forall i \in Z \\ p_i (1 + (Mp)_i)) = 1 & \forall i \in S \end{cases}. \tag{76}$$

Let $h_i(x) = x_i(1 + (Mx)_i)$. Note that $h_i \in C^\infty$ because it's a polynomial of $x$. Define $\tilde{h}_i : (-\epsilon_0, \epsilon_0) \to \mathbb{R}$ as $\tilde{h}_i(t) = h_i(p + td)$, which is also $C^\infty$. Since $p + td \in \mathscr{F}$, the range of $\tilde{h}_i$ is a subset of $[0, 1]$, i.e., $\tilde{h}_i(t) \in [0, 1]$ for all $t \in (-\epsilon_0, \epsilon_0)$. Observe that $\tilde{h}_i$ has either a minimum or a maximum at $t = 0$: $\tilde{h}_i(0) = 0$ if $i \in Z$ and $\tilde{h}_i(0) = 1$ if $i \in S$. Thus, $\tilde{h}_i'(0) = \nabla \tilde{h}_i(p) \cdot d = 0$. It means that the jacobian of $h(x) = (h_1(d), \cdots, h_d(x)) = x + x \odot (Mx)$ satisfies $\nabla h(p) \cdot d = 0$. Since $\nabla h(p)$ is a $P$-matrix and hence invertible, we have $d = 0$, contradicting $x \neq y$. Therefore no such distinct $x, y$ exist, and by definition $p$ is an extreme point of $\mathscr{F}$ in the relaxed sense.

$\square$

The next lemma proves the second inclusion.

**Lemma E.5.** *Given the feasible region* $\mathscr{F}$, *for any* $w \in [0, 1]^d \setminus \{0, 1\}^d$ *we have* $\Psi(w) \notin \text{Extr}_{\mathcal{R}} \mathscr{F}$.

*Proof.* Let $w = (w_1, \ldots, w_d) \in [0, 1]^d \setminus \{0, 1\}^d$ be a non-vertex point of the hypercube $[0, 1]^d$, and let $\Lambda := \Psi(w)$ be the image of $w$ by the map $\Psi$. Denote by $\mathcal{A}$ the set of indices corresponding to the active constraints for $\Lambda$, i.e.,

$$\mathcal{A} := \underbrace{\{k \in [d] \ : \ \lambda_k = 0\}}_{:=\mathcal{A}_1} \cup \underbrace{\{k \in [d] \ : \ \lambda_k(1 + (M\Lambda)_k) = 1\}}_{:=\mathcal{A}_2}. \tag{77}$$

Define the following functions:

$$f_k(x) := \begin{cases} \phi_k^1(x) := -x_k, & \text{if } k \in \mathcal{A}_1 \\ \phi_k^2(x) := x_k(1 + (Mx)_k) - 1, & \text{if } k \in \mathcal{A}_2 \end{cases}, \tag{78}$$

for every $k \in \mathcal{A}$ (with $\phi_k^1$ and $\phi_k^2$ are defined in a similar way for every $k \in [d]$), in such a way that the feasible region $\mathcal{F}$ can be re-written as follows:

$$\mathcal{F} = \left\{x \in \mathbb{R}^d \ : \ \phi_k^1(x) \leq 0, \phi_k^2(x) \leq 0 \text{ for all } k \in [d]\right\}. \tag{79}$$

The functions $(f_k)_{k \in \mathcal{A}}$ are differentiable and we have:

$$\nabla f_k(x) := \begin{cases} -e_k, & \text{if } k \in \mathcal{A}_1 \\ (1 + (Mx)_k)e_k + \lambda_k M_{k,\cdot} & \text{if } k \in \mathcal{A}_2 \end{cases}, \tag{80}$$

where $(e_k)_{k \in [d]}$ denotes the canonical basis of the $\mathbb{R}^d$, i.e., $e_k$ is the vector with the $k$-th entry equals 1 and all other entries equal 0, and $M_{k,\cdot}$ denotes the column vector of $\mathbb{R}^d$ whose $i$-th entry is $M_{k,i}$. Notice that since $w \notin \{0, 1\}^d$, then there exists at least one index $i_0 \in [d]$ such that $i_0 \notin \mathcal{A}$. Hence the vector space $E := \text{Span}\left(\{\nabla f_k(\Lambda)\}_{k \in \mathcal{A}}\right)$ has dimension less or equal than $d - 1$, then there exists a non-zero vector $v \neq 0$ in the orthogonal complement of the to this subspace, i.e., $v \in E$.

We prove that for sufficiently small $t$, $\Lambda \pm tv \in \mathcal{F}$. Let $k \in \mathcal{A}_1$. We have:

$$\phi_k^1(\Lambda \pm tv) = -(\lambda_k \pm tv_k), \tag{81}$$

and since $v \in E$, we have $\nabla f_k(\Lambda) \cdot v = 0$, which yields $v_k = 0$ using (80), and since $k \in \mathcal{A}_1$ we have $\lambda_k = 0$. This gives: $\phi_k^1(\Lambda \pm tv) = 0$, in particular:

$$\forall t > 0, \quad \phi_k^1(\Lambda \pm tv) \leq 0. \tag{82}$$

Besides, since $\phi_k^2(\Lambda) = -1 < 0$ and the map $\phi_k^2$ is continuous on $\mathbb{R}^d$, there exists $\varepsilon_1^k > 0$ such that:

$$\forall t \in (0, \varepsilon_1^k), \quad \phi_k^2(\Lambda \pm tv) \leq 0. \tag{83}$$

Now, fix $k \in \mathcal{A}_2$. We have:

$$\phi_k^2(\Lambda \pm tv) = (\lambda_k \pm tv_k)(1 + (M\Lambda)_k \pm t(Mv)_k) - 1. \tag{84}$$

Since this is a polynomial function of degree at most 2, we can identify the first two coefficients using *Taylor's theorem* as follows:

$$\phi_k^2(\Lambda \pm tv) = f_k(\Lambda) \pm t\nabla f_k(\Lambda) \cdot v + t^2 v_k(Mv)_k. \tag{85}$$

We have $f_k(\Lambda) = 0$ and by construction of the vector $v$, $\nabla f_k(\Lambda) \cdot v = 0$. Hence, $\phi_k^2(\Lambda \pm tv) = t^2 v_k(Mv)_k$. Using again (80), the condition $\nabla f_k(\Lambda) \cdot v = 0$ becomes:

$$(1 + (M\Lambda)_k)v_k + \lambda_k(Mv)_k = 0, \tag{86}$$

and since $k \in \mathcal{A}_2$, $\lambda_k > 0$ and $1 + (M\Lambda)_k > 0$, hence:

$$v_k(Mv)_k = -\frac{\lambda_k}{1 + (M\Lambda)_k}(Mv)_k^2 \leq 0. \tag{87}$$

Also, since $\phi_k^1(\Lambda) < 0$ and the map $\phi_k^1$ is continuous on $\mathbb{R}^d$, there exists $\varepsilon_2^k > 0$ such that:

$$\forall t \in (0, \varepsilon_2^k), \quad \phi_k^1(\Lambda \pm tv) \leq 0. \tag{88}$$

And finally, consider an index $k \in [d] \setminus \mathcal{A}$. By definition of $\mathcal{A}$, $\phi_k^1(\Lambda) < 0$ and $\phi_k^2(\Lambda) < 0$, so by the continuity of $\phi_k^1$ and $\phi_k^2$, there exists $\varepsilon_3^k > 0$ such that:

$$\forall t \in (0, \varepsilon_3^k), \quad \phi_k^1(\Lambda) \leq 0 \text{ and } \phi_k^2(\Lambda) \leq 0. \tag{89}$$

Combining all the previous results we have:

$$\forall t \in (0, \varepsilon), \quad \phi_k^1(\Lambda) \leq 0 \text{ and } \phi_k^2(\Lambda) \leq 0, \tag{90}$$

where

$$\varepsilon := \min\left(\min_{k \in \mathcal{A}_1} \varepsilon_1^k, \min_{k \in \mathcal{A}_2} \varepsilon_2^k, \min_{k \in [d] \setminus \mathcal{A}} \varepsilon_3^k\right) > 0.$$

Using Equation (79), this implies:

$$\forall t \in (0, \varepsilon), \quad \Lambda \pm tv \in \mathscr{F}. \tag{91}$$

Writing $\Lambda = \frac{(\Lambda + \varepsilon/2v) + (\Lambda - \varepsilon/2v)}{2}$, we conclude that $\Lambda \notin \text{Extr}_\mathcal{R}\, \mathscr{F}$. This achieves the proof. $\qquad\square$

$$\square$$

### E.4   OMITTED PROOFS IN SECTION 4.2

We start with a first lemma to show that the global maximizers of problem $(\mathscr{P}_d)$ from (37) cannot be in the interior region of $\mathscr{F}$.

**Lemma E.6** (Sub-optimality in the interior region $R_\mathscr{F}$ of $\mathscr{F}$). *For any point $p \in R_\mathscr{F}$, there exists $q \in R_\mathscr{F}$ such that*

$$\langle p \mid \mathbf{a} \rangle < \langle q \mid \mathbf{a} \rangle,$$

*that is, the global maximizers of problem $(\mathscr{P}_d)$ do not lie in the interior region $R_\mathscr{F}$ of $\mathscr{F}$.*

*Proof of Lemma E.6.* Let $p \in R_{\mathscr{F}}$ be some feasible point in the interior region of $\mathscr{F}$. Recall that according to Lemma D.7 (property 3), the interior region $R_{\mathscr{F}}$ is exactly the (topological) interior of $\mathscr{F}$, that is, $R_{\mathscr{F}} = \operatorname{int} \mathscr{F}$. Hence, as $p \in \operatorname{int} \mathscr{F}$ there exists some positive radius $r > 0$ such that the open ball $\mathrm{B}(p, r) \subseteq \mathscr{F}^{25}$ is still included in the feasible region. Then, take $q = p + \frac{r}{2} \cdot \frac{\mathbf{a}}{\|\mathbf{a}\|_2}$ so that $\|p - q\|_2 < r$ thus $q \in \mathscr{F}$ is still a feasible point and moreover

$$\langle q \mid \mathbf{a} \rangle = \left\langle p + \frac{r}{2\|\mathbf{a}\|_2}\mathbf{a} \mid \mathbf{a} \right\rangle = \langle p \mid \mathbf{a} \rangle + \underbrace{\frac{r\|\mathbf{a}\|_2}{2}}_{>0} > \langle p \mid \mathbf{a} \rangle,$$

as desired since $r > 0$ and $\|\mathbf{a}\|_2 > 0$. Thus, the point $p$ cannot be a global maximizer of problem $(\mathscr{P}_d)$. This achieves the proof of this lemma. $\qquad\square$

Now we give the proof of the main result.

**Theorem 4.4** (Global Maximizers of Problem $(\mathscr{P}_d)$)**.** *The set $X^*$ of the global maximizers of problem $(\mathscr{P}_d)$ as defined in (1) satisfies*

$$X^* \subseteq \left\{ \Psi(w) \ : \ w \in \{0, 1\}^d \right\}, \tag{92}$$

*that is, the global maximizers of $(\mathscr{P}_d)$ must be some points $p$ of the feasible region $\mathscr{F}$ which are mapped (through the bijection $\Psi^{-1}$) to the vertices of the unit hypercube $[0\,,1]^d$.*

So as to give a high-level overview of the proof, we start by a brief proof sketch.

*Proof (Sketch).* The proof of (92) is the culmination of several intermediate technical results (Lemmas D.10 to D.12) combined with previous results on the geometry of the feasible regions (Definitions D.4, D.5 and D.8 and Lemmas D.6 and D.7) and is based on an induction on "the number of tight constraints" in problem (1).

The proofs starts by Lemma E.6 showing that any points $p \in R_{\mathscr{F}}$, the *interior region* of $\mathscr{F}$ (see Definition D.5) is necessarily sub-optimal. This establishes the base case. Then, for the inductive step, starting at some point $p \in \mathscr{F}$ with at least one degree of freedom (see Definition D.8), we show thanks to Lemma D.12 that there exists some point $p' \in \mathscr{F}$ having at least one more degree of freedom than $p$ and such that, either

- $p'$ has the same objective value as $p$, i.e., $\langle \mathbf{a} \mid p \rangle = \langle \mathbf{a} \mid p' \rangle$,

- or we have $\langle \mathbf{a} \mid p \rangle < \langle \mathbf{a} \mid p' \rangle$ establishing the sub-optimality of $p$ regarding the objective value.

In the former case, we can apply the inductive hypothesis to conclude. Overall, our proof strategy can be summarized as follows: given $p \in \mathscr{F}$ with at least one degree of freedom, we construct a sequence $p = p_0, \ldots, p_\ell$ of feasible points such that

$$\langle \mathbf{a} \mid p_0 \rangle = \cdots = \langle \mathbf{a} \mid p_{\ell-1} \rangle < \langle \mathbf{a} \mid p_\ell \rangle.$$

We provide below a picture (Figure 4) explaining the construction of this sequence. This construction is permitted thanks to the technical lemma Lemma D.12. Given a point $p \in \mathscr{F}$ with at least one degree of freedom (e.g., being in the middle of one of the curved edges of $\mathscr{F}$ as in Figure 4), we can find a closed ball $\overline{\mathrm{B}}(p, \rho)$ with $\rho > 0$ such that its intersection with the intersection of all closed halfspaces associated to each of the tight constraints and "directed towards the region $\mathscr{F}$" is non-empty and included in the feasible set.

---

[25]The distance used here is the standard euclidean distance, induced by the 2-norm which we denote by $\|\cdot\|_2$.

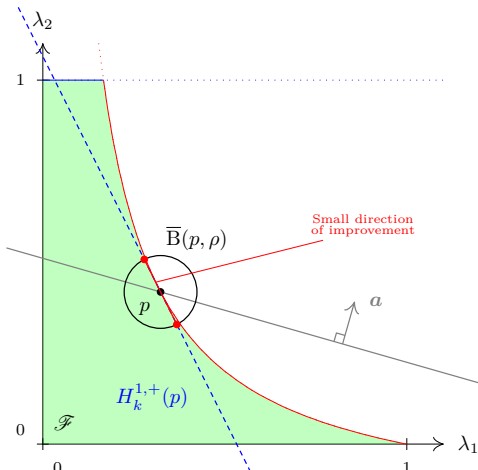

Figure 4: The technical result: Lemma D.12.

Moreover, if we consider the affine hyperplane induced by the objective function which goes through point $p$, i.e.,

$$H := \left\{ x \in \mathbb{R}^d \ : \ \langle \boldsymbol{a} \mid x \rangle = \langle \boldsymbol{a} \mid p \rangle \right\},$$

then to prove the sub-optimality of $p$, it remains to find some direction $v \in \mathbb{R}^d$ such that $(p+v) \in \mathscr{F}$ while $\langle \boldsymbol{a} \mid v \rangle > 0$. This can be done thanks to Lemmas D.2 and D.11. For instance, a quick inspection of Figure 4 shows that following the red segment inside the ball $\overline{\mathrm{B}}(p, \rho)$ is enough to prove the sub-optimality of $p$. $\qquad \square$

*Proof of Theorem 4.4.* Let $X^*$ be the set of the global maximizers of problem $(\mathscr{P}_d)$ and let $v^*$ be the optimal value of this problem. To show the above theorem, we proceed by strong backward induction on the number of degree of freedom of the components of $\mathscr{F}$. More precisely, we show that the hypothesis $(H_k)$: "for all $I \in \{-1, 0, 1\}^d$ with $|\mathrm{deg}(I)| = k$ then $\mathcal{C}_I \cap X^* = \varnothing$, i.e., for any $p \in \mathcal{C}_I$, we have $\langle p \mid \mathbf{a} \rangle < v^*$ (so that $p$ is a sub-optimal feasible point)" holds for all $k \in [d] = \{1, 2, \ldots, d\}$.

For the base case $k = d$, we know that there is a unique component of $\mathscr{F}$ which has exactly $d$ degrees of freedom and this component is the *interior region* $R_{\mathscr{F}}$ of $\mathscr{F}$ for which $I = (-1, \ldots, -1) \in \mathbb{R}^d$. Moreover, using Lemma E.6 we know that any point $p \in R_{\mathscr{F}}$ there exists another feasible point $q \in R_{\mathscr{F}}$ such that $\langle p \mid \mathbf{a} \rangle < \langle q \mid \mathbf{a} \rangle$ and since $q$ is feasible we obtain $\langle q \mid \mathbf{a} \rangle \leq v^*$ so

$$\langle p \mid \mathbf{a} \rangle < v^*, \tag{93}$$

which means the point $p$ is sub-optimal. As inequality (93) holds for any feasible point $p$ in the interior region $C_{(-1, \ldots, -1)}$ of $\mathscr{F}$, we deduce that the hypothesis $(H_d)$ holds for the unique component of degree $d$ of $\mathscr{F}$.

Now, assume the hypothesis $(H_\ell)$ holds for all $\ell \in [k+1 .. d]$, that is, for any such integer $\ell$ and any $I \in \{-1, 0, 1\}^d$ of degree $\ell$, the component $\mathcal{C}_I$ only contains sub-optimal points. For the inductive step, let $I = (i_1, \ldots, i_d) \in \{-1, 0, 1\}^d$ be a constraint index such that $|\mathrm{deg}(I)| = k$, we define

$$I_{\mathrm{free}} := \left\{ \ell \in [d] \ : \ i_\ell = -1 \right\},$$

and

$$J := \{ j \in [d] \ : \ i_j \neq -1 \} = [d] \setminus I_{\mathrm{free}}.$$

Then let $p \in \mathcal{C}_I$, since this point has at least one degree of freedom, we can apply Lemma D.12 thus, there exists some positive radius $\rho > 0$ such that for any point $y \in \mathrm{B}(p, \rho)$, if

$$y \in \bigcap_{\substack{\ell = 1 \\ i_\ell \in \{0, 1\}}}^{d} H_\ell^{i_\ell, +}(p),$$

then $y \in \mathscr{F}$ and moreover, $y$ has at least the same degrees of freedom $p$ has. In particular, this implies that the intersection of the affine supporting hyperplanes $H_\ell^{i_\ell,+}(p)$ for all $\ell \in J$, which is non-empty as it contains $p$ satisfies

$$\mathrm{B}(p,\rho) \cap \left(\bigcap_{\ell \in J} H_\ell^{i_\ell}(p)\right) \subseteq \mathscr{F}.$$

Moreover, by Lemma D.10 if we denote

$$A := \bigcap_{\ell \in J} H_\ell^{i_\ell}(p),$$

this affine subspace of $\mathbb{R}^d$ then $p \in A$ and $\dim A \geq d - |J| = |\deg(I)|$. Hence, we can extract from $A$ another affine subspace, say $B$, whose dimension is exactly $|\deg(I)|$. Additionally, since $p \in B$ and $p$ is the center of the non-empty open ball $\mathrm{B}(p,\rho)$ then let $\mathscr{C} := B \cap \mathrm{B}(p,\rho) \subseteq \mathscr{F}$ be the intersection between this affine subspace and the open ball. Notice that up to a invertible linear transformation (i.e., change of basis) $\mathscr{C}$ is a open disk of dimension $|\deg(I)|$[26]. Besides, let us consider the affine hyperplane $H_\mathbf{a}^\perp(p)$ orthogonal to the vector $\mathbf{a}$ which goes through point $p$, that is,

$$H_\mathbf{a}^\perp(p) := \left\{x \in \mathbb{R}^d \,:\, \langle x \mid \mathbf{a}\rangle = \langle p \mid \mathbf{a}\rangle\right\}.$$

Note that the points $x \in H_\mathbf{a}^\perp(p) \cap \mathscr{F}$ are all feasible and all have the same objective value than $p$ (which is $\langle p \mid \mathbf{a}\rangle$). Now, we distinguish two cases:

- if the affine subspace $B$ is not included in $H_\mathbf{a}^\perp(p)$ this means that we can find some non-zero vector $v \in (B - p)$ such that the line $(\ell)\colon p + tv$ for $t \in \mathbb{R}$ only intersects $H_\mathbf{a}^\perp(p)$ at point $p$, that is, $\langle v \mid \mathbf{a}\rangle \neq 0$. Hence, since $\mathscr{C} \cap (\ell) = (\ell) \cap \mathrm{B}(p,\rho)$ is a diameter of the open ball $\mathrm{B}(p,\rho)$ then there exists some $\varepsilon > 0$ such that the closed segment $[p-\varepsilon v, p+\varepsilon v] \subseteq \mathscr{C} \cap (\ell)$ hence, without loss of generality, we may assume $\langle v \mid \mathbf{a}\rangle > 0$ thus, since $p + \varepsilon v$ is both included in $\mathrm{B}(p,\rho)$ and in the affine subspace $B$ so it is a feasible point and its objective value is

  $$\langle p + \varepsilon v \mid \mathbf{a}\rangle = \langle p \mid \mathbf{a}\rangle + \varepsilon \langle v \mid \mathbf{a}\rangle > \langle p \mid \mathbf{a}\rangle,$$

  which implies that the point $p$ is sub-optimal.

- Otherwise, if the affine subspace $B$ is totally included in $H_\mathbf{a}^\perp(p)$ then it is also the case for $\mathscr{C}$ and again, we distinguish two cases

  - if there exists some point $y \in \mathscr{C}$ such that $y \notin \bigcap_{j \in J} \partial(\mathrm{epi}\, g_j^{i_j})$ then, if we denote by $I' = (i'_1, \ldots, i'_d)$ the constraint index of $y$, we know by Lemma D.12 and since $y \in \mathrm{B}(p,\rho) \cap \mathcal{F}$ that $y$ has at least the same degrees of freedom that $p$ so

    $$I_\text{free} \subseteq I'_\text{free} := \left\{j \in [d] \,:\, i'_j = -1\right\},$$

    and, if we have $I_\text{free} = I'_\text{free}$ then we would have $J' := \left\{j \in [d] \,:\, i'_j \in \{0,1\}\right\} = J$ hence by Lemma D.7 (property 4)

    $$y \in \bigcap_{j \in J} \partial(\mathrm{epi}\, g_j^{i_j}),$$

    which is not possible. Thus, necessarily, the point $y$ must have at least one more degree of freedom than $p$, i.e., $|\deg(I')| > |\deg(I)|$. Next, as $y$ and $p$ belong to the same affine hyperplane $H_\mathbf{a}^\perp(p)$ we have $\langle y \mid \mathbf{a}\rangle = \langle p \mid \mathbf{a}\rangle$ and, using the induction hypothesis we conclude that

    $$\langle p \mid \mathbf{a}\rangle = \langle y \mid \mathbf{a}\rangle < v^*,$$

    so $p$ is again, sub-optimal.

---

[26]For examples, if $|\deg(I)| = 1$ then $\mathscr{C}$ would be a diameter of $\mathrm{B}(p,\rho)$, if $|\deg(I)| = 2$ then $\mathscr{C}$ would be a 2-dimensional (open) disk included in $\mathrm{B}(p,\rho)$ and so on...

– Otherwise, assume the intersection of $B$ with the open ball $\mathrm{B}(p, \rho)$ is included in $\bigcap\limits_{j \in J} \partial(\mathrm{epi}\, g_i^{i_j})$. Then we first show that the affine subspace $B$ satisfies

$$B \subseteq \bigcap_{j \in J} \partial(\mathrm{epi}\, g_j^{i_j}).$$

To do so, for any vector $v \in (B - p)$ there exists some $\varepsilon > 0$ such that the point $(p + \varepsilon v) \in \mathscr{C} = \mathrm{B}(p, \rho) \cap B$ and due to the symmetry of the open ball we deduce that we also have $(p - \varepsilon v) \in \mathscr{C}$. Hence the segment $[p - \varepsilon v, p + \varepsilon v]$ is included in $\mathrm{B}(p, \rho) \cap B$ (it is a portion of a diameter of $\mathrm{B}(p, \rho)$) so it is included in every $\partial(\mathrm{epi}\, g_j^{i_j})$ for $j \in J$ by assumption thus, according to Lemma D.2 (property 2, "converse" part) we deduce that the whole line $(\ell_v)\colon p + tv,\ t \in \mathbb{R}$ is included in every hypersurface $\partial(\mathrm{epi}\, g_j^{i_j})$ for $j \in J$ and because this holds for all vector $v \in (B - p)$, we obtain the desired inclusion, $B \subseteq \bigcap\limits_{j \in J} \partial(\mathrm{epi}\, g_j^{i_j})$.

From here, we now use Lemma D.11 since $J = \{j \in [d] \ :\ i_j \in \{0, 1\}\}$ and $\dim B = |\deg(I)| = |I_{\mathrm{free}}|$. Hence, we obtain that $B = p + \mathrm{Vect}_{\mathbb{R}}\left((e_i)_{i \in I_{\mathrm{free}}}\right)$ and $I_{\mathrm{free}} \neq \varnothing$ but, as we assume in this case and the previous one that we have $B \subseteq H_{\mathbf{a}}^{\perp}(p)$ then $H_{\mathbf{a}}^{\perp}(p) - p$ contains the basis vector $(e_i)_{i \in I_{\mathrm{free}}}$ so by definition we obtain for any $i \in I_{\mathrm{free}}$

$$\langle e_i \mid \mathbf{a} \rangle = a_i = 0,$$

which is absurd since all the coordinates of the vector $\mathbf{a}$ are non-zero (see for instance the definition of the optimization problem $(\mathscr{P}_d)$ in (1)). Therefore, we conclude that this case is not possible hence, the intersection of $B$ with the open ball $\mathrm{B}(p, \rho)$, that is the open disk $\mathscr{C}$, cannot be fully included in $\bigcap\limits_{j \in J} \partial(\mathrm{epi}\, g_j^{i_j})$. Thus only the previous case can happen and we have showed that the point $p$ was sub-optimal.

Thus in all the cases, when some point $p \in \mathscr{F}$ belongs to a component of the feasible region with exactly $k$ degrees of freedom, we have shown that it is always sub-optimal. Hence, the hypothesis $(H_k)$ holds and by strong backward induction, we conclude that the hypothesis $(H_k)$ holds for all integer $k \in [1 .. d]$. Thus, all points $p \in \mathscr{F}$ having one or more degree of freedom are sub-optimal which shows that the set of the global maximizers $X^*$ of problem $(\mathscr{P}_d)$ must be included in the set of feasible points which have no degree of freedom, that is to say,

$$X^* \subseteq \left\{ \mathcal{E}_I \ :\ I \in \{0, 1\}^d \right\}.$$

This achieves the proof of the theorem. $\qquad\square$

### E.5 Omitted Proofs in Appendix I

**Lemma I.1.** *For any positive integer $d \geq 2$, there exists a strictly upper triangular $\mathbb{R}^{d \times d}$ matrix $M$ with non-negative entries and a vector $\mathbf{a} \in \mathbb{R}_+^d$ such that problem $(\mathscr{P}_d)$ admits at least two solutions in $\mathbb{R}_+^d$.*

*Proof of Lemma I.1.* Fix $d \geq 0$. We construct a counter-example to the uniqueness of the global maximizers to the problem $(\mathscr{P}_d)$. For that, we consider the instance of the problem (1) given by the matrix $M$ and the vector $\mathbf{a}$ defined as follows:

$$M = \begin{pmatrix} 0 & \cdots & 0 & 1 \\ 0 & \cdots & 0 & 0 \\ \vdots & \ddots & \vdots & \vdots \\ 0 & \cdots & 0 & 0 \end{pmatrix} \in \mathbb{R}^{d \times d} \tag{94}$$

$$\text{and } \mathbf{a} = \begin{pmatrix} 2 \\ 1 \\ \vdots \\ 1 \end{pmatrix} \in \mathbb{R}^d. \tag{95}$$

In this case, the problem $(\mathscr{P}_d)$ becomes equivalent to:

$$(\mathscr{P}_d): \quad \text{maximize } F(\Lambda) := 2\lambda_1 + \lambda_2 + \cdots + \lambda_d$$

$$\text{subject to} \begin{cases} 0 \le \lambda_1 \, (1 + \lambda_d) \le 1 \\ 0 \le \lambda_2 \le 1 \\ \vdots \\ 0 \le \lambda_d \le 1 \end{cases} . \tag{96}$$

First, we prove that the optimal value of this problem is $d$. For that, notice that the first bilinear constraint implies that $\lambda_1 \le \frac{1}{1+\lambda_d}$, which implies that for all feasible point $\Lambda \in \mathscr{F}$, we have:

$$F(\Lambda) \le \frac{2}{1 + \lambda_d} + \lambda_d + \lambda_2 + \cdots + \lambda_{d-1} \tag{97}$$

$$\le \underbrace{\frac{2}{1 + \lambda_d} + \lambda_d}_{:= f(\lambda_d)} + (d - 2), \tag{98}$$

where the last inequality follows from the constraints $\lambda_i \le 1$ for $i \in [2 .. d - 1]$. Notice that:

$$f'(\lambda_d) = 1 - \frac{2}{(1 + \lambda_d)^2} \tag{99}$$

$$f''(\lambda_d) = \frac{4}{(1 + \lambda_d)^3} \ge 0, \tag{100}$$

which implies that $f$ is strictly convex on $[0, 1]$ and hence it can only attain its maximum in one of the extreme points of the segment $[0, 1]$. Since $f(0) = f(1) = 2$, it follows that $f(\lambda_d) \le 2$ and hence $F(\Lambda) \le d$. Furthermore, notice that

$$F(\Lambda_1^*) = F(\Lambda_2^*) = d, \tag{101}$$

where

$$\Lambda_1^* := \begin{pmatrix} 1/2 \\ 1 \\ \vdots \\ 1 \end{pmatrix} \quad \text{and} \quad \Lambda_2^* := \begin{pmatrix} 1 \\ \vdots \\ 1 \\ 0 \end{pmatrix},$$

are both feasible points of the problem $(\mathscr{P}_d)$. Hence both points are global maximizers. This achieves the proof. $\square$

**Lemma I.2.** *For any $2 \times 2$ strictly upper triangular matrix $M$ with non-negative entries, if $\mathbf{a} = (1, 1)^\top$ then the problem $(\mathscr{P}_2)$ admits a unique global maximizer.*

*Proof of Lemma I.2.* Let $M$ be a $2 \times 2$ strictly upper triangular matrix with non-negative entries, then there exists some real number $m \ge 0$ such that

$$M = \begin{pmatrix} 0 & m \\ 0 & 0 \end{pmatrix}. \tag{102}$$

In this case where $\mathbf{a} = (1, 1)^\top$, the problem $(\mathscr{P}_2)$ can be written as:

$$(\mathscr{P}_2): \quad \text{maximize } F(\Lambda) := \lambda_1 + \lambda_2$$

$$\text{subject to} \begin{cases} 0 \le \lambda_1 \, (1 + m\lambda_2) \le 1 \\ 0 \le \lambda_2 \le 1 \end{cases} . \tag{103}$$

In the case where $m = 0$, it is clear that the problem $(\mathscr{P}_2)$ admits one unique global maximizer, which is given by $(\lambda_1, \lambda_2) = (1, 1)$. Now suppose that $m > 0$.

It follows from the first bilinear constraint that for all $\Lambda$ in the feasible region $\mathscr{F}$, we have:

$$F(\Lambda) \le f(\lambda_2) := \lambda_2 + \frac{1}{1 + m\lambda_2}. \tag{104}$$

We compute the two first derivatives of $f$:

$$f'(\lambda_2) = 1 - \frac{m}{(1 + m\lambda_2)^2} \tag{105}$$

$$f''(\lambda_2) = \frac{m^2}{(1 + m\lambda_2)^3}. \tag{106}$$

Since $f''(\lambda_2) > 0$ for every $\lambda_2 \in [0, 1]$, it follows that $f$ is a strictly convex function on $[0, 1]$ and hence it can only achieve its maximum in the extreme points of the interval $[0, 1]$, i.e., 0 and 1. We have:

$$f(0) = 1, \quad f(1) = 1 + \frac{1}{1 + m}. \tag{107}$$

Since $f(0) < f(1)$, the function $f$ admits a unique maximizer given by $\lambda_2 = 1$.

Now, notice that $(\frac{1}{1+m}, 1)$ is a feasible point and

$$F\left(\left(\frac{1}{1 + m}, 1\right)\right) = 1 + \frac{1}{1 + m}. \tag{108}$$

Besides, if $\Lambda$ is a feasible point such that $\lambda_2 < \frac{1}{1+m}$, then $F(\Lambda) \leq f(\lambda_2) < 1 + \frac{1}{1+m}$, so $\Lambda$ is not a maximizer of $(\mathscr{P}_2)$. And if $\Lambda$ is a feasible point such that $\lambda_1 < 1$ and $\lambda_2 = \frac{1}{1+m}$, then $F(\Lambda) < 1 + \frac{1}{1+m}$.

Hence, the only global maximizer of $(\mathscr{P}_2)$ is $(\frac{1}{1+m}, 1)$. $\qquad \square$

Now, we prove the correctness of the claim made in Appendix I, that is to say, the instance of $(\mathscr{P}_3)$ given by:

$$M = \begin{pmatrix} 0 & 2 & 0 \\ 0 & 0 & 1 \\ 0 & 0 & 0 \end{pmatrix} \text{ and } \mathbf{a} = \begin{pmatrix} 1 \\ 1 \\ 1 \end{pmatrix} \tag{109}$$

has the following two maximizers:

$$\Lambda_1^* = \begin{pmatrix} 1 \\ 0 \\ 1 \end{pmatrix} \text{ and } \Lambda_2^* = \begin{pmatrix} 1/2 \\ 1/2 \\ 1 \end{pmatrix}. \tag{110}$$

In this case, the problem $(\mathscr{P}_3)$ becomes equivalent to:

$$(\mathscr{P}_3): \quad \text{maximize } F(\Lambda) := \lambda_1 + \lambda_2 + \lambda_3$$

$$\text{subject to } \begin{cases} 0 \leq \lambda_1 (1 + 2\lambda_2) \leq 1 \\ 0 \leq \lambda_2 (1 + \lambda_3) \leq 1 \\ 0 \leq \lambda_3 \leq 1 \end{cases}. \tag{111}$$

From the first bilinear constraint, it follows that for all feasible $\Lambda \in \mathscr{F}$, $\lambda_1 \leq \frac{1}{1+2\lambda_2}$. Hence, for all $\Lambda \in \mathscr{F}$,

$$F(\Lambda) \leq \underbrace{\frac{1}{1 + 2\lambda_2}}_{:=g(\lambda_2)} + \lambda_2 + \lambda_3. \tag{112}$$

We have:

$$g'(\lambda_2) = 1 - \frac{2}{(1 + 2\lambda_2)^2} \tag{113}$$

$$g''(\lambda_2) = \frac{8}{(1 + 2\lambda_3)^3} \geq 0, \tag{114}$$

hence $g$ is strictly convex on $[0, \frac{1}{1+\lambda_3}]$, so it can attain its maximum only in an extreme point of $[0, \frac{1}{1+\lambda_3}]$. We have:

$$g(0) = 1, \quad g\left(\frac{1}{1+\lambda_3}\right) = \frac{1}{1 + \frac{2}{1+\lambda_3}} + \frac{1}{1+\lambda_3} \tag{115}$$

$$= \frac{1+\lambda_3}{3+\lambda_3} + \frac{1}{1+\lambda_3} \tag{116}$$

$$= 1 - \frac{2}{3+\lambda_3} + \frac{1}{1+\lambda_3}. \tag{117}$$

Hence

$$g(0) + \lambda_3 = 1 + \lambda_3 \leq 2 \tag{118}$$

$$\tag{119}$$

and

$$g\left(\frac{1}{1+\lambda_3}\right) + \lambda_3 = \underbrace{1 - \frac{2}{3+\lambda_3} + \frac{1}{1+\lambda_3} + \lambda_3}_{:=h(\lambda_3)}. \tag{120}$$

We have:

$$h'(\lambda_3) = \frac{2}{(3+\lambda_3)^2} - \frac{1}{(1+\lambda_3)^2} + 1 \tag{121}$$

$$h''(\lambda_3) = -\frac{4}{(3+\lambda_3)^3} + \frac{2}{(1+\lambda_3)^3}. \tag{122}$$

We have for all $\lambda_3 \in [0, 1]$,

$$\frac{2}{(1+\lambda_3)^3} \geq \frac{4}{(3+\lambda_3)^3} \tag{123}$$

$$\iff (3+\lambda_3)^3 \geq 2(1+\lambda_3)^3 \tag{124}$$

$$\iff \left(\frac{3+\lambda_3}{1+\lambda_3}\right)^3 \geq 2 \tag{125}$$

$$\iff \left(1 + \frac{2}{1+\lambda_3}\right)^3 \geq 2, \tag{126}$$

which clearly holds since for all $\lambda_3$, $1 + \frac{2}{1+\lambda_3} \geq 2$, so $\left(1 + \frac{2}{1+\lambda_3}\right)^3 \geq 8 \geq 2$. This implies that $h$ is strictly convex on $[0, 1]$, and given that $h(0) = \frac{1}{3}$ and $h(1) = 2$, it follows that:

$$g\left(\frac{1}{1+\lambda_3}\right) + \lambda_3 \leq 2. \tag{127}$$

Thus, for all feasible $\Lambda \in \mathcal{F}$, we have $F(\Lambda) \leq 2$. Furthermore, it is clear that $\Lambda_1^*$ and $\Lambda_2^*$ defined by:

$$\Lambda_1^* := \begin{pmatrix} 1 \\ 0 \\ 1 \end{pmatrix} \text{ and } \Lambda_2^* := \begin{pmatrix} 1/2 \\ 1/2 \\ 1 \end{pmatrix} \tag{128}$$

are both feasible points of $(\mathscr{P}_3)$ and are such that:

$$F(\Lambda_1^*) = F(\Lambda_2^*) = 2, \tag{129}$$

hence 2 is the maximal value of $(\mathscr{P}_3)$ and both $\Lambda_1^*$ and $\Lambda_2^*$ are global maximizers. This concludes the proof of the claim.

**Theorem I.3** (A Sufficient Condition for Uniqueness). *For any positive integer $d$, if the matrix $M$ is strictly upper triangular with non-negative entries and satisfies, for all $k \in [d]$*

$$\sum_{\substack{i=1 \\ i<k}}^{d} M_{i,k} < 1, \tag{130}$$

*then with the vector $\mathbf{a} = (1, \ldots, 1)^\top \in \mathbb{R}^d$ the problem $(\mathscr{P}_d)$ admits a unique global maximizer.*

*Proof.* When $\mathbf{a}$ has only one entries, the objective function to maximize is $F(\Lambda) := \sum_{i=1}^{d} \lambda_i$.

We start by stating and proving the next lemma that an optimal solution has necessarily tight inequalities from the right side for all the bilinear constraints.

**Lemma E.7.** *Let $\Lambda$ be any feasible solution to $(\mathscr{P}_d)$ such that $\lambda_k(1 + (M\Lambda)_k) < 1$ for some $k \in [d]$, then, under the assumptions of Theorem I.3, there exists another feasible point $\tilde{\Lambda}$ such that $F(\tilde{\Lambda}) > F(\Lambda)$, i.e., $\Lambda$ cannot be a global maximizer of the problem $(\mathscr{P}_d)$.*

*Proof.* Fix a feasible $\Lambda$ and an index $k \in [d]$ such that $\lambda_k(1 + (M\Lambda)_k) < 1$, i.e.,

$$\lambda_k < \frac{1}{1 + \sum_{j>k} M_{j,k}\lambda_k}, \tag{131}$$

and set:

$$\varepsilon := \min\left( \frac{1}{1 + \sum_{j>k} M_{k,j}\lambda_j} - \lambda_k, \frac{1}{2\left(1 + \sum_{i<k} M_{i,k}\right)} \right). \tag{132}$$

By assumption we have $\varepsilon > 0$. We construct the new point $\tilde{\Lambda}$ as follows:

$$\tilde{\lambda}_i := \begin{cases} \frac{\lambda_i}{1 + M_{i,k}\varepsilon}, & \text{if } i < k \\ \lambda_k + \varepsilon, & \text{if } i = k \\ \lambda_i, & \text{if } i > k \end{cases}. \tag{133}$$

First, we prove that $\tilde{\Lambda}$ is also a feasible solution to $(\mathscr{P}_d)$. It is clear that $\tilde{\lambda}_i \geq 0$ for every $i \in [d]$ (because $\Lambda$ is a feasible point and $\varepsilon > 0$). The (bilinear) constraints corresponding to indices $i$ with $i > k$ are clearly satisfied by the new point since $\tilde{\lambda}_i = \lambda_i$ for any $j > i$. For the $k$-th constraint, we have:

$$h_k(\tilde{\Lambda}) := \tilde{\lambda}_k \left( 1 + \sum_{j>k} M_{k,j}\tilde{\lambda}_j \right) \tag{134}$$

$$= (\lambda_k + \varepsilon)\left( 1 + \sum_{j>k} M_{k,j}\lambda_j \right) \tag{135}$$

$$\leq \frac{1}{1 + \sum_{j>k} M_{k,j}\lambda_j}\left( 1 + \sum_{j>k} M_{k,j}\lambda_j \right) \tag{136}$$

$$= 1, \tag{137}$$

where the inequality follows from the definition of $\varepsilon$ as a minimum, yielding

$$\varepsilon \leq \frac{1}{1 + \sum_{j>k} M_{k,j}\lambda_j} - \lambda_k.$$

Besides, since $\Lambda$ is a feasible point, $\lambda_i \geq 0$ for every $i \in [d]$, which implies that $h_k(\tilde{\Lambda}) \geq 0$, hence $\tilde{\Lambda}$ satisfies the $k$-th constraint.

Now fix $i < k$. We have:

$$1 + \sum_{j>i} M_{i,j}\tilde{\lambda}_j = 1 + M_{i,k}\tilde{\lambda}_k + \sum_{j>i,j\neq k} M_{i,j}\tilde{\lambda}_j \tag{138}$$

$$= 1 + \underbrace{\sum_{j>i} M_{i,j}\lambda_j}_{:=S_i \geq 1} + M_{i,k}\varepsilon + \sum_{i<j<k} M_{i,j}\left(\frac{\lambda_j}{1 + M_{j,k}\varepsilon} - \lambda_j\right) \tag{139}$$

$$= S_i + M_{i,k}\varepsilon - \sum_{i<j<k} \frac{M_{i,j}\lambda_j M_{j,k}\varepsilon}{1 + M_{j,k}\varepsilon} \tag{140}$$

$$\leq S_i + M_{i,k}\varepsilon, \tag{141}$$

where the last inequality follows from the non-positivity of the last term in (140). Now multiply by $\tilde{\lambda}_i = \lambda_i/(1 + M_{i,k}\varepsilon)$ (which is non-negative):

$$h_i(\tilde{\Lambda}) = \tilde{\lambda}_i \left(1 + \sum_{j>i} M_{i,j}\tilde{\lambda}_j\right) \tag{142}$$

$$\leq \frac{\lambda_i(S_i + M_{i,k}\varepsilon)}{1 + M_{i,k}\varepsilon} \tag{143}$$

$$\leq \lambda_i S_i \tag{144}$$

$$\leq 1, \tag{145}$$

where the second inequality holds because $S_i \geq 0$, and the last inequality follows from the feasibility of $\Lambda$. Hence the point $\tilde{\Lambda}$ verifies the $i$-th bilinear constraint. We conclude that $\tilde{\Lambda}$ is a feasible solution to $(\mathcal{P}_d)$.

Finally, we prove that $\tilde{\Lambda}$ has a (strictly) greater objective value than $\Lambda$, i.e.,

$$F(\tilde{\Lambda}) - F(\Lambda) = \sum_{i\in[d]} (\tilde{\lambda}_i - \lambda_i) > 0.$$

The gain at coordinate $k$ is:

$$\tilde{\lambda}_k - \lambda_k = \varepsilon. \tag{146}$$

The maximum loss we can get at coordinate $i$ with $i < k$ is:

$$\lambda_i - \tilde{\lambda}_i = \lambda_i \left(1 - \frac{1}{1 + M_{i,k}\varepsilon}\right) \tag{147}$$

$$= \frac{\lambda_i M_{i,k}\varepsilon}{1 + M_{i,k}\varepsilon} \tag{148}$$

$$\leq \lambda_i M_{i,k}\varepsilon. \tag{149}$$

We sum the losses over $i < k$:

$$\sum_{i<k} (\lambda_i - \tilde{\lambda}_i) \leq \varepsilon \sum_{i<k} M_{i,k}\lambda_i \leq \varepsilon \sum_{i<k} M_{i,k}. \tag{150}$$

$$\tag{151}$$

Hence,

$$F(\tilde{\Lambda}) - F(\Lambda) = \sum_{i\in[d]} (\tilde{\lambda}_i - \lambda_i) \geq \varepsilon \left(1 - \sum_{i<k} M_{i,k}\right) > 0, \tag{152}$$

where the last inequality follows from the assumption of Theorem I.3. $\qquad\square$

Now, using Lemma E.7 implies that any optimal solution $\Lambda^*$ of $(\mathcal{P}_d)$ must verify:

$$\forall k \in [d], \quad \lambda_k^* \left(1 + \sum_{j=k+1}^{d} M_{k,j}\lambda_j^*\right) = 1. \tag{153}$$

Hence $(\mathscr{P}_d)$ admits a unique maximizer $\Lambda^*$ which can be constructed by backward induction as follows:

$$\lambda_d^* = 1, \tag{154}$$

and for all $k < d$

$$\lambda_k^* = \frac{1}{1 + \sum\limits_{j=k+1}^{d} M_{k,j} \lambda_j^*}. \tag{155}$$

This achieves the proof of our sufficient condition for uniqueness. $\qquad\square$

# F THE STRICTLY UPPER TRIANGULAR CASE

## F.1 CHARACTERIZATION OF THE EXTREME POINTS OF $\mathscr{F}$

**Theorem 4.2** (Extreme points of $\mathscr{F}$ in the Strictly Upper Triangular Case). *For the feasible region $\mathscr{F}$ of the problem $(\mathscr{P}_d)$ in the particular case where the matrix $M$ is strictly upper triangular with non-negative entries, we have*

$$\operatorname{Extr} \mathscr{F} = \left\{ \Psi(w) \,:\, w \in \{0, 1\}^d \right\}, \tag{156}$$

*that is, the extreme points of $\mathscr{F}$ are exactly the vertices of the hypercube $[0\,,1]^d$ mapped by the diffeomorphism $\Psi$.*

*Proof of Theorem 4.2.* We first prove the first inclusion:

**Lemma F.1.** *Given the feasible region $\mathscr{F}$, we have the inclusion*

$$\left\{ \Psi(w) \,:\, w \in \{0, 1\}^d \right\} \subseteq \operatorname{Extr} \mathscr{F}.$$

*Proof of Lemma F.1.* Let $w = (w_1, \ldots, w_d) \in \{0, 1\}^d$ be a vertex of the hypercube $[0\,,1]^d$ and assume, for the sake of contradiction that $\Psi(w) \in \mathscr{F}$ is not an extreme point, i.e., $\Psi(w) \notin \operatorname{Extr} \mathscr{F}$. Then, following Definition 3.2, there must exist $x, y \in \mathscr{F}$ with $x \neq y$ such that $p := \Psi(w) \in (x\,, y)$. Since $p$ lies in the interior of the closed segment $[x\,, y]$, there exists some vector $v = (v_1, \ldots, v_d) \in \mathbb{R}^d \setminus \{0\}$ and scalars $t_x, t_y \in \mathbb{R}^*$ such that $t_x t_y < 0$ (because $x$ and $y$ are on both side of $p$) and

$$x = p + t_x v \quad \text{and} \quad y = p + t_y v. \tag{157}$$

Without loss of generality, we assume $t_x > 0$ so $t_y < 0$.

We first prove the following lemma.

**Lemma F.2.** *If for some $i \in [d]$ we have $w_i = 0$ then $p_i = 0$ and $x_i = 0 = y_i$.*

*Proof of Lemma F.2.* If $w_i = 0$ for some $i \in [d]$, we show that $v_i = 0$ and this will imply that both $x_i = 0 = y_i$ since, as defined in (157), both $x = p + t_x v$ and $y = p + t_y v$. So assume for the sake of contradiction that $v_i \neq 0$, and without loss of generality, we may assume $v_i > 0$. Since $t_x > 0$ and $t_x t_y < 0$, we deduce that $t_y < 0$ so $t_y v_i < 0$ thus

$$y_i = p_i + t_y v_i < p_i.$$

But, recall that $w_i = 0$ and since $p = \Psi(w)$, the $i$-th coordinate of $p$ reads (following the definition of $\Psi$ from (3.6)),

$$p_i \left( 1 + \sum_{j=i+1}^{d} M_{i,j} p_j \right) = w_i = 0,$$

so $p_i = 0$ since $p \in \mathscr{F} \subseteq \mathbb{R}_+^d$ and

$$1 + \sum_{j=i+1}^{d} M_{i,j} p_j \geq 1.$$

Hence, we found that $y_i < 0$ which is a contradiction since $y \in \mathscr{F}$. Finally, we conclude that we must have $p_i = 0$ and $v_i = 0$ so $x_i = 0 = y_i$ as claimed. $\qquad \square$

Besides, recall that $p = \Psi(w)$ thus, by definition of $\Psi$

$$p_i \left( 1 + \sum_{j=i+1}^{d} M_{i,j} p_j \right) = w_i,$$

for all $i \in [d]$. Hence, $p$ lies at the boundary of all the hypersurface $\partial \left( \text{epi } g_i^w \right)$, i.e.

$$\{p\} = \bigcap_{i=1}^{d} \partial \left( \text{epi } g_i^w \right),$$

where for all $i \in [d]$, the hypersurface $\partial \left( \text{epi } g_i^w \right)$ is

$$\partial \left( \text{epi } g_i^w \right) = \left\{ (x_1, \ldots, x_d) \ : \ x_i = w_i \left( 1 + \sum_{j=i+1}^{d} M_{i,j} x_j \right)^{-1} \right\}. \tag{158}$$

We now proceed by strong backward induction on $i \in [d]$ to show that $x_i = p_i = y_i$ and $v_i = 0$. For the base case $i = d$, since

$$\partial \left( \text{epi } g_d^w \right) = \{ (x_1, \ldots, x_d) \ : \ x_d = w_d \},$$

then $p_d = w_d \in \{0, 1\}$. If $w_d = 0$ then using Lemma F.2 we would have directly $x_d = 0 = y_d$. Now, if $w_d = 1$, we assume for the sake of contradiction that $v_d \neq 0$, and without loss of generality, we may suppose $v_d > 0$. Then, since $t_x > 0$ we obtain

$$x_d = p_d + t_x v_d = w_d + t_x v_d = 1 + t_x v_d > 1,$$

which is impossible since $x$ would lie outside of the closed unit hypercube $[0, 1]^d$. Thus, we deduce that $x_d = p_d = y_d$ and $v_d = 0$.

Next, suppose the hypothesis holds for all $i \in \{k + 1, \ldots, d\}$ for some integer $k \in [d - 1]$ that is, $x_i = p_i = y_i$ and $v_i = 0$ for all $i \in [k + 1 \,..\, d]$. Then for the $k$-coordinate, either $w_k = 0$ in which case Lemma F.2 allows us to conclude that $x_k = 0 = y_k$. Otherwise, if $w_k = 1$ then $p$ belongs to

$$\partial \left( \text{epi } g_k^w \right) = \left\{ (x_1, \ldots, x_d) \ : \ x_k = w_k \left( 1 + \sum_{j=k+1}^{d} M_{k,j} x_j \right)^{-1} \right\},$$

Assume for the sake of contradiction that $v_k \neq 0$, and without loss of generality, we still suppose $v_k > 0$. Then, we obtain (recall here $w_k = 1$):

$$x_k = p_k + t_x v_k$$

$$= \left( 1 + \sum_{j=k+1}^{d} M_{k,j} p_j \right)^{-1} + t_x v_k$$

$$\overset{(a)}{=} w_k \left( 1 + \sum_{j=k+1}^{d} M_{k,j} x_j \right)^{-1} + t_x v_k$$

$$\overset{(b)}{>} \left( 1 + \sum_{j=k+1}^{d} M_{k,j} p_j \right)^{-1},$$

where in (a) we use the fact that $x_j = p_j$ for all $j \in [k + 1 \,..\, d]$ by the induction hypothesis while in (b) we use the inequality $t_x v_k > 0$. Hence, we deduce that

$$x_k > w_k \left( 1 + \sum_{j=k+1}^{d} M_{k,j} x_j \right)^{-1},$$

from where $x \in \text{int } \left( \text{epi } g_k^w \right)$ which is not possible since by Lemma D.9 we have

$$\mathscr{F} = [0, 1]^d \setminus \bigcup_{i=1}^{d} \text{int } \left( \text{epi } g_i^1 \right),$$

and as $w_k = 1$ then $\text{epi } g_k^w = \text{epi } g_k^1$. Thus, we must have $v_k = 0$ from where $x_k = p_k = y_k$ and this completes the inductive step and the proof of the lemma. $\qquad \square$

The next lemma states the second inclusion:

**Lemma F.3.** *Given the feasible region $\mathscr{F}$, for any $w \in [0,1]^d \setminus \{0,1\}^d$ we have $\Psi(w) \notin \mathrm{Extr}\,\mathscr{F}$.*

*Proof of Lemma F.3.* Let $w = (w_1, \ldots, w_d) \in [0,1]^d \setminus \{0,1\}^d$ then there exists some $i \in [d]$ such that $w_i \in (0,1)$. Our goal is to construct $w^1$ and $w^2$ in $[0,1]^d$ such that $w^1 \neq w^2$ and $\Psi(w) \in (\Psi(w^1), \Psi(w^2))$. Notice that this implies that $\Psi(w) \notin \mathrm{Extr}\,\mathscr{F}$ since $\Psi(w^1), \Psi(w^2) \in \mathscr{F}$. To simplify the notations, we introduce the three vectors $p := \Psi(w)$, $p^1 := \Psi(w^1)$ and $p^2 := \Psi(w^2)$. More precisely, we construct $w^1$ and $w^2$ such that the following holds

$$p_k = (1 - w_i)p_k^1 + w_i p_k^2, \quad \text{for every } k \in [d]. \tag{159}$$

For every $k > i$, we take $w_k^1 = w_k^2 = w_k \in [0,1]$. By strong backward induction on $k \in [i+1..d]$, we show that $p_k = p_k^1 = p_k^2$. Besides, recall that $p = \Psi(w)$ thus, by definition of $\Psi$

$$p_i \left(1 + \sum_{j=i+1}^{d} M_{i,j} p_j\right) = w_i, \tag{160}$$

for all $i \in [d]$.

For the base case $k = d$, we have $p_d = w_d$, $p_d^1 = w_d^1$ and $p_d^2 = w_d^2$. Hence, $p_d = p_d^1 = p_d^2$. Next, suppose the hypothesis holds for all $k \in [\ell+1..d]$ for some $\ell > i$. We have

$$p_\ell = w_\ell \left(1 + \sum_{j=\ell+1}^{d} M_{\ell,j} p_j\right)^{-1} \tag{161}$$

$$= \begin{cases} w_\ell^1 \left(1 + \displaystyle\sum_{j=\ell+1}^{d} M_{\ell,j} p_j^1\right)^{-1} \\ w_\ell^2 \left(1 + \displaystyle\sum_{j=\ell+1}^{d} M_{\ell,j} p_j^2\right)^{-1} \end{cases} \tag{162}$$

$$= \begin{cases} p_\ell^1 \\ p_\ell^2 \end{cases}, \tag{163}$$

where the second equality uses the induction hypothesis and the fact that $w_\ell^1 = w_\ell^2 = w_\ell$. This completes the inductive step. This result ensures that

$$p_k = (1 - w_i)p_k^1 + w_i p_k^2, \quad \text{for every } k \in [i+1..d]. \tag{164}$$

We complete the construction of the remaining coordinates of $w^1$ and $w^2$ by a backward induction. For $k = i$, we choose

$$w_i^1 = 0 \quad \text{and} \quad w_i^2 = 1. \tag{165}$$

Using Equation (160), we conclude that

$$p_i^1 = 0 \quad \text{and} \quad p_i^2 = \left(1 + \sum_{j=i+1}^{d} M_{i,j} p_j\right)^{-1}, \tag{166}$$

and furthermore,

$$p_i = w_i \left(1 + \sum_{j=i+1}^{d} M_{i,j} p_j\right)^{-1}. \tag{167}$$

This yields that our aimed property holds for $k = i$, i.e.,

$$p_i = (1 - w^i)p_i^1 + w^i p_i^2. \tag{168}$$

Now, suppose that $w_i^1, w_{i-1}^1, \ldots, w_k^1$ and $w_i^2, w_{i-1}^2, \ldots, w_k^2$ are constructed (in $[0\,,1]$) for some $k > 1$. We construct the $k-1$-th coordinates such that $w_{k-1}^1 = w_{k-1}^2$ and $p_{k-1} = (1-w_i)p_{k-1}^1 + w_i p_{k-1}^2$. This property is equivalent to

$$w_{k-1}\left(1 + \sum_{j=k}^{d} M_{i,j}p_j\right)^{-1} = \frac{(1-w_i)w_{k-1}^1}{\left(1 + \sum\limits_{j=k}^{d} M_{i,j}p_j^1\right)} + \frac{w_i w_{k-1}^2}{\left(1 + \sum\limits_{j=k}^{d} M_{i,j}p_j^2\right)}, \qquad (169)$$

it is sufficient to take

$$w_{k-1}^1 = w_{k-1}^2 = \frac{w_{k-1}\left(1 + \sum\limits_{j=k}^{d} M_{i,j}p_j\right)^{-1}}{\frac{(1-w_i)}{\left(1 + \sum\limits_{j=k}^{d} M_{i,j}p_j^1\right)} + \frac{w_i}{\left(1 + \sum\limits_{j=k}^{d} M_{i,j}p_j^2\right)}}, \qquad (170)$$

Using the backward induction hypothesis we have

$$\left(1 + \sum_{j=k}^{d} M_{i,j}p_j\right)^{-1} = \left((1-w_i)\left[1 + \sum_{j=k}^{d} M_{i,j}p_j^1\right] + w_i\left[1 + \sum_{j=k}^{d} M_{i,j}p_j^2\right]\right)^{-1}. \qquad (171)$$

Then, using Jensen's inequality (Lemma C.31) on the convex function $f : x \mapsto \frac{1}{x}$ using weights $(1-w_i, w_i)$ we obtain

$$\left(1 + \sum_{j=k}^{d} M_{i,j}p_j\right)^{-1} \overset{(171)}{=} \left((1-w_i)\left[1 + \sum_{j=k}^{d} M_{i,j}p_j^1\right] + w_i\left[1 + \sum_{j=k}^{d} M_{i,j}p_j^2\right]\right)^{-1} \qquad (172)$$

$$\overset{\text{Lem. C.31}}{\leq} (1-w_i)\left(1 + \sum_{j=k}^{d} M_{i,j}p_j^1\right)^{-1} + w_i\left(1 + \sum_{j=k}^{d} M_{i,j}p_j^2\right)^{-1}, \qquad (173)$$

besides $0 \leq w_{k-1} \leq 1$, thus $0 \leq w_{k-1}^1 = w_{k-1}^2 \leq w_{k-1} \leq 1$ and since $p^1 \neq p^2$ ($p_i^1 = 0$ and $p_i^2 \neq 0$), this concludes the proof. $\qquad \square$

Combining Lemmas F.1 and F.3, this achieves the proof of the theorem. $\qquad \square$

$$\begin{pmatrix} 0 & 0 & 0 & 0 \\ 0 & 0 & 0 & 0 \\ 0 & 0 & 0 & 0 \\ 0 & 0 & 0 & 0 \end{pmatrix} \quad \begin{pmatrix} 0 & 0 & 0 & 0 & 4 & 0 \\ 0 & 0 & 0 & 0 & 4 & 0 \\ 0 & 0 & 0 & 0 & 4 & 0 \\ 0 & 0 & 0 & 0 & 4 & 0 \\ 0 & 0 & 0 & 0 & 0 & 1 \\ 0 & 0 & 0 & 0 & 0 & 0 \end{pmatrix} \qquad \begin{pmatrix} 0 & 1 & 2 & 3 \\ 0 & 0 & 2 & 3 \\ 0 & 0 & 0 & 3 \\ 0 & 0 & 0 & 0 \end{pmatrix} \quad \begin{pmatrix} 0 & 1 & 2 & 3 & 4 & 0 \\ 0 & 0 & 2 & 3 & 4 & 4 \\ 0 & 0 & 0 & 3 & 4 & 4 \\ 0 & 0 & 0 & 0 & 4 & 4 \\ 0 & 0 & 0 & 0 & 0 & 4 \\ 0 & 0 & 0 & 0 & 0 & 0 \end{pmatrix}$$

(a) Case of a fast and a slow worker.      (b) Case of equally fast workers.

Figure 5: The matrices $M_3^\delta$ (left) and $M_5^\delta$ (right).

# G APPLICATION TO ASYNCHRONOUS (S)GD

In this part, we start by providing some examples of the "matrix of delays" as introduced in Section 5.2 and which arises during the convergence analysis of asynchronous gradient descent (AGD). This matrix, which we denote by $M^\delta$, consists of all the coefficients $M_{i,j}$ where[27] for $i, j \in [0 .. K]$ we have

$$M_{i,j} = \begin{cases} 0, & \text{if } j \notin M_i, \\ \delta^j, & \text{if } j \in M_i, \end{cases}$$

with, as we recall, the set $M_i$ is defined as

$$M_i := \left\{ j \in [0 .. K] \ : \ j - \delta^j \leq i \leq j - 1 \right\},$$

and $\left\{ \delta^j \right\}_{j \geq 0}$ is the sequence of delays while $K$ is the last iterations of AGD.

Then, for completeness, we not only provide the convergence analysis of AGD (Algorithm 1) but also of its stochastic counterpart, that is, asynchronous stochastic gradient descent ASGD (Algorithm 2) which will be enough to prove Theorem 5.4. The proof follows the analysis performed in Mishchenko et al. (2022); Koloskova et al. (2022); Maranjyan et al. (2025) while we make it more general by allowing arbitrary non-negative stepsizes $\{\gamma_k\}_{k \geq 0}$ in the gradient descent step (contrary to the original version where the stepsizes are assumed to be constant). Next, we refine the choice of the $\{\gamma_k\}_{k \geq 0}$ to the best possible choice. In addition to Algorithm 2 we also recall in Algorithm 3 the pseudo-code of the recently proposed Ringmaster ASGD algorithm (Maranjyan et al., 2025) which is the first asynchronous SGD method with provably optimal *time complexity*[28]. This new algorithm introduces a tunable threshold $R > 0$ on top of the original asynchronous SGD so as to discard the stale stochastic gradients which can be harmful for the global convergence of the method.

## G.1 A FEW TOY EXAMPLES

In the examples below, we provide a few *realistic* scenarios for the sequence of delays along with the associated matrix of delays $M^\delta$ for small value of $K$ (last iteration count). The examples mentioned below are relevant in real-world scenarios as they reflect on one hand, heterogeneity among the workers (different computation time, which is often witnessed in federated learning) but also, similarity among them to account for settings where the worker are equally fast.

*Example* G.1 (One Fast and One Slow Worker). Here we assume to have only $n = 2$ workers, one being very fast (say worker 1) while the other (worker 2) is slow. For instance, say worker 1 sent to the server the first 4 stochastic gradients while worker 2 sent the fifth one, then worker 1 sent the four next stochastic gradients and so on. This gives rise to the Table 1 below

Table 1: Illustration of which worker sends a gradient.

| Iteration number | 0 | 1 | 2 | 3 | 4 | 5 | 6 | 7 | 8 | 9 |
|---|---|---|---|---|---|---|---|---|---|---|
| Worker index | 1 | 1 | 1 | 1 | 2 | 1 | 1 | 1 | 1 | 2 |

---

[27] To align with the notation of Lemmas G.9 and G.10 and theorem G.11 and not to confuse the reader, we purposely tweak the indices of this matrix to start at 0 instead of 1.

[28] We do not expand on the time complexity framework (Tyurin & Richtárik, 2023; Tyurin & Richtárik, 2024; Tyurin, 2025) here, this framework will be slightly discussed in a subsequent paragraph.

which can be written concisely in the form $\mathcal{L}_W := [1, 1, 1, 1, 2, 1, 1, 1, 1, 2]$ after dropping the iteration number. Based on this we can construct the associated sequence of delays

$$\mathcal{L}_\delta := [0, 0, 0, 0, 4, 1, 0, 0, 0, 4],$$

since, by definition, if worker $i$ sends a stochastic gradient to the server at iteration $k \geq 0$ then, the delay associated to its worker in Algorithm 2 will be

$$\delta^k := k - \max\left\{r \in [1 \mathbin{..} k] \,:\, \mathcal{L}_W[r-1] = i\right\},$$

where we implicitly assume here that $\max \varnothing = 0$ (the lowest non-negative integer) in order to have $\delta^0 = 0$. We display above in Figure 5a the two matrices of delays $M_3^\delta$ and $M_5^\delta$ corresponding[29] to $\mathcal{L}_\delta$ for $K = 3$ and $K = 5$.

*Example* G.2 (Equally Fast Workers). In this paragraph, we assume to have $n = 5$ workers capable of working equally fast, i.e., the workers send their stochastic gradient one after the other in a periodic fashion (say, first worker 1, then worker 2, then worker 3, then 4, 5 and next worker 1 again and so on). We can represent this scenario as the list $\mathcal{L}_W := [1, 2, 3, 4, 5, 1, 2, 3, 4, 5]$ where we store the workers' index and the corresponding sequence of delays is

$$\mathcal{L}_\delta := [0, 1, 2, 3, 4, 4, 4, 4, 4, 4].$$

The matrices $M_3^\delta$ and $M_5^\delta$ corresponding to $\mathcal{L}_\delta$ for $K = 3$ and $K = 5$ are given in Figure 5b.

## G.2 ASSUMPTIONS

### G.2.1 ASSUMPTIONS FROM THE NONCONVEX WORLD

We recall below the assumptions satisfied by the function $f$ in the minimization problem (11) and the stochastic gradients $\nabla f(x, \xi)$; these assumptions are standard in the analysis of SGD-type methods in the nonconvex setting (Ghadimi & Lan, 2013; Bottou et al., 2018).

**Assumption G.3.** Function $f \colon \mathbb{R}^d \to \mathbb{R}$ is differentiable, and its gradients are $L$–Lipschitz continuous, i.e.,

$$\|\nabla f(x) - \nabla f(y)\| \leq L \|x - y\|, \ \forall x, y \in \mathbb{R}^d.$$

**Assumption G.4.** There exist $f^{\inf} \in \mathbb{R}$ such that $f(x) \geq f^{\inf}$ for all $x \in \mathbb{R}^d$.

Based on Assumption 5.2, we define the initial sub-optimality $\Delta := f(x^0) - f^{\inf}$, where $x^0$ is the starting point of optimization method.

**Assumption G.5.** The stochastic gradients $\nabla f(x; \xi)$ are unbiased and have bounded variance $\sigma^2 \geq 0$. Specifically,

$$\mathbb{E}_\xi \left[\nabla f(x; \xi)\right] = \nabla f(x), \ \forall x \in \mathbb{R}^d,$$
$$\mathbb{E}_\xi \left[\|\nabla f(x; \xi) - \nabla f(x)\|^2\right] \leq \sigma^2, \ \forall x \in \mathbb{R}^d.$$

The following assumption is also standard in the literature but rarely explicitly stated.

**Assumption G.6.** Let $x \in \mathbb{R}^d$ be a (possibly random) vector then, conditionally on $x$ the randomness $\xi$ in the stochastic gradient $\nabla f(x, \xi)$ is independent from all the past.

### G.2.2 ADDITIONAL ASSUMPTIONS

Throughout this part we consider the *universal computation model* introduced in Tyurin (2025). In this model, each worker can have arbitrary computation dynamic and such dynamic is characterized by a *computational power* function, as we recall below.

**Assumption G.7.** For any worker $i \in [n]$, its computational power function $v_i \colon \mathbb{R}_+ \to \mathbb{R}_+$ is non-negative and continuous almost everywhere.

---

[29]Note that the last iteration count is $K$ but the total number of iterations if $K + 1$.

Even though we do not derive time complexities (Tyurin & Richtárik, 2023; Tyurin, 2025; Maranjyan et al., 2025) in our convergence analysis, the universal computation model is important to keep in mind since it influences directly the sequence of delays $\left\{\delta^k\right\}_{k\geq 0}$.

Following Tyurin (2025), the number of stochastic gradients received by the server from worker $i \in [n]$ on some interval of time $[T_0, T_1]$ (with $0 \leq T_0 < T_1$) is either

$$\left\lfloor \int_{T_0}^{T_1} v_i(t)\,\mathrm{d}t \right\rfloor \quad \text{or} \quad 1 + \left\lfloor \int_{T_0}^{T_1} v_i(t)\,\mathrm{d}t \right\rfloor,$$

depending on if client $i$ was already computing a stochastic gradient before $T_0$ or not.

Additionally, so as to ensure our algorithms will never end prematurely due to the lack of computational power, e.g., for instance all workers crash suddenly and never get repaired, we also assume the following assumptions:

**Assumption G.8.** For any time $t \geq 0$, there exists some $i \in [n]$ and some $t' \geq t$ such that

$$\left\lfloor \int_{t}^{t'} v_i(\tau)\,\mathrm{d}\tau \right\rfloor \geq 1,$$

that is, if not stop the server will receive infinitely many stochastic gradients from the workers.

### G.3 ASYNCHRONOUS SGD ALGORITHMS

We consider asynchronous SGD (ASGD) whose pseudo-code is recalled below. We allow arbitrary non-negative stepsizes $\{\gamma_k\}_{k\geq 0}$ as of now. These stepsizes will be refined during the convergence analysis in Theorems G.11 and G.13.

In the three pseudo-codes below, Algorithm 2 and Procedure 2 have already been stated in Section 5 while Algorithm 3 is the pseudo-code of Ringmaster ASGD which will be discussed and analyzed in Appendices G.7 to G.9. While its convergence analysis is very similar to Algorithm 2, we show that actually Algorithm 3 is nothing else than a special case of Algorithm 2. Notably, Algorithm 3 relies on the sequence of *effective* delays $\{\widetilde{\delta}^k\}_{k\geq 0}$ which will play an important role as it allows to obtain refined convergence analysis of Algorithm 2, established in Theorem G.13. For clarity, we recall the definition of the *effective* delays $\{\widetilde{\delta}^k\}_{k\geq 0}$:

$$\widetilde{\delta}^k := \delta^k - \left|\left\{ j \in \left[k - \delta^k \,..\, k-1\right] \,:\, \gamma_j = 0 \right\}\right|.$$

---

**Algorithm 2:** Asynchronous SGD

1  **Initialization:**
2     $k \leftarrow 0$, the iteration counter
3     $x^0 \in \mathbb{R}^d$, the starting point
4     $\{\gamma_k\}_{k\geq 0}$, the stepsizes, $\gamma_k \geq 0$
5  Run **Procedure 2** in all workers
6  Send to all workers the point $x^0$
7  **while** true **do**
8     Wait until receiving $g_i^k := \nabla f\left(x^{k-\delta^k}; \xi_i^{k-\delta^k}\right)$ from worker $i$
9     $x^{k+1} \leftarrow x^k - \gamma_k g_i^k$
     // Reset the delay of worker $i$
10    Send to worker $i$ the point $x^{k+1}$
11    Update the iteration counter: $k \leftarrow k+1$

---

**Procedure 2:** Workers' (infinite) loop

1  **while** true **do**
2     Wait until receiving $x^k \in \mathbb{R}^d$ from the server
     // May take some time.
3     Compute a (stochastic) gradient $g \leftarrow \nabla f(x^k, \xi)$ where $\xi \sim \mathcal{D}$
4     Send $g$ to the server

---

---

**Algorithm 3:** Ringmaster ASGD

1 **Initialization:**
2     $k \leftarrow 0$, the iteration counter
3     $\ell \leftarrow 0$, the loop counter
4     $x^0 \in \mathbb{R}^d$, the starting point
5     $\gamma > 0$, the stepsize
6     $R > 0$, the delay threshold (to discard old gradients)
7 Run **Procedure 1** in all workers
8 Send to all workers the point $x^0$
9 **while** `true` **do**
    // Wait for some time...
10     Receive $g_i^\ell := \nabla f\left(x^{\ell-\delta^\ell}; \xi_i^{\ell-\delta^\ell}\right)$ from worker $i$
    // If the gradient is not too old.
11     **if** $\widetilde{\delta}^\ell < R$ **then**
       // Do one descent step.
12        $x^{k+1} \leftarrow x^k - \gamma g_i^\ell$
13        Update the iteration counter: $k \leftarrow k + 1$
14     **else**
15        Ignore the stochastic gradient $g_i^\ell$
    // Reset the delay of worker $i$
16     Send to worker $i$ the point $x^k$
17     Update the loop counter: $\ell \leftarrow \ell + 1$

---

Let us show how Ringmaster ASGD (Algorithm 3) can be seen a a special case of the general Algorithm 2. In Ringmaster ASGD the stochastic gradients whose *effective* delays $\widetilde{\delta}^\ell$ are smaller than the threshold $R$ are accepted and contribute to the optimization process, in other word, during the $\ell^{\text{th}}$ loop, the stepsize $\gamma_\ell^{(R)}$ used by Ringmaster ASGD is

$$\gamma_\ell^{(R)} := \gamma \,\mathbb{I}\left\{\widetilde{\delta}^\ell < R\right\},$$

where $\gamma := \min\left\{\frac{1}{2LR}, \frac{\varepsilon}{4L\sigma^2}\right\}$ is provided in Maranjyan et al. (2025, Theorem 4.1). Here $\mathbb{I}\{\cdot\}$ denotes the indicator function. Hence, a tight analysis of the general asynchronous SGD algorithm provided in Algorithm 2 would allow one to recover the convergence rate of Ringmaster SGD; this is what we show in Theorem G.14.

## G.4 A DESCENT LEMMA

The next descent lemma is adapted from (Maranjyan et al., 2025, Lemma C.1).

**Lemma G.9** (A Descent Lemma). *Under Assumptions G.3, G.5 and G.8[30], for any choice of non-negative stepsizes $\{\gamma_k\}_{k\geq 0}$ in* ASGD *(Algorithm 2), the inequality*

$$\begin{aligned}
\mathbb{E}_{k+1}\left[f(x^{k+1})\right] \leq &f(x^k) - \frac{\gamma_k}{2}\left\|\nabla f\left(x^k\right)\right\|^2 \\
&- \frac{\gamma_k}{2}(1 - \gamma_k L)\left\|\nabla f\left(x^{k-\delta^k}\right)\right\|^2 \\
&+ \frac{\gamma_k L^2}{2}\left\|x^k - x^{k-\delta^k}\right\|^2 + \frac{\gamma_k^2 L}{2}\sigma^2,
\end{aligned}$$

*holds, where $\mathbb{E}_{k+1}[\cdot]$ represents the expectation conditioned on all randomness up to iteration $k$.*

*Proof.* Assume, that we get a stochastic gradient from the worker with index $i_k$ when calculating $x^{k+1}$. Since the function $f$ has $L$–Lipschitz gradients according to Assumption 5.1, it is $L$-smooth and we have (Nesterov, 2018):

$$\begin{aligned}
\mathbb{E}_{k+1}\left[f\left(x^{k+1}\right)\right] \overset{\text{Lem. C.25}}{\leq} &f\left(x^k\right) - \gamma_k \underbrace{\mathbb{E}_{k+1}\left[\left\langle\nabla f\left(x^k\right) \mid \nabla f\left(x^{k-\delta^k}, \xi_{i_k}^{k-\delta^k}\right)\right\rangle\right]}_{=:t_1} \\
&+ \frac{L}{2}\gamma_k^2 \underbrace{\mathbb{E}_{k+1}\left[\left\|\nabla f\left(x^{k-\delta^k}, \xi_{i_k}^{k-\delta^k}\right)\right\|^2\right]}_{=:t_2},
\end{aligned}$$

---

[30]This assumption serves only to ensure that the $(k + 1)$-th iteration is well-defined and the iterate $x^{k+1}$ exists. Assumption G.8 is enough to ensure this property, so that the iterate $x^k$ always exists for any $k \geq 0$.

which comes from upper bounding the Bregman divergence of $f$ at $x^{k+1} = x^k - \gamma_k \nabla f\left(x^{k-\delta^k}, \xi_{i_k}^{k-\delta^k}\right)$ and $x^k$. Then, using the unbiasedness of the stochastic gradients from Assumption G.5, we estimate the first term $t_1$ as

$$t_1 \stackrel{\text{Ass. G.5}}{=} \left\langle \nabla f\left(x^k\right), \nabla f\left(x^{k-\delta^k}\right)\right\rangle$$

$$\stackrel{(26)}{=} \frac{1}{2}\left(\left\|\nabla f\left(x^k\right)\right\|^2 + \left\|\nabla f\left(x^{k-\delta^k}\right)\right\|^2 - \left\|\nabla f\left(x^k\right) - \nabla f\left(x^{k-\delta^k}\right)\right\|^2\right), \qquad (174)$$

and for the second term $t_2$, we use the variance decomposition (Lemma C.26) and Assumption G.5, we get

$$t_2 \stackrel{\text{Lem. C.26}}{=} \mathbb{E}_{k+1}\left[\left\|\nabla f\left(x^{k-\delta^k}, \xi_{i_k}^{k-\delta^k}\right) - \nabla f\left(x^{k-\delta^k}\right)\right\|^2\right] + \left\|\nabla f\left(x^{k-\delta^k}\right)\right\|^2$$

$$\stackrel{\text{Ass. G.5}}{\leq} \sigma^2 + \left\|\nabla f\left(x^{k-\delta^k}\right)\right\|^2. \qquad (175)$$

Now, combining the results for both terms $t_1$ and $t_2$, and using the $L$–Lipchitz gradients property of $f$ to bound the squared norm $\|\nabla f\left(x^k\right) - \nabla f(x^{k-\delta^k})\|^2$, we obtain the inequality

$$\mathbb{E}_{k+1}\left[f\left(x^{k+1}\right)\right] \stackrel{(174)+(175)}{\leq} f\left(x^k\right) - \frac{\gamma_k}{2}\left\|\nabla f\left(x^k\right)\right\|^2$$

$$- \frac{\gamma_k}{2}(1 - \gamma_k L)\left\|\nabla f\left(x^{k-\delta^k}\right)\right\|^2$$

$$+ \frac{\gamma_k L^2}{2}\left\|x^k - x^{k-\delta^k}\right\|^2 + \frac{\gamma_k^2 L}{2}\sigma^2,$$

which is what we wanted to prove. $\qquad\square$

## G.5 RESIDUAL ESTIMATION (A FIRST VERSION)

**Lemma G.10** (Residual Estimation). *Under Assumptions G.3, G.5, G.6 and G.8, for any integer $k \geq 0$ and any choice of non-negative stepsizes $\{\gamma_j\}_{j\geq 0}$, the iterates $\{x^j\}_{j\geq 0}$ of* ASGD *(algorithm 2) satisfy*

$$\mathbb{E}\left[\left\|x^k - x^{k-\delta^k}\right\|^2\right] \leq 2\delta^k \sum_{j=k-\delta^k}^{k-1}\gamma_j^2 \mathbb{E}\left[\left\|\nabla f\left(x^{j-\delta^j}\right)\right\|^2\right] + 2\sigma^2 \sum_{j=k-\delta^k}^{k-1}\gamma_j^2.$$

*Proof.* Assume that for any $j \in [0 .. k]$, we receive a stochastic gradient from the worker with index $i_j \in [n]$ when calculating $x^j$. Then, to upper bound the residual $x^k - x^{k-\delta^k}$, we begin by expanding the difference between the two points to obtain[31]

$$x^k - x^{k-\delta^k} = \sum_{j=k-\delta^k}^{k-1}\gamma_j \nabla f\left(x^{j-\delta^j}, \xi_{i_j}^{j-\delta^j}\right), \qquad (176)$$

and now, according to the tower property of expectation (Lemma C.27) and Assumption G.5 we have

$$\mathbb{E}\left[\sum_{j=k-\delta^k}^{k-1}\gamma_j \nabla f\left(x^{j-\delta^j}, \xi_{i_j}^{j-\delta^j}\right)\right] \stackrel{\text{Lem. C.27}}{=} \sum_{j=k-\delta^k}^{k-1}\gamma_j \mathbb{E}\left[\mathbb{E}\left[\nabla f\left(x^{j-\delta^j}, \xi_{i_j}^{j-\delta^j}\right) \mid x^{j-\delta^j}\right]\right] \quad (177)$$

$$\stackrel{\text{Ass. G.5}}{=} \mathbb{E}\left[\sum_{j=k-\delta^k}^{k-1}\gamma_j \nabla f\left(x^{j-\delta^j}\right)\right]. \qquad (178)$$

Now, as notced in Mishchenko et al. (2022), we cannot apply directly the variance decomposition (Lemma C.26) as the asynchronicity causes certain stochastic gradients to depend on each other.

---

[31]See, for instance **lemma 1** of Mishchenko et al. (2022).

Instead, we first apply Young's inequality (Lemma C.29) to the sum of random variables in (176) which gives

$$
\mathbb{E}\left[\left\|x^k - x^{k-\delta^k}\right\|^2\right] \overset{(176)}{=} \mathbb{E}\left[\left\|\sum_{j=k-\delta^k}^{k-1} \gamma_j \nabla f\left(x^{j-\delta^j}, \xi_{i_j}^{j-\delta^j}\right)\right\|^2\right]
$$

$$
\overset{\text{Lem. C.26}}{=} \mathbb{E}\left[\left\|\sum_{j=k-\delta^k}^{k-1} \gamma_j\left[\nabla f\left(x^{j-\delta^j}, \xi_{i_j}^{j-\delta^j}\right) - \nabla f\left(x^{j-\delta^j}\right)\right] + \sum_{j=k-\delta^k}^{k-1} \gamma_j \nabla f\left(x^{j-\delta^j}\right)\right\|^2\right]
$$

$$
\overset{\text{Lem. C.29}}{\leq} 2\,\mathbb{E}\left[\left\|\sum_{j=k-\delta^k}^{k-1} \gamma_j\left[\nabla f\left(x^{j-\delta^j}, \xi_{i_j}^{j-\delta^j}\right) - \nabla f\left(x^{j-\delta^j}\right)\right]\right\|^2\right]
$$

$$
+ 2\,\mathbb{E}\left[\left\|\sum_{j=k-\delta^k}^{k-1} \gamma_j \nabla f\left(x^{j-\delta^j}\right)\right\|^2\right] \tag{179}
$$

Moreover, thanks to Assumption G.6 and unbiasedness from Assumption G.5, when conditioned on the random points $x^0, \ldots, x^k$ the stochastic gradients

$$
\nabla f\left(x^{j-\delta^j}; \xi_{i_j}^{j-\delta^j}\right),
$$

for $k - \delta^k \leq j \leq k - 1$ are pairwise independent and we can apply Lemma C.30 in the first term of (179) with the conditional expectation over $x^0, \ldots, x^k$. First, we apply the tower property (Lemma C.27) to get

$$
2\,\mathbb{E}\left[\left\|\sum_{j=k-\delta^k}^{k-1} \gamma_j\left[\nabla f\left(x^{j-\delta^j}, \xi_{i_j}^{j-\delta^j}\right) - \nabla f\left(x^{j-\delta^j}\right)\right]\right\|^2\right]
$$

$$
\overset{\text{Lem. C.27}}{=} 2\,\mathbb{E}\left[\mathbb{E}\left[\left\|\sum_{j=k-\delta^k}^{k-1} \gamma_j\left[\nabla f\left(x^{j-\delta^j}, \xi_{i_j}^{j-\delta^j}\right) - \nabla f\left(x^{j-\delta^j}\right)\right]\right\|^2 \,\middle|\, x^0, \ldots, x^k\right]\right] \tag{180}
$$

$$
\overset{\text{Lem. C.30}}{=} 2\,\mathbb{E}\left[\sum_{j=k-\delta^k}^{k-1} \gamma_j^2\,\mathbb{E}\left[\left\|\nabla f\left(x^{j-\delta^j}, \xi_{i_j}^{j-\delta^j}\right) - \nabla f\left(x^{j-\delta^j}\right)\right\|^2 \,\middle|\, x^0, \ldots, x^k\right]\right]
$$

$$
= 2\sum_{j=k-\delta^k}^{k-1} \gamma_j^2\,\mathbb{E}\left[\left\|\nabla f\left(x^{j-\delta^j}, \xi_{i_j}^{j-\delta^j}\right) - \nabla f\left(x^{j-\delta^j}\right)\right\|^2\right],
$$

and since all the stochastic gradient considered are $\sigma^2$–variance bounded by Assumption G.5 then we can further upper bound the sum (180) by

$$
2\sum_{j=k-\delta^k}^{k-1} \gamma_j^2\,\mathbb{E}\left[\left\|\nabla f\left(x^{j-\delta^j}, \xi_{i_j}^{j-\delta^j}\right) - \nabla f\left(x^{j-\delta^j}\right)\right\|^2\right] \overset{\text{Ass. G.5}}{\leq} 2\sigma^2 \sum_{j=k-\delta^k}^{k-1} \gamma_j^2. \tag{181}
$$

Then to deal with the second term of (179), we apply Jensen's inequality in the form of Lemma C.32 to obtain

$$
\left\|\sum_{j=k-\delta^k}^{k-1} \gamma_j \nabla f\left(x^{j-\delta^j}\right)\right\|^2 \overset{\text{Lem. C.32}}{\leq} \delta^k \sum_{j=k-\delta^k}^{k-1} \gamma_j^2 \left\|\nabla f\left(x^{j-\delta^j}\right)\right\|^2, \tag{182}
$$

and finally, taking expectation inside the inequality (182) and, combining the upper bounds (181) and (182) on both terms of (179) respectively gives

$$
\mathbb{E}\left[\left\|x^k - x^{k-\delta^k}\right\|^2\right] \overset{(180)+(182)}{\leq} 2\sigma^2 \sum_{j=k-\delta^k}^{k-1} \gamma_j^2 + 2\delta^k \sum_{j=k-\delta^k}^{k-1} \gamma_j^2\,\mathbb{E}\left[\left\|\nabla f\left(x^{j-\delta^j}\right)\right\|^2\right],
$$

which achieves the proof of this lemma. □

## G.6 CONVERGENCE ANALYSIS OF ALGORITHM 2

**Theorem G.11** (Convergence Analysis of Algorithm 2). *Under Assumptions G.3 to G.6 and G.8, for any integer $K \geq 0$ and any choice of non-negative stepsizes $\{\gamma_k\}_{k \geq 0}$ such that there exists $k \in [0 .. K]$ for which $\gamma_k > 0$, the iterates $\{x^k\}_{k \geq 0}$ of* ASGD *(Algorithm 2) satisfy, with $\Gamma_K := \gamma_0 + \cdots + \gamma_K > 0$*

$$\frac{1}{\Gamma_K} \sum_{k=0}^{K} \gamma_k \, \mathbb{E}\left[\left\|\nabla f\left(x^k\right)\right\|^2\right] \leq \frac{2\Delta}{\Gamma_K} + R(K) + \frac{L\sigma^2}{\Gamma_K} \sum_{k=0}^{K} \gamma_k^2 \left(1 + 2L \sum_{j \in M_k} \gamma_j\right), \quad (183)$$

*where $R(K) := \frac{1}{\Gamma_K} \sum\limits_{k=0}^{K} R_k \gamma_k \, \mathbb{E}\left[\left\|\nabla f\left(x^{k-\delta^k}\right)\right\|^2\right]$,*

$$R_k := \gamma_k L + 2\gamma_k L^2 \sum_{j \in M_k} \gamma_j \, \delta^j - 1,$$

*and the sets $M_k$ for $k \in [0 .. K]$ are defined as*

$$M_k := \left\{j \in [0 .. K] \,:\, j - \delta^j \leq k \leq j - 1\right\}. \quad (184)$$

*Proof.* According to Lemma G.9, under the above assumptions for any $k \in [0 .. K]$ we have

$$\mathbb{E}_{k+1}\left[f\left(x^{k+1}\right)\right] \overset{\text{Lem. G.9}}{\leq} f\left(x^k\right) - \frac{\gamma_k}{2}\left\|\nabla f\left(x^k\right)\right\|^2$$
$$- \frac{\gamma_k}{2}(1 - \gamma_k L)\left\|\nabla f\left(x^{k-\delta^k}\right)\right\|^2$$
$$+ \frac{\gamma_k L^2}{2}\left\|x^k - x^{k-\delta^k}\right\|^2 + \frac{\gamma_k^2 L}{2}\sigma^2,$$

hence, taking expectation on both sides and using the tower property gives

$$\mathbb{E}\left[\mathbb{E}_{k+1}\left[f(x^{k+1})\right]\right] \overset{\text{Lem. C.27}}{=} \mathbb{E}\left[f(x^{k+1})\right] \quad (185)$$
$$\overset{\text{Lem. G.9}}{\leq} \mathbb{E}\left[f(x^k)\right] - \frac{\gamma_k}{2}\mathbb{E}\left[\left\|\nabla f\left(x^k\right)\right\|^2\right] \quad (186)$$
$$- \frac{\gamma_k}{2}(1 - \gamma_k L)\mathbb{E}\left[\left\|\nabla f\left(x^{k-\delta^k}\right)\right\|^2\right]$$
$$+ \frac{\gamma_k L^2}{2}\mathbb{E}\left[\left\|x^k - x^{k-\delta^k}\right\|^2\right] + \frac{\gamma_k^2 L}{2}\sigma^2,$$

and reshuffling the above inequality yields

$$\frac{\gamma_k}{2}\mathbb{E}\left[\left\|\nabla f\left(x^k\right)\right\|^2\right] \leq \underbrace{\left(\mathbb{E}\left[f(x^k)\right] - \mathbb{E}\left[f(x^{k+1})\right]\right)}_{:= A_k^{(1)}} \quad (187)$$
$$- \frac{\gamma_k}{2}(1 - \gamma_k L)\mathbb{E}\left[\left\|\nabla f\left(x^{k-\delta^k}\right)\right\|^2\right] \quad (188)$$
$$+ \underbrace{\frac{\gamma_k L^2}{2}\mathbb{E}\left[\left\|x^k - x^{k-\delta^k}\right\|^2\right]}_{:= A_k^{(2)}} + \frac{\gamma_k^2 L}{2}\sigma^2. \quad (189)$$

Now, if we sum the above inequality (189) over all $k \in [0 .. K]$, the sum of all $A_k^{(1)}$ terms can be telescoped, i.e.,

$$
\begin{aligned}
\sum_{k=0}^{K} A_k^{(1)} &= \sum_{k=0}^{K} \left( \mathbb{E}\left[ f(x^k) \right] - \mathbb{E}\left[ f(x^{k+1}) \right] \right) \\
&= \mathbb{E}\left[ f(x^0) - f(x^{K+1}) \right] \\
&\overset{\text{Ass. 5.2}}{\leq} \mathbb{E}\left[ f(x^0) - f^{\text{inf}} \right] \\
&= \Delta,
\end{aligned}
$$

while for the residual term $A_k^{(2)}$ we upper bound it using Lemma G.10 since for any $k \in [0 .. K]$ the quantity $\gamma_k L^2 / 2$ is non-negative. This gives the upper bound

$$
\begin{aligned}
\sum_{k=0}^{K} A_k^{(2)} &= \sum_{k=0}^{K} \frac{\gamma_k L^2}{2} \mathbb{E}\left[ \left\| x^k - x^{k-\delta^k} \right\|^2 \right] \\
&\overset{\text{Lem. G.10}}{\leq} \sum_{k=0}^{K} \frac{\gamma_k L^2}{2} \left[ 2\delta^k \sum_{j=k-\delta^k}^{k-1} \gamma_j^2 \mathbb{E}\left[ \left\| \nabla f\left( x^{j-\delta^j} \right) \right\|^2 \right] + 2\sigma^2 \sum_{j=k-\delta^k}^{k-1} \gamma_j^2 \right] \\
&\overset{(a)}{=} \underbrace{L^2 \sum_{k=0}^{K} \sum_{j=k-\delta^k}^{k-1} \gamma_k \, \delta^k \gamma_j^2 \mathbb{E}\left[ \left\| \nabla f\left( x^{j-\delta^j} \right) \right\|^2 \right]}_{:= B_1} + \underbrace{L^2 \sigma^2 \sum_{k=0}^{K} \sum_{j=k-\delta^k}^{k-1} \gamma_k \gamma_j^2}_{:= B_2},
\end{aligned}
$$

where in (a) we expand the outer sum.

Then, we reshuffle both sums $B_1$ and $B_2$ by exchanging the indices $k$ and $j$ of the two nested sums. To do so, we use Lemma C.34 with $S = [0 .. K]$ and for any $k \in S$, we have $S(k) = \left[ k - \delta^k .. k - 1 \right] \subseteq [0 .. K]$ so we choose $S' = [0 .. K]$ so that it contains every $S(k)$ and now for every $j \in S'$ we have

$$
\begin{aligned}
S'(j) &\overset{\text{Lem. C.34}}{=} \left\{ k \in [0 .. K] : j \in S(k) \right\} \\
&= \left\{ k \in [0 .. K] : k - \delta^k \leq j \leq k - 1 \right\} \\
&\overset{(184)}{=} M_j,
\end{aligned}
$$

thus we can rewrite the term $B_1$ as

$$
B_1 = \sum_{j=0}^{K} \sum_{k \in M_j} \gamma_k \, \delta^k \gamma_j^2 \mathbb{E}\left[ \left\| \nabla f\left( x^{j-\delta^j} \right) \right\|^2 \right], \tag{190}
$$

and the term $B_2$ can we rewritten as

$$
B_2 = \sum_{j=0}^{K} \sum_{k \in M_j} \gamma_k \gamma_j^2. \tag{191}
$$

Now, plugging both (190) and (191) in inequality (189) after summing over $k \in [0 .. K]$ leads to

$$\frac{1}{2} \sum_{k=0}^{K} \gamma_k \, \mathbb{E} \left[ \left\| \nabla f \left( x^k \right) \right\|^2 \right]$$

$$\leq \Delta - \frac{1}{2} \sum_{k=0}^{K} \gamma_k (1 - \gamma_k L) \, \mathbb{E} \left[ \left\| \nabla f \left( x^{k-\delta^k} \right) \right\|^2 \right] + L^2 \sum_{j=0}^{K} \sum_{k \in M_j} \gamma_k \, \delta^k \gamma_j^2 \, \mathbb{E} \left[ \left\| \nabla f \left( x^{j-\delta^j} \right) \right\|^2 \right]$$

$$+ L^2 \sigma^2 \sum_{j=0}^{K} \sum_{k \in M_j} \gamma_k \gamma_j^2 + \frac{L\sigma^2}{2} \sum_{k=0}^{K} \gamma_k^2$$

$$\overset{\text{(a)}}{=} \Delta - \frac{1}{2} \sum_{k=0}^{K} \gamma_k (1 - \gamma_k L) \, \mathbb{E} \left[ \left\| \nabla f \left( x^{k-\delta^k} \right) \right\|^2 \right] + L^2 \sum_{k=0}^{K} \sum_{j \in M_k} \gamma_j \, \delta^j \gamma_k^2 \, \mathbb{E} \left[ \left\| \nabla f \left( x^{k-\delta^k} \right) \right\|^2 \right]$$

$$+ L^2 \sigma^2 \sum_{k=0}^{K} \sum_{j \in M_k} \gamma_j \gamma_k^2 + \frac{L\sigma^2}{2} \sum_{k=0}^{K} \gamma_k^2$$

$$\overset{\text{(b)}}{=} \Delta + \frac{1}{2} \sum_{k=0}^{K} \gamma_k \, \mathbb{E} \left[ \left\| \nabla f \left( x^{k-\delta^k} \right) \right\|^2 \right] \left[ L\gamma_k \left( 1 + 2L \sum_{j \in M_k} \gamma_j \, \delta^j \right) - 1 \right] \tag{192}$$

$$+ \frac{L\sigma^2}{2} \sum_{k=0}^{K} \gamma_k^2 \left( 1 + 2L \sum_{j \in M_k} \gamma_j \gamma_k^2 \right)$$

where in (a) we permute the labels of the indices of the second and third sum (those involving the sets $\{M_j\}_{j \in [0 .. K]}$), i.e. $j \leftrightarrow k$, while in (b) we merge the first two sums involving the gradients $\nabla f(\cdot)$ and the last two sums involving the stochastic term in $\sigma^2$. More precisely, for the "gradient terms", the resulting $k$-th term for $k \in [0 .. K]$ reads

$$-\gamma_k (1 - \gamma_k L) + 2 \gamma_k^2 L^2 \left( \sum_{j \in M_k} \gamma_j \, \delta^j \right) \tag{193}$$

$$= \gamma_k \left[ \gamma_k L + 2 \gamma_k L^2 \left( \sum_{j \in M_k} \gamma_j \, \delta^j \right) - 1 \right] \tag{194}$$

$$= \gamma_k \left[ L\gamma_k \left( 1 + 2L \sum_{j \in M_k} \gamma_j \, \delta^j \right) - 1 \right], \tag{195}$$

while for the "stochastic terms", the $k$- term reads

$$2 \gamma_k^2 L^2 \left( \sum_{j \in M_k} \gamma_j \right) + \gamma_k^2 L = \gamma_k^2 L \left( 1 + 2L \sum_{j \in M_k} \gamma_j \right).$$

Now, multiplying (192) by two and dividing both sides of the inequality by $\gamma_0 + \gamma_1 + \cdots + \gamma_K > 0$[32] leads to

$$\frac{1}{\sum_{k=0}^{K} \gamma_k} \sum_{k=0}^{K} \gamma_k \, \mathbb{E} \left[ \left\| \nabla f \left( x^k \right) \right\|^2 \right] \leq \frac{2\Delta}{\sum_{k=0}^{K} \gamma_k} + R(K) + L\sigma^2 \frac{\sum_{k=0}^{K} \gamma_k^2 \left( 1 + 2L \sum_{j \in M_k} \gamma_j \right)}{\sum_{k=0}^{K} \gamma_k},$$

---

[32]Recall that in statement of Theorem G.11 where we assume there exists $k \in [0 .. K]$ such that $\gamma_k > 0$ ensuring the division to be legal.

where we define

$$R(K) := \frac{1}{\sum\limits_{k=0}^{K} \gamma_k} \sum_{k=0}^{K} \gamma_k \, \mathbb{E}\left[\left\|\nabla f\left(x^{k-\delta^k}\right)\right\|^2\right] \left[\gamma_k L \left(1 + 2L \sum_{j \in M_k} \gamma_j \, \delta^j\right) - 1\right],$$

which achieves the proof the the theorem. □

In the case where $\sigma^2 = 0$, we recover Assumption 5.3 and Theorem G.11 reduces to Theorem 5.4 which we recall here for completeness.

**Theorem 5.4.** *Under Assumptions 5.1 to 5.3, for any integer $K \geq 0$ and any choice of non-negative stepsizes $\{\gamma_k\}_{k \geq 0}$ the iterates $\{x^k\}_{k \geq 0}$ of* AGD *(Algorithm 1) satisfy, with $\Gamma_K := \gamma_0 + \cdots + \gamma_K$*

$$\frac{1}{\Gamma_K} \sum_{k=0}^{K} \gamma_k \, \mathbb{E}\left[\left\|\nabla f\left(x^k\right)\right\|^2\right] \leq \frac{2\Delta}{\Gamma_K} + R(K), \tag{196}$$

*where $R(K) := \frac{1}{\Gamma_K} \sum\limits_{k=0}^{K} R_k \gamma_k \, \mathbb{E}\left[\left\|\nabla f\left(x^{k-\delta^k}\right)\right\|^2\right]$,*

$$R_k := \gamma_k L + 2\gamma_k L^2 \sum_{j \in M_k} \gamma_j \, \delta^j - 1,$$

*and $M_k := \{r \in [0 \ldots K] : r - \delta^r \leq k \leq r - 1\}$.*

*Proof.* Setting $\sigma^2 = 0$ in the left-hand side of (183) immediately gives (196), as desired. □

## G.7 IMPROVING THE CONVERGENCE ANALYSIS

As observed in Section 5.2, the sequence of delays $\{\delta^k\}_{k \geq 0}$ is not influenced at all by how we choose the stepsizes $\{\gamma_k\}_{k \geq 0}$ which is unreasonable since only the accepted gradients (corresponding to a positive stepsize) contribute to the optimization process. So the discarded gradients should not impact the choice of the stepsizes but, considering the matrix of delay $M^\delta$ and the associated optimization problem $(\mathscr{P}_d)$, this is not the case since the delays $\delta^k$ corresponding to a stepsize $\gamma_k > 0$ also counts some of the discarded gradients. This may results in smaller stepsizes when solving the corresponding optimization problem.

Hence naturally, (e.g., as in Ringmaster ASGD) it seems much more relevant for the delay $\delta^k$ to account for the total number of *accepted* gradients. To this end, we introduce a new sequence of delays $\{\widetilde{\delta}^k\}_{k \geq 0}$ where which will count, among all stochastic gradients received by the server on some interval, precisely those which have been accepted. This result in the following definition: for any integer $k \geq 0$

$$\widetilde{\delta}^k := \delta^k - \left|\left\{j \in \left[k - \delta^k \ldots k - 1\right] : \gamma_j = 0\right\}\right|, \tag{197}$$

where we assume that $\max \varnothing = 0$ (so that $\widetilde{\delta}^0 = 0$). Notably, we have $\widetilde{\delta}^\ell \leq \delta^\ell$ for all integer $\ell \geq 0$.

In the next two parts (Appendices G.8 and G.9) we improve the residual estimation using the sequence $\{\widetilde{\delta}^\ell\}_{\ell \geq 0}$ and state the new convergence rate obtained. As a byproduct of our general analysis, we also recover the convergence rate of Ringmaster ASGD (Maranjyan et al., 2025) in Theorem G.14. The improvement stems from the application of Jensen's inequality (Lemma C.31) in (182). Following most state-of-the-art analysis of asynchronous methods, we also apply Jensen's inequality to bound the staleness error. While these analysis rely on the special case stated in Lemma C.32, so as to tighten our bounds we apply the "refined" inequality in Remark C.33: since some of the stepsizes can be zero, we can apply the inequality Lemma C.32 only on the positive terms rather than all of them. This strengthening is crucial to recover the rate of Ringmaster ASGD (see Theorem G.14).

## G.8 Residual Estimation (a Refined Version)

While the descent lemma proved in Appendix G.4 is still the same, the residual estimation in Appendix G.5 can be improved using the new sequence of delays $\{\widetilde{\delta}^\ell\}_{\ell \geq 0}$ which is the purpose of the following lemma.

**Lemma G.12** (Residual Estimation: A Refined Version). *Under Assumptions G.3, G.5, G.6 and G.8, for any integer $k \geq 0$ and any choice of non-negative stepsizes $\{\gamma_j\}_{j \geq 0}$, the iterates $\{x^j\}_{j \geq 0}$ of* ASGD *(Algorithm 2) satisfy*

$$\mathbb{E}\left[\left\|x^k - x^{k-\delta^k}\right\|^2\right] \leq 2\widetilde{\delta}^k \sum_{j=k-\delta^k}^{k-1} \gamma_j^2 \, \mathbb{E}\left[\left\|\nabla f\left(x^{j-\delta^j}\right)\right\|^2\right] + 2\sigma^2 \sum_{j=k-\delta^k}^{k-1} \gamma_j^2.$$

*where the sequence $\{\widetilde{\delta}^k\}_{k \geq 0}$ is defined in (197).*

*Proof.* We follows exactly the same steps as in the proof of Lemma G.10 with the sole exception that in (182) instead of using Jensen's inequality in the form of Lemma C.32, we use Remark C.33 to obtain the upper bound

$$\left\|\sum_{j=k-\delta^k}^{k-1} \gamma_j \nabla f\left(x^{j-\delta^j}\right)\right\|^2 \overset{\text{Rem. C.33}}{\leq} \widetilde{\delta}^k \sum_{j=k-\delta^k}^{k-1} \gamma_j^2 \left\|\nabla f\left(x^{j-\delta^j}\right)\right\|^2, \tag{198}$$

for all integer $k \geq 0$. We then combine the tighter upper bound (198) with the other bound in (181) to obtain that, for any $k \geq 0$ we have

$$\mathbb{E}\left[\left\|x^k - x^{k-\delta^k}\right\|^2\right] \leq 2\widetilde{\delta}^k \sum_{j=k-\delta^k}^{k-1} \gamma_j^2 \, \mathbb{E}\left[\left\|\nabla f\left(x^{j-\delta^j}\right)\right\|^2\right] + 2\sigma^2 \sum_{j=k-\delta^k}^{k-1} \gamma_j^2,$$

which achieves the proof of the lemma. $\qquad\square$

## G.9 Convergence Analysis of Algorithm 3

**Improving the Convergence Analysis.** Equipped with the improved residual estimation in Lemma G.12, we can now state our main result for the convergence analysis of ASGD in full generality.

**Theorem G.13.** *Under Assumptions G.3 to G.6 and G.8, for any integer $K \geq 0$ and any choice of non-negative stepsizes $\{\gamma_k\}_{k \geq 0}$ such that there exists $k \in [0 \,..\, K]$ for which $\gamma_k > 0$, the iterates $\{x^k\}_{k \geq 0}$ of* ASGD *(Algorithm 2) satisfy, with $\Gamma_K := \gamma_0 + \cdots + \gamma_K > 0$*

$$\frac{1}{\Gamma_K} \sum_{k=0}^{K} \gamma_k \, \mathbb{E}\left[\left\|\nabla f\left(x^k\right)\right\|^2\right] \leq \frac{2\Delta}{\Gamma_K} + \widetilde{R}(K) + \frac{L\sigma^2}{\Gamma_K} \sum_{k=0}^{K} \gamma_k^2 \left(1 + 2L \sum_{j \in M_k} \gamma_j\right), \tag{199}$$

*where $\widetilde{R}(K) := \frac{1}{\Gamma_K} \sum\limits_{k=0}^{K} \widetilde{R}_k \gamma_k \, \mathbb{E}\left[\left\|\nabla f\left(x^{k-\delta^k}\right)\right\|^2\right]$, with*

$$\widetilde{R}_k := \gamma_k L + 2\gamma_k L^2 \sum_{j \in M_k} \gamma_j \, \widetilde{\delta}^j - 1,$$

*and the sets $M_k$ and delays $\widetilde{\delta}^k$ are defined in (184) and (197).*

*Proof.* The proof is a straightforward adaption of the previous proof of Theorem G.11 where instead of the residual estimation from Lemma G.10 we use the its sharper version Lemma G.12. $\qquad\square$

**Recovering the convergence rate of** Ringmaster ASGD. Now using the improved upper bound from Theorem G.13, we can recover the same rate as in the paper Maranjyan et al. (2025), which is the purpose of the next theorem. Moreover, our proof is more transparent than the one in Maranjyan et al. (2025) as in our proof we capture all stochastic gradients received by the server and not just the gradients which are accepted.

**Theorem G.14** (Recovering Ringmaster ASGD Convergence Rate). *Let $R \geq 1$ be the delay threshold of* Ringmaster ASGD *(Maranjyan et al., 2025) then, under Assumptions G.3 to G.6 and G.8, if we let the stepsizes of* ASGD *(Algorithm 3) be*

$$\gamma_k = \gamma \, \mathbb{I}\left\{\widetilde{\delta}^k < R\right\}, \quad \text{with} \quad \gamma = \min\left\{\frac{1}{2RL}, \frac{\varepsilon}{4L\sigma^2}\right\}, \tag{200}$$

*for all integer $k \geq 0$ then we have*

$$\frac{1}{\Gamma_K} \sum_{k=0}^{K} \gamma_k \, \mathbb{E}\left[\left\|\nabla f\left(x^k\right)\right\|^2\right] \leq \varepsilon, \tag{201}$$

*with $\Gamma_K := \gamma_0 + \gamma_1 + \cdots + \gamma_K$, as long as*

$$|S| \geq \frac{4\Delta}{\varepsilon\gamma} = \max\left\{\frac{8RL\Delta}{\varepsilon}, \frac{16L\Delta\sigma^2}{\varepsilon^2}\right\},$$

*where $S := \left\{k \in [0..K] : \widetilde{\delta}^k < R\right\}$.*

*Remark G.15.* Note that the set $S$ in Theorem G.14 corresponds to the loop numbers where a positive stepsize is applied to the stochastic gradient received. Hence, $|S|$ exactly counts the number of iterative updates which was denoted by $K$ in the analysis of Ringmaster ASGD.

*Proof.* Let the stepsizes of Ringmaster ASGD $\{\gamma_k\}_{k \geq 0}$ be as in (200) then

$$\sum_{k=0}^{K} \gamma_k = \gamma \sum_{k=0}^{K} \mathbb{I}\left\{\widetilde{\delta}^k < R\right\} = \gamma \, |S|, \tag{202}$$

where we defined the set $S := \left\{k \in [0..K] : \widetilde{\delta}^k < R\right\}$. Now, we need to check that the constraints

$$\gamma_k L \left(1 + 2L \sum_{j \in M_k} \gamma_j \, \widetilde{\delta}^j\right) \leq 1, \quad k = 0, 1, 2, \ldots, K, \tag{203}$$

where $M_k := \left\{j \in [0..K] : j - \delta^j \leq k \leq j - 1\right\}$, are all fulfilled. Given $k \in [0..K]$, we distinguish two cases:

- if $\widetilde{\delta}^k \geq R$ then $\gamma_k = 0$ and $k$-th constraint from (203) is (clearly) satisfied,

- otherwise, if $\widetilde{\delta}^k < R$ then $\gamma_k = \gamma > 0$ and we have

$$\gamma_k L \left(1 + 2L \sum_{j \in M_k} \gamma_j \, \widetilde{\delta}^j\right)$$

$$\stackrel{\text{(a)}}{=} \gamma L \left(1 + 2L\gamma \sum_{j \in M_k \cap S} \widetilde{\delta}^j\right) \tag{204}$$

$$\stackrel{\text{(b)}}{\leq} \gamma L \left(1 + 2L\gamma(R-1) \, |M_k \cap S|\right)$$

$$= \gamma L + 2 \left(\gamma L\right)^2 (R-1) \, |M_k \cap S|,$$

where in (a) we use the definition of $S$, that is, for any $j \in [0..K]$ the stepsize $\gamma_j > 0$ if, and only if $j \in S$ in which case $\gamma_j = \gamma$. In (b) we use the fact that for any $j \in M_k \cap S \subseteq S$

the delay $\widetilde{\delta}^k < R$ and since it is an integer, $\widetilde{\delta}^k \leq R - 1$. Now it remains to upper bound the cardinal of the set $M_k \cap S$; we show that

$$|M_k \cap S| \leq R - 1. \tag{205}$$

To do so, we distinguish two cases: either the set is empty in which case inequality (205) holds. Otherwise, if $M_k \cap S \neq \varnothing$ then, let $m = |M_k \cap S|$ denotes the cardinal of the set and $j_1 < j_2 < \cdots < j_m$ its elements. By definition of $S$ and since all $j_1, \ldots, j_m$ are in $S$, all the stepsizes $\gamma_{j_1}, \ldots, \gamma_{j_m}$ are positive as $\widetilde{\delta}^{j_1} < R, \ldots, \widetilde{\delta}^{j_m} < R$. Moreover, by definition of $M_k$ we have, for all $i \in [m]$

$$j_i - \delta^{j_i} \leq k \leq j_i - 1,$$

hence notably $j_m - \delta^{j_m} \leq k < k + 1 \leq j_1 < j_2 < \cdots < j_m$ thus for any $i \in [m-1]$

$$j_i \in \left\{ r \in \left[ j_m - \delta^{j_m} .. j_m - 1 \right] \; : \; \gamma_r > 0 \right\},$$

and $k \in \left\{ r \in \left[ j_m - \delta^{j_m} .. j_m - 1 \right] \; : \; \gamma_r > 0 \right\}$. Moreover by definition of $\widetilde{\delta}^{j_m}$ we have

$$\widetilde{\delta}^{j_m} = \left| \left\{ r \in \left[ j_m - \delta^{j_m} .. j_m - 1 \right] \; : \; \gamma_r > 0 \right\} \right| \geq m,$$

since it contains $k, j_i, j_2, \ldots, j_{m-1}$. Hence, as $j_m \in S$ then $\widetilde{\delta}^{j_m} \leq R - 1$ thus we obtain $m \leq R - 1$ as desired.

Now, if we continue to upper bound quantity from (204), we have

$$\gamma L + 2 \left( \gamma L \right)^2 (R - 1) |M_k \cap S| \overset{(205)}{\leq} \gamma L + 2 \left( \gamma L (R - 1) \right)^2$$
$$\overset{(a)}{\leq} \frac{1}{2R} + \frac{1}{2}$$
$$\overset{(b)}{\leq} \frac{1}{2} + \frac{1}{2}$$
$$= 1,$$

where in (a) we use both the fact that the $\gamma \leq \frac{1}{2RL}$ so that

$$\gamma L \leq \frac{1}{2R} \quad \text{and} \quad 2\gamma^2 L^2 (R-1)^2 \leq \frac{2(R-1)^2}{4R^2} < \frac{1}{2},$$

while in (b) we use the fact that $R \geq 1$.

Hence, all the constraints are fulfilled. Therefor, it remains to further upper bound the quantity (199) from Theorem G.13 without the $\widetilde{R}(K)$ residual term. The first term in (199) is equal to

$$\frac{2\Delta}{\sum_{k=0}^{K} \gamma_k} \overset{(202)}{=} \frac{2\Delta}{\gamma |S|},$$

while for the stochastic term, the numerator can be upper bounded as

$$L\sigma^2 \sum_{k=0}^{K} \gamma_k^2 \left( 1 + 2L \sum_{j \in M_k} \gamma_j \right) \overset{(a)}{=} L\sigma^2 \gamma^2 \sum_{k \in S} \left( 1 + 2\gamma L |M_k \cap S| \right)$$
$$\overset{(205)}{\leq} L\sigma^2 \gamma^2 |S| \left( 1 + 2\gamma L R \right),$$

hence, when dividing by $\sum_{k=0}^{K} \gamma_k$ it gives

$$L\sigma^2 \frac{\sum_{k=0}^{K} \gamma_k^2 \left( 1 + 2L \sum_{j \in M_k} \gamma_j \right)}{\sum_{k=0}^{K} \gamma_k} \leq L\sigma^2 \frac{\gamma^2 |S| \left( 1 + 2\gamma L R \right)}{\gamma |S|} = L\sigma^2 \gamma (1 + 2\gamma L R).$$

Thus, to obtain the inequality (201) it is enough to have

$$\frac{2\Delta}{\gamma\,|S|} \leq \frac{\varepsilon}{2} \quad \text{and} \quad L\sigma^2\gamma(1 + 2\gamma LR) \leq \frac{\varepsilon}{2},$$

and, for the later inequality, it is enough to ensure $\gamma LR \leq \frac{1}{2}$ along with $L\sigma^2\gamma \leq \frac{\varepsilon}{4}$ and we recover the stepsize given in the statement, i.e., $\gamma = \min\left\{\frac{1}{2RL}, \frac{\varepsilon}{4L\sigma^2}\right\}$. Now, for the other inequality, we need to have

$$\frac{4\Delta}{\varepsilon\gamma} \leq |S|,$$

which, after plugging the expression of $\gamma$ given before leads to the desired lower bound of

$$|S| \geq \max\left\{\frac{8RL\Delta}{\varepsilon}, \frac{16L\Delta\sigma^2}{\varepsilon^2}\right\}.$$

$\square$

## G.10 A *Mixed-Integer* OPTIMIZATION PROBLEM

We review here the different optimization problems derived with our analysis of ASGD and AGD.

**The General Optimization Problem:** According to the analysis done in Theorem G.13, a natural approach to get rid of the $\widetilde{R}(K)$ term appearing in (12) is to ensure each individual factor $R_k$ to be nonpositive, i.e.,

$$R_k := \gamma_k L + 2\gamma_k L^2 \sum_{j \in M_k} \gamma_j \, \widetilde{\delta}^j - 1 \leq 0, \quad k = 0, 1, \dots, K \tag{206}$$

and, if we let

$$M_{i,j} = \begin{cases} 0, & \text{if } j \notin M_i, \\ \widetilde{\delta}^j, & \text{if } j \in M_i, \end{cases} \tag{207}$$

for all $i, j \in [0 .. K]$ then as $R(K) \leq 0$ by (206), finding theoretically optimal stepsizes $\{\gamma_k^*\}_{k \geq 0}$ is equivalent to minimize the left-hand side of (12) over the constrained region

$$\mathscr{F} = \left\{\Lambda \in [0, 1]^{K+1} : 0 \leq L\Lambda + (L\Lambda) \odot (M^\delta[L\Lambda]) \leq 1\right\},$$

where $\Lambda = (\gamma_0, \dots, \gamma_K)$ and $M^\delta = (M_{i,j})_{i,j \in [0 .. K]}$ is the "matrix of delays" defined in (207). The resulting optimization problem to solve for the optimal stepsizes $\{\gamma_k^*\}_{k \geq 0}$ can be stated as follows:

$$(\widetilde{\mathscr{P}}_K^{\sigma^2}): \quad \text{minimize} \quad \frac{1}{\gamma_0 + \dots + \gamma_K}\left[2\Delta + L\sigma^2 \sum_{k=0}^K \gamma_k^2\left(1 + 2L\sum_{j \in M_k}\gamma_j\right)\right],$$

$$\text{over} \quad (\gamma_0, \dots, \gamma_K) \in \left[0, \tfrac{1}{L}\right]^{K+1},$$
$$\text{subject to} \quad 0 \leq \gamma_k L + 2\gamma_k L^2 \sum_{j \in M_k}\gamma_j\, \widetilde{\delta}^j \leq 1 \quad \text{for } k = 0, 1, 2, \dots, K. \tag{208}$$

*Remark* G.16. Notice that, in the special case where all delays are 0, in the case of synchronous SGD for instance, then all $M_k = \varnothing$ and the constraints in (208) reduces to $0 \leq \gamma_k L \leq 1$, and to minimize the quantity

$$\frac{1}{\gamma_0 + \dots + \gamma_K}\left[2\Delta + L\sigma^2 \sum_{k=0}^K \gamma_k^2\right],$$

it's enough, due to the symmetry, to assume $\gamma_0 = \dots = \gamma_K = \gamma$ which gives $\gamma = \min\left\{\frac{1}{L}, \sqrt{\frac{2\Delta}{KL\sigma^2}}\right\}$. then, taking $K \geq \frac{2L\Delta\sigma^2}{\varepsilon^2}$ ensures an $\varepsilon$–stationary point is found which leads to

$$\gamma = \min\left\{\frac{1}{L}, \frac{\varepsilon}{L\sigma^2}\right\},$$

an improvement over the stepsizes of Ringmaster ASGD with a factor $\times 2$ to $\times 4$.

The "matrix of delay" defined in (207) has some interesting properties as stated in the next result. Examples of the matrix of delays will be provided in a subsequent paragraph.

**Lemma G.17** (Properties of the matrix of delays). *For the matrix of delays $M^\delta$ introduced in (207), we have*

1. *the matrix $M^\delta$ is strictly upper triangular, that is, $M_{i,j}^\delta = 0$ for any $0 \leq j \leq i \leq K$,*

2. *for any $j \in [0 \mathinner{..} K]$ we have $M_{j-1,j} = M_{j-2,j} = \cdots = M_{j-\delta^j,j} = \widetilde{\delta}^j$.*

*Proof.* For the first claim, let $0 \leq i, j \leq K$ such that $j \leq i$ then clearly we can't have $i \leq j - 1$ hence necessarily $j \notin M_i := \left\{ j' \in [0 \mathinner{..} K] : j' - \delta^{j'} \leq i \leq j' - 1 \right\}$. Consequently, we deduce that $M_{i,j}^\delta = 0$, i.e., the matrix $M^\delta$ is strictly upper triangular.

For the second statement, we use again the definition of the sets $\{M_i\}_{0 \leq i \leq K}$. Let $j \in [0 \mathinner{..} K]$ from (184) then for any integer $i$ between $j - \delta^j$ and $j - 1$ we have $j \in M_i$, because $j - \delta^j \leq i \leq j - 1$. Hence, we deduce that $j \in M_{j-1}, j \in M_{j-2}, \ldots, j \in M_{j-\delta^j}$ that is ot say $M_{j-1,j} = M_{j-2,j} = \cdots = M_{j-\delta^j,j} = \widetilde{\delta}^j$, as desired. Note that the quantity $M_{j-\delta^j,j}$ is well-defined since $0 \leq \delta^j \leq j$. $\qquad\square$

Observe that the optimization problem (208) is a nonlinear mixed-integer program which in practice is hard to solve, notably the objective function is even nonlinear. This "mixed-integer" characteristic comes from the effective delays $\{\widetilde{\delta}^k\}_{k \geq 0}$ which intrinsically depends on the binary variables

$$ b_k := \mathbb{I}\{\gamma_k = 0\} . $$

A part of the "hardness" of problem $(\widetilde{\mathscr{P}}_K^{\sigma^2})$ arises from the presence of the stochastic term in $\sigma^2$. For now on, we focus on the simpler case where $\sigma^2 = 0$, i.e., the machines compute *full* gradients instead of noisy ones in the sense that when asked to compute a gradient of $f$ at $x \in \mathbb{R}^d$ they will reply, deterministically, $\nabla f(x)$ after some time. Assuming $\sigma^2 = 0$ we can rewrite the minimization problem (208) as a maximization problem:

$$(\widetilde{\mathscr{P}}_K): \quad \text{maximize } \gamma_0 + \gamma_1 + \cdots + \gamma_K,$$

$$\text{over} \quad (\gamma_0, \ldots, \gamma_K) \in [0, \tfrac{1}{L}]^{K+1}, \tag{209}$$

$$\text{subject to } 0 \leq \gamma_k L + 2\gamma_k L^2 \sum_{j \in M_k} \gamma_j \widetilde{\delta}^j \leq 1 \quad \text{for } k = 0, 1, 2, \ldots, K.$$

This simpler problem seems much more tractable at first glance since now it has a linear objective in the variables $(\gamma_0, \ldots, \gamma_K)$ and we can use general-purpose solvers like Gurobi 11 (Gurobi Optimization, LLC, 2024) to attempt solving it. Gurobi approach to solve optimization problems of the form of $(\widetilde{\mathscr{P}}_K)$ uses branch-and-bound to systematically partition the feasible space into subproblems and constructs relaxations at each node. The algorithm provides mathematically guaranteed global optimality by maintaining upper and lower bounds across all active nodes until the optimality gap closes. However, this approach can demand millions of simplex iterations on some instances.

**A Bilinear Program:** While it is tractable to solve problem (209) numerically, the presence of the *effective* delays $\{\widetilde{\delta}^k\}_{k \geq 0}$ makes it difficult to study directly the theoretical properties of the optimal solutions. To further simplify $(\widetilde{\mathscr{P}}_K)$ we consider the following problem:

$$(\mathscr{P}_K): \quad \text{maximize } \gamma_0 + \gamma_1 + \cdots + \gamma_K,$$

$$\text{over} \quad (\gamma_0, \ldots, \gamma_K) \in [0, \tfrac{1}{L}]^{K+1}, \tag{210}$$

$$\text{subject to } 0 \leq \gamma_k L + 2\gamma_k L^2 \sum_{j \in M_k} \gamma_j \delta^j \leq 1 \quad \text{for } k = 0, 1, 2, \ldots, K.$$

where, instead of the effective delays, we directly use $\{\delta^k\}_{k \geq 0}$ which are simply constants in our problem. Of course, the optimal solutions of this new maximization problem are, in general, looser than those provided by the mixed-integer problem $(\widetilde{\mathscr{P}}_K)$ (in term of objective function value); this

can be seen by taking a feasible solution $\{\gamma_k\}_{k\geq 0}$ of $(\mathscr{P}_K)$ and using the inequality $\widetilde{\delta}^k \leq \delta^k$, this gives

$$0 \leq \gamma_k L + 2\gamma_k L^2 \sum_{j \in M_k} \gamma_j \, \widetilde{\delta}^j \leq 0 \leq \gamma_k L + 2\gamma_k L^2 \sum_{j \in M_k} \gamma_j \, \delta^j \overset{(a)}{\leq} 1,$$

where (a) follows by the feasibility of $\{\gamma_k\}_{k\geq 0}$. So, $\{\gamma_k\}_{k\geq 0}$ is still a feasible solution for $(\widetilde{\mathscr{P}}_K)$, showing that the optimal value of problem (209) is always at least as large as the one of (210).

The new optimization problem $(\mathscr{P}_K)$ belongs to the family of *bilinear programs* (and also to the class of *reverse-convex programs*). Surprisingly, with a little more effort, we can also extend our main Theorem 4.4 (characterization of the optimal solution(s) of problem $(\mathscr{P}_K)$ in (210)) to our original mixed-integer problem $(\widetilde{\mathscr{P}}_K)$.

**Reformulating Problem $(\widetilde{\mathscr{P}}_K)$:**   We now reformulate problem $(\widetilde{P}_K)$ in a more friendly way using binary variables. This gives rises to the optimization problem $(\mathscr{P}_K^{\mathrm{mi}})$ where "*mi*" stands for *mixed-integer*. First, let us recall the constraints of the mixed-integer problem $(\widetilde{\mathscr{P}}_d)$, i.e.,

$$0 \leq \gamma_k L + 2\gamma_k L^2 \sum_{j \in M_k} \gamma_j \, \widetilde{\delta}^j - 1 \leq 0, \tag{211}$$

for all integer $k \in [0..K]$. Since $\widetilde{\delta}^k$ depends on whether some of the stepsizes $\gamma_j$ for $j \in \left[k - \delta^k .. k - 1\right]$ are positive or zero, we introduce *binary* variables

$$b_k := \mathbb{I}\{\gamma_k = 0\} \in \{0, 1\}, \tag{212}$$

where $k \in [0..K]$. So, by the definition of $\widetilde{\delta}^k$ from (14) we can rewrite it as

$$\widetilde{\delta}^k \overset{(14)}{=} \delta^k - \left|\left\{j \in \left[k - \delta^k .. k - 1\right] \,:\, \gamma_j = 0\right\}\right| \overset{(212)}{=} \delta^k - \sum_{p=k-\delta^k}^{k-1} b_p, \tag{213}$$

and since

$$\delta^j - \sum_{p=j-\delta^j}^{j-1} b_p = \sum_{p=j-\delta^j}^{j-1} (1 - b_p),$$

then, plugging (213) back in (211) gives for all $k \in [0..K]$

$$\gamma_k L + 2\gamma_k L^2 \sum_{j \in M_k} \sum_{p=j-\delta^j}^{j-1} \gamma_j (1 - b_p) - 1 \leq 0. \tag{214}$$

The above reformulation is more compact for practical implementation and lead to the following mixed-integer nonlinear program

$(\mathscr{P}_K^{\mathrm{mi}})$:   maximize $\gamma_0 + \gamma_1 + \cdots + \gamma_K$

over        $(\gamma_0, \ldots, \gamma_K) \in [0, \tfrac{1}{L}]^{K+1}$

subject to $0 \leq \gamma_k L \left(1 + 2L \sum_{j \in M_k} \sum_{p=j-\delta^j}^{j-1} \gamma_j (1 - b_p)\right) \leq 1$   for $k = 0, 1, \ldots, K$;

and $b_p = \mathbb{I}\{\gamma_p = 0\}$, for $p = 0, 1, \ldots, K$.

$$\tag{215}$$

Even though (215) is not anymore a bilinear program, we can still implement it in Gurobi 11 using the *Big-M method*. We further expand on implementation details concerning problem $(\mathscr{P}_K^{\mathrm{mi}})$ in a subsequent paragraph.

**Extending Theorem 4.4.**   In this paragraph, we will extend our main result (Theorem 4.4) to optimization problems of the form of $(\mathscr{P}_K^{\mathrm{mi}})$, which is formalized in the next result:

**Theorem G.18.** *For any positive real number $L > 0$, any integer $K \geq 0$ and any sequence of integers $\{\delta^k\}_{k \geq 0}$ such that $0 \leq \delta^k \leq k$ for all $k \geq 0$ then, any global maximizers $\{\gamma_k^*\}_{k \geq 0}$ of problem $(\mathscr{P}_K^{\mathrm{mi}})$ satisfies, for all $k \in [0 .. K]$*

$$\gamma_k^* = 0 \ \ or \ \ \gamma_k^* L \left(1 + 2L \sum_{j \in M_k} \sum_{p=j-\delta^j}^{j-1} \gamma_j^* (1 - b_p^*)\right) = 1, \tag{216}$$

*where $b_k^* = \mathbb{I}\{\gamma_k^* = 0\}$.*

*Proof.* Up to a scaling factor of $L$ in the optimal solutions, let us assume without loss of generality that $L = 1$. First, let us recall that for all $k \in [0 .. K]$ we have

$$\widetilde{\delta}^k := \delta^k - \left|\left\{j \in [k - \delta^k .. k - 1] \ : \ \gamma_j = 0\right\}\right|. \tag{217}$$

Additionally, observe that the sets $M_k$ for $k \in [0 .. K]$ does only depends on the delays $\{\delta^j\}_{j \geq 0}$ with Now let us suppose, for the sake of contradiction, that there exists an optimal solution $\{\gamma_k^*\}_{k \geq 0}$ for which (216) does not hold, that is, there exists $k_0 \in [0 .. K]$ such that

$$0 < \gamma_{k_0}^* \left(1 + 2 \sum_{j \in M_{k_0}} \sum_{p=j-\delta^j}^{j-1} \gamma_j^* (1 - b_p^*)\right) < 1. \tag{218}$$

For now on, let us fix $S_0 = \{i \in [0 .. K] \ : \ \gamma_i^* = 0\}$ and $T_0 = [0 .. K] \setminus S_0$, notably by (218) we have $k_0 \in T_0$. Then, observe that $\{\gamma_k^*\}_{k \in T_0}$ is a feasible solution for the optimization problem

$$(\mathscr{P}_K^*): \quad \text{maximize} \sum_{k \in T_0} \gamma_k$$

$$\text{over} \quad \{\gamma_k\}_{k \in T_0} \in [0, 1]^{|T_0|}$$

$$\text{subject to } 0 \leq \gamma_k \left(1 + 2 \sum_{j \in M_k \cap T_0} \bar{\delta}^j \gamma_j\right) \leq 1 \tag{219}$$

$$\text{for } k \in T_0;$$

where we just kept the indices $k \in [0 .. K]$ for which $\gamma_k^* > 0$ since the other indices (for which the corresponding variable $\gamma_k^*$ is zero) do neither impact the objective value nor the variables $\gamma_k$ for $k \in T_0$. Additionally, we defined in (219)

$$\bar{\delta}^k := \delta^k - \left|\left\{j \in [k - \delta^k .. k - 1] \ : \ \gamma_j^* = 0\right\}\right| \geq 0. \tag{220}$$

Note that in problem $(\mathscr{P}_K^*)$ the "delays" $\{\bar{\delta}^k\}_{k \geq 0}$ are fixed contrary to $(\mathscr{P}_K^{\mathrm{mi}})$. It is important to observe that the coefficient $\bar{\delta}^k$ is simply $\widetilde{\delta}^k$ when in (217) we use the tuple $\{\gamma_k^*\}_{k \geq 0}$. We can now apply Theorem 4.4 on the optimization problem (219), notably, using (218) which is equivalent to

$$0 < \gamma_k^* \left(1 + 2 \sum_{j \in M_k \cap T_0} \bar{\delta}^j \gamma_j^*\right) < 1,$$

we obtain that the feasible solution $\{\gamma_k^*\}_{k \in T_0}$ of $(\mathscr{P}_K^*)$ is not extremal and thus is not optimal. Hence, let us denote by $\{\bar{\gamma}_k\}_{k \in T_0}$ an optimal solution of $(\mathscr{P}_K^*)$ (which by Theorem 4.4 is extremal too) so

$$\sum_{k \in T_0} \gamma_k^* < \sum_{k \in T_0} \bar{\gamma}_k. \tag{221}$$

Next, let us complete $\{\bar{\gamma}_k\}_{k \in T_0}$ into a tuple $\{\bar{\gamma}_k\}_{k \geq 0}$ where $\bar{\gamma}_k = 0$ for all integer $k \notin T_0$. First, by construction of the optimization problem $(\mathscr{P}_K^*)$ and the optimal solution $\{\bar{\gamma}_k\}_{k \geq 0}$, for any $k \in S_0$ we have $\bar{\gamma}_k = 0$ hence, for all $k \in [0 .. K]$

$$\bar{\delta}^k \overset{(220)}{=} \delta^k - \left|\left\{j \in [k - \delta^k .. k - 1] \ : \ \gamma_j^* = 0\right\}\right|$$

$$\geq \delta^k - \left|\left\{j \in [k - \delta^k .. k - 1] \ : \ \bar{\gamma}_j = 0\right\}\right|, \tag{222}$$

so since all $\{\bar{\gamma}_k\}_{k \geq 0}$ are non-negative then for $k \in [0 .. K]$

$$0 \leq \bar{\gamma}_k \left( 1 + 2 \sum_{j \in M_k} \sum_{p=j-\delta^j}^{j-1} \bar{\gamma}_j \left( 1 - \bar{b}_p \right) \right), \tag{223}$$

where $\bar{b}_p := \mathbb{I}\{\bar{\gamma}_p = 0\}$ with $p \in [0 .. K]$. Using (213) and (222) we obtain

$$\sum_{p=j-\delta^j}^{j-1} \bar{\gamma}_j \left( 1 - \bar{b}_p \right)$$

$$\stackrel{(213)}{=} \bar{\gamma}_j \left( \delta^j - \left| \{ j \in \left[ k - \delta^k .. k - 1 \right] : \bar{\gamma}_j = 0 \} \right| \right) \tag{224}$$

$$\stackrel{(222)}{\leq} \bar{\gamma}_j \left( \delta^j - \left| \{ j \in \left[ k - \delta^k .. k - 1 \right] : \gamma_j^* = 0 \} \right| \right)$$

$$\stackrel{(220)}{=} \bar{\gamma}_j \, \bar{\delta}^j,$$

hence, for any $k \in [0 .. K]$

$$\bar{\gamma}_k \left( 1 + 2 \sum_{j \in M_k} \sum_{p=j-\delta^j}^{j-1} \bar{\gamma}_j \left( 1 - \bar{b}_p \right) \right)$$

$$\stackrel{(224)}{\leq} \bar{\gamma}_k \left( 1 + 2 \sum_{j \in M_k} \bar{\delta}^j \, \bar{\gamma}_j \right) \tag{225}$$

$$\stackrel{(a)}{=} \bar{\gamma}_k \left( 1 + 2 \sum_{j \in M_k \cap T_0} \bar{\delta}^j \, \bar{\gamma}_j \right)$$

$$\stackrel{(b)}{\leq} 1,$$

where in (a) we use the fact that for all $k \notin T_0$, by construction, $\bar{\gamma}_k = 0$ while in (b) we use the fact that $\{\bar{\gamma}_k\}_{k \in T_0}$ is a feasible solution of $(\mathscr{P}_K^*)$.

Combining the inequalities (223) and (225) for all integer $k \in [0 .. K]$ we deduce that $\{\bar{\gamma}_k\}_{k \geq 0}$ is a feasible solution of problem $(\mathscr{P}_K^{\mathrm{mi}})$ thus, using the strict inequality (221) we obtain

$$\sum_{k=0}^{K} \gamma_k^* = \sum_{k \in T_0} \gamma_k^* < \sum_{k \in T_0} \bar{\gamma}_k \stackrel{(a)}{=} \sum_{k=0}^{K} \bar{\gamma}_k \leq \mathrm{val}\left( \mathscr{P}_K^{\mathrm{mi}} \right), \tag{226}$$

where (a) follows by construction of the $\{\bar{\gamma}_k\}_{k \geq 0}$ and $\mathrm{val}\left( \mathscr{P}_K^{\mathrm{mi}} \right)$ denotes the optimal value of problem $(\mathscr{P}_K^{\mathrm{mi}})$. Inequality (226) establishes the sub-optimality of the feasible solution $\{\gamma_k^*\}_{k \geq 0}$ which leads to a contradiction since we assume originally that is was optimal. Hence, we conclude that all the optimal solutions of the optimization problem $(\mathscr{P}_K^{\mathrm{mi}})$ satisfy the "alternative" (216) and this achieves the proof of the theorem. $\qquad \square$

**Practical Implementation in Gurobi: the *Big-M Method*.** In order to implement the mixed-integer nonlinear optimization problem (215), we need to handle a trilinear product of variables of the form

$$\gamma_k \gamma_j (1 - b_p),$$

where $b_p = \mathbb{I}\{\gamma_p = 0\}$. In Gurobi 11 and older versions, while bilinear terms in the constraint are supported, products of 3 or more variables like in the constraints of problem $(\mathscr{P}_{\mathscr{L}}^{\mathrm{mi}})$ are not directly supported and require some tricks, especially since in our case one of the variable involved is binary (the $1 - b_p$ in $(\mathscr{P}_{\mathscr{L}}^{\mathrm{mi}})$). To overcome this issue, we employ a technique called the *Big-M Method*. For this we introduce a new continuous variables $z_{j,p}$ whose value will be forced to $\gamma_j (1 - b_p)$. It is enough to notice that the equality $z_{j,p} = \gamma_j (1 - b_p)$ is equivalent to the set of inequalities

$$\begin{cases} 0 \leq z_{j,p}, \\ z_{j,p} \leq \gamma_j, \\ z_{j,p} \leq 1 - b_p, \\ \gamma_j + b_p \leq z_{j,p}. \end{cases}$$

Effectively, as $0 \leq z_{j,p} \leq 1 - b_p$ then if $b_p = 1$ we deduce that $z_{j,p} = 0$. Otherwise, if $b_p = 0$ then we have both $\gamma_j \leq z_{j,p} \leq \gamma_{j,p}$ as desired.

### G.11 A Provable Factor–2 Approximation

**Theorem 5.5** (Near Optimality of Ringmaster AGD). *For any integer $K \geq 0$ the stepsizes $\{\gamma_k^{(R)}\}_{k \geq 0}$ of* Ringmaster AGD *(with a threshold[33] of $R = 1$) satisfy*

$$\sum_{k=0}^{K} \gamma_k^{(R)} \leq \sum_{k=0}^{K} \gamma_k^* \leq 2 \sum_{k=0}^{K} \gamma_k^{(R)},$$

*with $\{\gamma_k^*\}_{k \geq 0}$ the optimal stepsizes and $\gamma_k^{(R)} = \frac{1}{L}\mathbb{I}\{\widetilde{\delta}^k = 0\}$.*

*Proof.* The proof of the above theorem builds on several intermediate lemmas we state and prove below.

**Lemma G.19.** *We have $\gamma_0^{(R)} = \frac{1}{L}$.*

*Proof.* Since $\delta^0 = 0$ by definition of the sequence of delays (see (14)) and as $0 \leq \widetilde{\delta}^0 \leq \delta^0$ we deduce that

$$\gamma_0^{(R)} = \frac{1}{L}\mathbb{I}\{\widetilde{\delta}^0 = 0\} = \frac{1}{L},$$

as desired. $\qquad\square$

Hence, based on Lemma G.19, we can define the (finite) sequence $t_0 = 0 < t_1 < \cdots < t_i \leq K$ (with eventually $i = 0$) of loop number for which the stepsizes of Ringmaster ASGD when $R = 1$ are nonzero, i.e., for all $j \in [0 .. K]$

$$\gamma_j^{(R)} \neq 0 \ \text{ iff } \ j \in \{t_0, t_1, \ldots, t_i\}.$$

It is important to note that the *effective* delay $\{\widetilde{\delta}^k\}_{k \geq 0}$ depends on how the stepsizes are chosen. To prevent confusion, we denote by $\{\widetilde{\delta}_*^k\}_{k \geq 0}$ the effective delays for an (arbitrarily taken, but fixed) optimal solution $\{\gamma_k^*\}_{k \geq 0}$.

**Lemma G.20.** *For any $j \in [0 .. i - 1]$, there do not exists integers $t_j \leq \ell_1 < \ell_2 \leq t_{j+1} - 1$ such that the same worker sends a stochastic gradient at loop number $\ell_1$ and $\ell_2$.*

*Proof.* For the sake of contradiction, assume not and suppose worker $p \in [n]$ sends a stochastic gradient to the server at both loop number $\ell_1$ and $\ell_2$. Without loss of generality, we can assume $\ell_1$ and $\ell_2$ to be the first two times where worker $p$ sends a stochastic gradient in the time frame $[t_j, t_{j+1} - 1]$. By definition of the sequence $\{t_j\}_{j \in [0 .. i]}$ we know that all the stochastic gradients received by the server from loop number $t_j + 1$ to $\ell_2 - 1$ are discarded. Hence,

$$\widetilde{\delta}^{\ell_2} = \delta^{\ell_2} - \left|\left\{ j \in [\ell_2 - \delta^{\ell_2} .. \ell_2 - 1] \ : \ \gamma_j^{(R)} = 0 \right\}\right| = 0, \tag{227}$$

since by definition of the delay $\delta^{\ell_2} = \ell_2 - \ell_1 - 1$ is the number of stochastic gradients received by the server between times $\ell_1$ and $\ell_2$ (endpoints excluded). But (227) and the fact that $t_j < \ell_2 < t_{j+1}$ contradict the definition of the sequence $\{t_j\}_{j \in [0 .. i]}$. Thus, the claimed property holds. $\qquad\square$

Hence, the previous lemma asserts that for all $j \in [0 .. i - 1]$, on the time frame $[t_j, t_{j+1} - 1]$ the server receives stochastic gradients from distinct workers only. In particular, this shows that

$$\delta^\ell \geq \ell - t_j, \tag{228}$$

for all $\ell \in [t_j, t_{j+1} - 1]$: this remark is actually at the core of the proof and is crucial for the next part. For now, let us fix $j \in [0 .. i - 1]$ and focus on the time frame $[t_j, t_{j+1} - 1]$ (in case $i = 0$, we can just replace $t_{j+1} - 1$ by the last loop number). We would like to compare the stepsizes

---

[33]Following the choice of Maranjyan et al. (2025), when $\sigma^2 = 0$ then $R = 1$.

$\gamma_{t_j}^*, \ldots, \gamma_{t_{j+1}-1}^*$ to those arising when solving a similar *mixed-integer* optimization problem but restricted to the time frame $[t_j, t_{j+1} - 1]$. Let $\gamma_0^\star, \ldots, \gamma_{s-1}^\star$ be an optimal solution of

$$(\widetilde{\mathscr{P}_K^\star}): \quad \text{maximize} \ \ \gamma_0 + \gamma_1 + \cdots + \gamma_{s-1},$$

$$\text{over} \quad (\gamma_0, \ldots, \gamma_{s-1}) \in [0, \tfrac{1}{L}]^s, \tag{229}$$

$$\text{subject to} \ \ 0 \le \gamma_k L + 2\gamma_k L^2 \sum_{j=k+1}^{s-1} \gamma_j \widetilde{\delta}^j \le 1 \quad \text{for } k = 0, 1, 2, \ldots, s - 1.$$

where $s = t_{j+1} - t_j$ (or $s = K$ if $i = 0$) is the size of the time frame $[t_j, t_{j+1} - 1]$. The optimization problem (229) arises for instance when only distinct workers send a stochastic gradient to the server. In this case we have $\delta^k = k$ for all $k \in [0 .. s - 1]$ and the sets $M_k$ reduces to

$$M_k = \left\{ j \in [0 .. s - 1] \ : \ j - \delta^j \le k \le j - 1 \right\} = [k + 1 .. s - 1].$$

Let $\{\widetilde{\delta}_\star^j\}_{j \in [0 .. s-1]}$ and $\{\widetilde{\delta}_{\star,r}^\ell\}_{\ell \in [t_j .. t_{j+1}-1]}$ be respectively the effective delays associated to $\gamma_0^\star, \ldots, \gamma_{s-1}^\star$ and $\gamma_{t_j}^*, \ldots, \gamma_{t_{j+1}-1}^*$ when restricted to the time frame $[t_j, t_{j+1} - 1]$, i.e., for $\ell \in [t_j .. t_{j+1} - 1]$ we define

$$\widetilde{\delta}_{\star,r}^\ell = (\ell - t_j) - \left| \left\{ j \in [t_j .. \ell - 1] \ : \ \gamma_j^* = 0 \right\} \right|. \tag{230}$$

We prove the following lemma.

**Lemma G.21.** *For any $j \in [0 .. i - 1]$ we have*

$$\sum_{\ell=t_j}^{t_{j+1}-1} \gamma_\ell^* \le \sum_{\ell=0}^{s-1} \gamma_\ell^\star. \tag{231}$$

*Proof.* Fix some $j \in [0 .. i - 1]$, we know that $0 \le \gamma_\ell^* \le \tfrac{1}{L}$ for all $\ell \in [t_j, t_{j+1} - 1]$. It is enough for proving (231) to establish that $\gamma_{t_j}^*, \ldots, \gamma_{t_{j+1}-1}^*$ is a feasible solution of (229). Let $k \in [t_j .. t_{j+1} - 1]$, we have

$$\gamma_k^* L + 2\gamma_k^* L^2 \sum_{j=k+1}^{t_{j+1}-1} \gamma_j^* \widetilde{\delta}_{\star,r}^j \overset{(a)}{\le} \gamma_k^* L + 2\gamma_k^* L^2 \sum_{j=k+1}^{t_{j+1}-1} \gamma_j^* \widetilde{\delta}_\star^j \overset{(b)}{\le} \gamma_k^* L + 2\gamma_k^* L^2 \sum_{j \in M_k} \gamma_j^* \widetilde{\delta}_\star^j \le 1,$$

where the last inequality follows from the feasibility of $\{\gamma_k^*\}_{k \in [0 .. K]}$. The inequality (a) follows from

$$\widetilde{\delta}_\star^\ell := \delta^\ell - \left| \left\{ j \in [\ell - \delta^\ell .. \ell - 1] \ : \ \gamma_j^* = 0 \right\} \right|$$

$$= \left( [\ell - t_j] - \left| \left\{ j \in [t_j .. \ell - 1] \ : \ \gamma_j^* = 0 \right\} \right| \right) + \left( \delta^\ell - [\ell - t_j] - \left| \left\{ j \in [\ell - \delta^\ell .. t_j - 1] \ : \ \gamma_j^* = 0 \right\} \right| \right)$$

$$(\ell - t_j) - \left| \left\{ j \in [t_j .. \ell - 1] \ : \ \gamma_j^* = 0 \right\} \right| \tag{232}$$

$$\overset{(230)}{=} \widetilde{\delta}_{\star,r}^\ell,$$

where in (232) we use (228), i.e.,

$$\delta^\ell - [\ell - t_j] \ge 0 \quad \text{and} \quad \left| \left\{ j \in [\ell - \delta^\ell .. t_j - 1] \ : \ \gamma_j^* = 0 \right\} \right| \le \delta^\ell - [\ell - t_j],$$

and (b) follows from the non-negativity of all $\gamma_j^*$ and all $\widetilde{\delta}_\star^j$ along with the inclusion

$$[k + 1 .. t_{j+1} - 1] \subseteq M_k = \left\{ j \in [0 .. K] \ : \ j - \delta^j \le k \le j - 1 \right\},$$

since for all $\ell \in [k + 1 .. t_{j+1} - 1]$ we have $\ell - 1 \ge k$ and

$$\ell - \delta^\ell \overset{(230)}{\le} t_j \le k,$$

as desired. This shows that $\gamma_{t_j}^*, \ldots, \gamma_{t_{j+1}-1}^*$ is a feasible solution of (229) from where inequality (231) is a consequence. $\qquad \square$

*Remark G.22.* The inequality (231) also holds on the last block $[t_i, K]$ for the same reasons.

Equipped with Lemma G.19 we now need to upper bound the sum $\gamma_0^\star + \cdots + \gamma_{s-1}^\star$, which we do in the next lemmas. We first start by a technical lemma.

**Lemma G.23** (A Technical Result). *Let $n > 0$ be an integer and define the sequence $(u_i)_{i \in [n]}$ by $u_1 = 1$ and for all $i \in [n-1]$ by the recurrent relation*

$$u_{i+1} = \frac{u_i}{1 + 2u_i^2(n-i)},$$

*then, we have*[34]

$$S_n := \sum_{i=1}^{n} u_i \le 2.$$

*Proof.* First, we prove by induction on $i \in [n-1]$ that $0 \le u_{i+1} \le u_i$. For the base case $i = 1$ we have

$$u_2 = \frac{u_1}{1 + 2u_1^2(n-1)} = \frac{1}{2n-1} \le 1 = u_1, \tag{233}$$

and $u_2 \ge 0$ too. Now, assuming $0 \le u_{i+1} \le u_i$ holds for some integer $0 \le i \le n-2$ we have

$$u_{i+2} = \frac{u_{i+1}}{1 + 2u_{i+1}^2(n-(i+1))} \le u_{i+1},$$

since $1 + 2u_{i+1}^2(n-(i+1)) \ge 1$ (because $i+1 \le n$). Moreover, we also deduce that $u_{i+2} \ge 0$ since by the induction hypothesis we have $u_{i+1} \ge 0$. This proves the claim, as desired.

Now, as the sequence $(u_i)_{i \in [n]}$ is monotonically non-increasing we have

$$u_i \le u_2 \overset{(233)}{=} \frac{1}{2n-1}, \tag{234}$$

for all $i \in [2 \,..\, n]$ thus

$$S_n = \sum_{i=1}^{n} u_i = u_1 + \sum_{i=2}^{n} u_i \overset{(234)}{\le} 1 + \frac{n}{2n-1} \le 2,$$

and this achieves the proof of the lemma. $\square$

**Lemma G.24.** *For all $s \ge 0$, any optimal solution $\gamma_0^\star, \ldots, \gamma_{s-1}^\star$ of (229) satisfies*

$$\sum_{\ell=0}^{s-1} \gamma_\ell^\star \le \frac{2}{L}. \tag{235}$$

*Proof.* Let $S = \{j \in [0 \,..\, s-1] : \gamma_s^\star = 0\}$ and denote by $T = [s] \setminus S$ the indices for which the stepsizes are positive. Let us prove that

$$\sum_{\ell=0}^{s-1} \gamma_\ell^\star = \sum_{\ell \in T} \gamma_\ell^\star \le \frac{S_{|T|}}{L}, \tag{236}$$

where the sequence $(S_n)_{n \ge 1}$ is the one defined in Lemma G.23. Once inequality (236) is established, the desired claim (235) will follow since $S_n \le 2$ for all $n \ge 1$ by Lemma G.23. Let us now prove the inequality (236). By Theorem G.18, we know that the optimal solution $\{\gamma_k^\star\}_{k \in [0 \,..\, s-1]}$ is such that all constraints in Equation (209) (and so in (229)) are tight thus, for all $\ell \in T$, since $\gamma_\ell^\star > 0$ then if we let $T = \{j_0, \ldots, j_{|T|-1}\}$ where $0 = j_0 < j_1 < \cdots < j_m$ with $m = |T| - 1 \ge 0$[35], we have

$$\gamma_\ell^\star = \frac{1}{L} \cdot \frac{1}{1 + 2L \sum_{j=\ell+1}^{s-1} \gamma_j^\star \widetilde{\delta}_\star^j} \overset{(a)}{=} \frac{1}{L} \cdot \frac{1}{1 + 2L \sum_{\substack{r \in [0 \,..\, m] \\ j_r > \ell}} \gamma_{j_r}^\star r},$$

---

[34]It can be proved that $S_n \xrightarrow[n \to +\infty]{} 1 + \arctan\left(\sqrt{5 - 2\sqrt{6}}\right)\sqrt{2} \approx 1.4352098756$. As of now, it is an open question to prove $(S_n)_{n \ge 1}$ is a monotonically non-decreasing sequence.

[35]The optimal solution is never $(0, \ldots, 0)$ since we can always take $\gamma_0^\star = \frac{1}{L}$ and all other variables to 0. Additionally, the first stepsize $\gamma_0^\star$ if never zero.

where in (a) we use the definition of the effective delays: as soon as one of the stepsizes is zero, it is "removed" from the effective delays. In other words, since the *effective* delay counts exactly how many stochastic gradients have been accepted by the server since the iteration $0$ (this is specific to our case here), we have

$$\widetilde{\delta}_\star^j = \begin{cases} r, & \text{if } j = j_r \text{ for some } r \in [0 \mathinner{..} m]; \\ 0, & \text{otherwise}; \end{cases}$$

thus, if we let $u_i = L\gamma_{j_{m-i}}^\star$ for all $i \in [0 \mathinner{..} m]$ then the stepsizes $\{\gamma_{j_r}^\star\}_{r \in [0 \mathinner{..} m]}$ can be computed using the following recurrent system:

$$u_0 = 1 \quad \text{and} \quad u_i = \frac{1}{1 + 2\sum\limits_{r=0}^{i-1}(m-r)u_r}, \tag{237}$$

for all $i \in [0 \mathinner{..} m]$. Using (237) we obtain

$$\frac{1}{u_{i+1}} = \frac{1}{u_i} + 2(m-i)u_i,$$

for all $i \in [0 \mathinner{..} m-1]$ thus

$$u_{i+1} = \frac{1}{\frac{1}{u_i} + 2(m-i)u_i} = \frac{u_i}{1 + 2u_i^2(m-i)}; \tag{238}$$

Hence, using Lemma G.23 combined with (238) yields

$$\sum_{\ell=0}^{s}\gamma_\ell^\star = \sum_{r=0}^{m}\gamma_{j_r}^\star = \sum_{r=0}^{m}\frac{u_r}{L} \overset{(238)}{=} \frac{S_{m+1}}{L} \overset{\text{Lem. G.23}}{\leq} \frac{2}{L},$$

as desired. This achieves the proof of the lemma. $\qquad\square$

Finally, combining Lemmas G.21 and G.24 we obtain

$$\sum_{\ell=0}^{K}\gamma_\ell^* = \sum_{j=0}^{i-1}\sum_{\ell=t_j}^{t_{j+1}-1}\gamma_\ell^* + \sum_{j=t_i}^{K}\gamma_j^*$$

$$\overset{\text{Lem. G.21}}{\leq} \sum_{j=0}^{m-1}\sum_{\ell=t_j}^{t_{j+1}-1}\gamma_\ell^\star + \sum_{j=t_i}^{K}\gamma_j^\star$$

$$\overset{\text{Lem. G.24}}{\leq} \sum_{j=0}^{i-1}\frac{2}{L} + \frac{2}{L}$$

$$= \frac{2(i+1)}{L}$$

$$= 2\sum_{j=0}^{i}\gamma_{t_j}^{(R)}$$

$$= 2\sum_{\ell=0}^{K}\gamma_\ell^{(R)},$$

and this concludes the proof of the main theorem. $\qquad\square$

# H  EXPERIMENTS

## H.1  THE STOCHASTIC REPETITION BENCHMARK

We present on Figure 6a the measures of runtime and the number of iterations of both Gurobi 11 and the MMAHH solver on the *Stochastic Repetition benchmark*, which is the benchmark that corresponds to $\mathcal{L}_W$ which consists of repeating $c$ times a randomly sampled elementary sequence of length $n$ (with entries chosen uniformly in random between 1 and 100). We run both solvers on three instances of this benchmark, namely, $(n,c) = (9,5)$, $(n,c) = (8,4)$ and $(n,c) = (9,4)$. While the MMAHH keeps a comparable performance compared to the *Cyclic Staircase Benchmark* (see Figure 3) in both the runtime and in number of iterations, instead, Gurobi has much more difficulties with this benchmark. More precisely, the MMAHH attains up to a $10^5\times$ speed-up in runtime while requiring up to $5000\times$ less iterations.

## H.2  THE RANDOM SEQUENCES BENCHMARK

In this section we present the performance results of Gurobi and the MMAHH on the *Random Sequences benchmark*, which corresponds to lists $\mathcal{L}_W$ in $\mathbb{R}^d$ whose entries are randomly chosen between 0 and 10000. For this benchmark again, the MMAHH again outperforms Gurobi across all tested dimensions, achieving speed-ups of up to $5 \cdot 10^4$ factor, and reducing the number of iterations by up to a factor of 100. We present the results in Figure 6b.

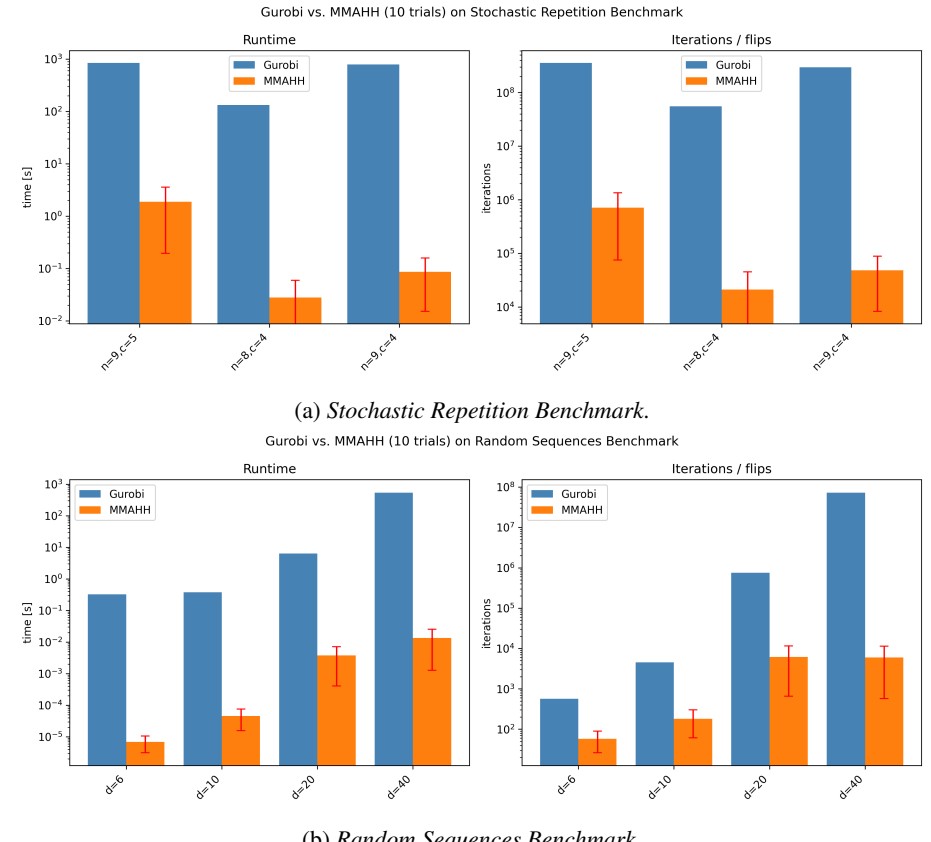

(a) *Stochastic Repetition Benchmark.*

(b) *Random Sequences Benchmark.*

Figure 6: Comparison of solver runtime (left) and number of iterations (right) for Gurobi (blue) vs. MMAHH (orange). For the MMAHH, means and standard deviations are taken over 10 runs.

## H.3   LANDSCAPE OF THE DISCRETE FUNCTION

This experiment aims at representing the function $\varphi(w) := \langle \mathbf{a} \mid \Psi(w) \rangle$ for $w \in \{0,1\}^d$, we choose to represent this function for the instance $(n, c) = (5, 4)$ of the *Cyclic Staircase Benchmark* (Figure 7a), the instance $(n, c) = (5, 4)$ of the *Stochastic Repetition Benchmark* (Figure 7b). For that, we plot the values of $2^d$ the bit-strings in $\{0,1\}^d$. We group the points $w$ by their Hamming distance to the optimum $w^*$, more precisely, the $x$-axis corresponds to the quantity $d - d_H(w, w^*)$.

For comparison between the landscapes in Figures 7a and 7b and the standard functions used to compare hyper-heuristics, we provide in Figure 8 plots for the three most used benchmarks. These functions presents valleys and hills which are clearly visible. It is worth mentioning that the theoretical work of Bendahi et al. (2025) applies to a class of functions similar to these three, which is not the case of the landscapes in Figures 7a and 7b.

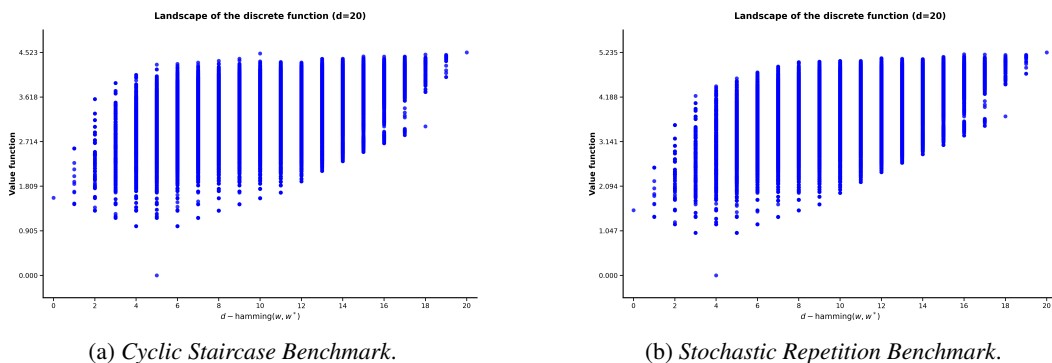

(a) *Cyclic Staircase Benchmark.*    (b) *Stochastic Repetition Benchmark.*

Figure 7: Instance $(n, c) = (5, 4)$ on the Two Benchmarks

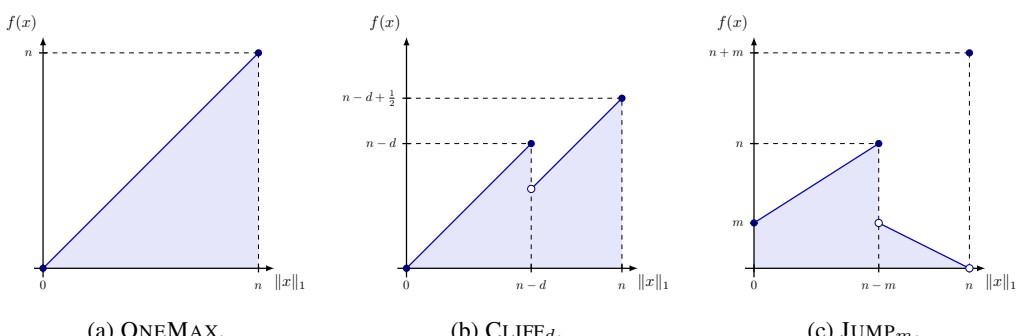

(a) ONEMAX.    (b) CLIFF$_d$.    (c) JUMP$_m$.

Figure 8: Plot of the Three Most Common Benchmarks in Hyper-Heuristics.

## I NOTES ON THE UNIQUENESS OF OPTIMAL SOLUTIONS

A natural question following Theorem 4.4 is whether there exists a unique optimal solution to the problem $(\mathscr{P}_d)$ or not and under which sufficient condition(s) uniqueness can hold.

First, we show that we can always construct an instance of the problem $(\mathscr{P}_d)$ that has more than one optimal solution.

**Lemma I.1** (Proof in Appendix E.5). *For any positive integer $d \geq 2$, there exists a strictly upper triangular $\mathbb{R}^{d \times d}$ matrix $M$ with non-negative entries and a vector $\mathbf{a} \in \mathbb{R}_+^d$ such that problem $(\mathscr{P}_d)$ admits at least two solutions in $\mathbb{R}_+^d$.*

The specific instance built in the previous lemma relied on the fact that $\mathbf{a}$ can have distinct coordinates. We can ask the same question when all coordinates of $\mathbf{a}$ are equal[36], which reduces, due to the scale-invariance of $(\mathscr{P}_d)$ in $\mathbf{a}$, to $\mathbf{a} = (1, \ldots, 1)^\top$.

**Lemma I.2.** *For any $2 \times 2$ strictly upper triangular matrix $M$ with non-negative entries, if $\mathbf{a} = (1,1)^\top$ then the problem $(\mathscr{P}_2)$ admits a unique global maximizer.*

However, Lemma I.2 fails to hold in higher dimensions. For example, the following instance of $(\mathscr{P}_d)$ in dimension $d = 3$

$$M = \begin{pmatrix} 0 & 2 & 0 \\ 0 & 0 & 1 \\ 0 & 0 & 0 \end{pmatrix} \text{ and } \mathbf{a} = \begin{pmatrix} 1 \\ 1 \\ 1 \end{pmatrix} \tag{239}$$

has the following two maximizers: $\Lambda_1^* = (1, 0, 1)^\top$ and $\Lambda_2^* = \left(\frac{1}{2}, \frac{1}{2}, 1\right)^\top$. Nonetheless, the following simple and sufficient condition ensures the uniqueness of the optimal solution of $(\mathscr{P}_d)$.

**Theorem I.3** (A Sufficient Condition for Uniqueness). *For any positive integer $d$, if the matrix $M$ is strictly upper triangular with non-negative entries and satisfies, for all $k \in [d]$*

$$\sum_{\substack{i=1 \\ i<k}}^{d} M_{i,k} < 1, \tag{240}$$

*then with the vector $\mathbf{a} = (1, \ldots, 1)^\top \in \mathbb{R}^d$ the problem $(\mathscr{P}_d)$ admits a unique global maximizer.*

For further details and proofs of Lemmas I.1 and I.2 and Theorem I.3, the interested reader is invited to consult Appendix E.5 where all the claims stated in this section are rigorously established.

---

[36]This choice is motivated in Section 5. In the analysis of asynchronous gradient descent, the problem $(\mathscr{P}_d)$ naturally arises and the vector $\mathbf{a}$ is simply $(1, \ldots, 1)^\top$.

## J    NOTE ON THE USAGE OF LARGE LANGUAGE MODELS

The authors acknowledge the use of Large Language Models to assist in polishing the writing of this manuscript. The LLMs were used only for language refinement and did not contribute to the research ideas, experimental design, analysis, or conclusions exposed here.

