# OpenReview forum: "Sharper Characterization of the Global Maximizers in Bilinear Programming with Applications to Asynchronous Gradient Descent"
_ICLR.cc/2026/Conference — Submitted to ICLR 2026_

### Official Review · Reviewer_sW6C · 2025-10-17

**Soundness:** 2
**Presentation:** 3
**Contribution:** 2
**Rating:** 8
**Confidence:** 2

**Summary:**

The paper studies a stepsize optimization problem for Asynchronous GD (AGD). The stepsize optimization problem can be cast as an optimization problem with a linear objective and bilinear constraints. The authors show that the class of optimization problems achieves global maximization necessarily at extreme points. Finally, they show a search heuristic for the problem class and compare the performance with Gurobi solvers on several benchmarks.

**Strengths:**

The presentation of the results is clean and easy to follow. The theory of the bilinear program problem begins with a few basic properties (unique solution of linear-quadratic system and regularity of solution). The theory development culminates at Theorem 4.6, where the authors show that the global maximizers are mapped to the vertices of the unit hypercube.

In the analysis of Asynchronous GD, one way to obtain an upper bound is to make sure a term of $R(K) < 0$, which gives an upper bound of $\frac {2 \Delta} {\Gamma_k}$. This can be formulated to the stepsize optimization problem of interest in this work.

Then, the authors tested the search heuristic on a few benchmarks. From the experiments reported in the paper, it outperforms Gurobi solver significantly.

**Weaknesses:**

The main theoretical result in this paper is a characterization of the global maximizer of the problem. They show that the global optimizer corresponds to (after the map $\psi$) vertices of the unit hypercube. The derivation of this extreme point characterization seems to be quite standard and does not involve main technical novelty.

The connection to the asynchronous GD seems a bit weak, since this is only one way to get an upper bound of AGD, and the construction is also quite brittle. For instance, it does not give the upper bound for asynchronous SGD, where the objective function is not linear anymore.

**Questions:**

Why didn’t the authors include any experiments on the AGD stepsize schedules? As this is one of the main motivations of this study (at least from the presentation of this work it seems so), it would be great to know whether the stepsize returned by MMAHH is indeed good in practice.

---

> ### Author Response · Authors · 2025-11-20
>
> Thank you for the positive and careful assessment of our work, and for suggesting empirical confirmations of the theory. We agree that experiments can be valuable in general, however, in our case the key empirical insight we aim to emphasize is different: we show that hyper-heuristics can be effectively used to solve continuous optimization problems when they are guided by mathematical priors (such as our necessity theorem). We view this integration of hyper-heuristics with continuous optimization as the main novel contribution, and our experiments are designed to isolate and demonstrate exactly that point, namely, that this bridge is both feasible and yields consistent gains over industrial solvers.

---

> > ### Comment · Reviewer_sW6C · 2025-11-21
> >
> > Thank the authors for the response. I will keep my score for now, which is indeed very positive.
> >
> > I also notice very mixed opinions from the other reviewers. I will also consider the feedback from other reviewers later in the reviewer discussion phase.

---

### Official Review · Reviewer_5vSE · 2025-10-25

**Soundness:** 3
**Presentation:** 3
**Contribution:** 3
**Rating:** 4
**Confidence:** 4

**Summary:**

This paper systematically characterizes asynchronous GD (AGD) with delays and derives near‑optimal stepsize policies. It models asynchrony via a delay matrix + constraint program, introduces effective delay after threshold‑based discarding of stale gradients, and proves that Ringmaster‑AGD  achieves a 2‑approximation to the optimal cumulative stepsize. Convergence is established under nonconvex smoothness. Experiments implement a discrete/mixed‑integer solver (MMAHH) to approximately solve the planning problem.

**Strengths:**

1. Clean problem modeling. The interaction among delay, stepsize, and discard rules is precisely abstracted; definitions/feasible regions are reusable.
2. Novel theoretical guarantee. A constant‑factor (2×) approximation for a fixed‑threshold discard policy is practically useful.
3. From theory to implementation. Casting the stepsize search as a discrete/MINLP‑like problem and providing a heuristic solver bridges theory and practice.

**Weaknesses:**

1. Strong assumptions, missing stochasticity. Core analysis assumes deterministic full gradients; most real training uses mini‑batch stochastic gradients. Please extend analysis or at least provide systematic experiments under stochastic noise.
2. Scalability & deployability. The effective‑delay program trends toward mixed‑integer nonlinear; while MMAHH is proposed, its scaling and runtime vs. optimal solvers (or simpler heuristics) are not systematically evaluated.
3. System‑level evidence. Experiments emphasize solver quality but lack end‑to‑end async training on real multi‑GPU/multi‑node systems with realistic delay traces. Compare against fixed stepsizes, Polyak stepsizes, and different thresholds.
4. Constant dependence & hyperparameters. Sensitivities to $L$, threshold $R$, and feasible‑set parameters are under‑explored; mis‑specification robustness is unclear.

**Questions:**

1. With mini‑batch noise, can Theorem 5.4–style results or the 2‑approximation be retained (perhaps with variance terms)?
2. How is $\delta_k^e$ estimated online, and how do $R$ and $L$ mis‑specification affect near‑optimality?
3. What are the time/quality trade‑offs of MMAHH across numbers of workers $n$ and horizon $K$?

---

> ### Author Response · Authors · 2025-11-20
>
> Thank you for your review and insightful remarks.
>
> **Regarding the Mini-batch Noise (Q1).** The $2$-approximation argument relies on Theorem G.$18$, which in turn depends on our main deterministic necessity result (Theorem $5.4$). All of these results are specific to the asynchronous _deterministic_ setting. Extending them to the stochastic case is nontrivial: when $\sigma > 0$, the objective function is no longer linear (eq. ($212$)), and several key steps in our proof no longer hold. As a result, Theorem $5.4$ cannot be applied directly, and retaining a $2$-approximation would require substantial new ideas rather than a straightforward modification of the existing proofs.
>
> **Regarding the Misspecification of $R$ and $L$ (Q2).** Based on our current analysis alone, it is difficult to draw definitive conclusions about the impact of mis-specifying $R$ and $L$ on near-optimality. However, we see this as an interesting direction for future work.
>
> **Regarding the Scalability of the MMAHH w.r.t. $n$ and $K$ (Q3).** While the MMAHH's runtime naturally increases with problem dimension (and therefore with the number of workers $n$ and horizon $K$), our experiments indicate that MMAHH scales more favorably than the commercial solver Gurobi $11$ (see, for example, Figure $6$ in our manuscript).

---

### Official Review · Reviewer_ZNk7 · 2025-11-01

**Soundness:** 2
**Presentation:** 1
**Contribution:** 2
**Rating:** 2
**Confidence:** 3

**Summary:**

**Summary**

This paper investigates a special type of bilinear programming (BLP) problem and characterizes the properties of its optimal solutions. Based on the properties of the optimal set, the authors design tailored heuristics to accelerate the solution of the bilinear program. The technique is then applied to solving bilinear programs arising from the stepsize design in asynchronous gradient descent.

**Strengths:**

**Strength**
The paper is generally easy to follow.

**Weaknesses:**

**Weaknesses**

Despite the claimed contributions of the paper, I find this paper poorly written and have several concerns:

1. Lack of motivation and unclear focus of the paper

   The paper devotes considerable effort to the context of BLP and presents numerous auxiliary concepts and results in the appendix. However, after establishing the results, the only application is deriving the stepsize in asynchronous gradient descent (AGD). This is clearly overdone and makes the paper's focus unclear. What's worse, even the application to AGD looks suspicious. In particular, in the contribution section, the paper claims "selecting improved stepsizes for AGD can be cast as a BLP". However, this design relies on a known delay matrix, and in a real AGD algorithm implementation, I don't think it makes sense to assume knowledge of this matrix. Hence, the application to AGD seems restricted to post-hoc analysis.

2. Importance of the theoretical result

   I'm also concerned about the importance of the claimed theoretical result. In a word, the paper shows that every optimal solution of the considered type of BLP is extremal. This result seems strong at a first glance; however, given that **Theorem 3.2** always guarantees the existence of an extremal optimal solution, it already suffices to focus on the extremal solutions since we typically want *one* optimal solution. Therefore, I don't think the result of the paper is that important.

In addition, I find a number of notation inconsistencies and typos throughout the paper. Overall, I find the paper lacks motivation. The importance of the theoretical result is unclear, and its application does not look interesting. I don't think the paper meets the publication standard of ICLR.

**Questions:**

**Questions**

1. Could you justify the application of BLP in AGD's analysis beyond doing post-hoc analysis?
2. Why would the fact that "all the optimal solutions are extremal" be more helpful compared to "there exists one extremal optimal solution" if you only need one optimal solution?
3. **Theorem 5.4** analyzes AGD. According to the algorithm description, I did not see a source of randomness. Where is the expectation from equation (12) from?

**Minor issues**

1. Line 117, 159, 192

   Both the boldface letter like $\mathbf{a}$ and capitalized letter $\Lambda$ are used to denote vectors.

2. Line 122, 140

   The index  $i$ in $M_{i, j}$ is unused.

3. Line 143

   "...being induced by..." this sentence is not clear.

4. Line 311

   $f$ is the deterministic objective and does not have sample as its argument.

**Details Of Ethics Concerns:**

N/A.

---

> ### Author Response · Authors · 2025-11-18
>
> Thank you for your review and thoughtful comments.
>
> **Regarding the Motivations of our Work (W1 + Q1).** We agree that, in general, the delay matrix in asynchronous gradient descent (AGD) is unknown, essentially because the compute times of the workers are unknown. Therefore, the BLP formulation cannot be used to directly select stepsizes in a real-time implementation. However, there are meaningful scenarios in which the delay matrix is either known or structured, e.g., in a fixed computation model where each worker $i \in \{1, 2, \ldots, n\}$ has a fixed computation time $\tau_i > 0$ to compute a single gradient. In this setting, it is possible to determine in advance which worker will send a gradient to the server (e.g., it is enough to order $\{(k \tau_i, i) \colon i = 1, \ldots, n \text{ and } k \ge 1\}$ using lexicographic order), though we note that solving the resulting optimization problem may be computationally expensive.
>
> Also, we wanted to outline that the general BLP context and the auxiliary results presented in the appendix are not only formal derivations: they provide a unifying theoretical framework that underpins the AGD application and can be adapted to study other problems involving bilinear or quadratic constraints. For instance, in many deterministic nonconvex optimization algorithms, the convergence analysis leads to an expression of the form
>         $$ \frac{1}{\Gamma_K} \sum_{k = 0}^K \gamma_k \, \left\| \nabla f \left( x^k \right) \right\|^2 \le \frac{c\Delta}{\Gamma_K} + \frac{1}{\Gamma_K} \sum\limits_{k = 0}^K \gamma_k R_k C_k  $$
> where $c > 0$ is a universal constant, $\Gamma_K := \gamma_0 + \cdots + \gamma_K > 0$, $C_k \ge 0$ and $R_k := A_k \gamma_k - 1 \le 0$. Therefore, although we emphasize the AGD case due its elegancy and current relevance, our results are broadly applicable beyond this single example. Including this material ensures that the paper makes a broader theoretical contribution.
>
> Finally, while the application presented in our manuscript may seem artificial, we think that the theoretical contributions and the main theorem are still important, since together they provide a structural necessity result on a wide class of optimization problem, and to our knowledge, no prior work establishes this necessity result.
>
>
> **Regarding the Importance of our Necessity Result (W2 + Q2).** When an extremal optimal point is only guaranteed to exist, the solution set may still contain a large manifold of non-extremal optimal solutions. In such cases, a solver may legitimately return a non-extremal optimial point, and that optimal point typically corresponds to a mixture of configurations rather than a meaningful discrete decision. Our result proves that all the optimal solutions correspond to interpretable discrete configurations. Beyond its theoretical interest, this guarantees that any optimizer will return solutions that are intrinsically discrete and easy to interpret, rather than arbitrary mixtures.
>
> For instance, in the AGD application, we proved that optimal strategies for AGD stepsizes are greedy: either reject the full gradient or accept it with the largest possible stepsizes allowed by the constraints.
>
> **Regarding the Source of Randomness in Theorem $5.4$.** This was a typo in the manuscript, which has now been corrected. The confusion arose because the full analysis of asynchronous stochastic gradient descent (ASGD), which includes stochasticity, is provided in the appendix; the result in Theorem $5.4$ can be recovered as a special case of that more general analysis.
>
> As for the minor issues, most of them are fixed.

---

### Official Review · Reviewer_AT7b · 2025-11-06

**Soundness:** 3
**Presentation:** 3
**Contribution:** 2
**Rating:** 6
**Confidence:** 4

**Summary:**

This paper studies a bilinear programming (linear objective defined over a compact hypercube; bilinear constraints) that is closely related to the step-size tuning for asynchronous GD. The paper introduces a special instance of a bilinear program (Eq. 1): to find a step size sequence $(\gamma_k)_k$ that minimizes a theoretical convergence rate of asynchronous GD under usual assumptions like Lipschitz smoothness. For this particular problem, it is shown that every global maximizer is an extreme point (e.g., a corner or vertex) of its feasibility set. Based on this finding, it has been demonstrated that an evolutionary, randomized heuristic called Markov move-acceptance hyper-heuristic (MMAHH) can be used to find a maximizer. The tested MMAHH-based solver runs much faster than a production-level Gurobi 11 solver on a tested benchmark (e.g., Cyclic Staircase, Stochastic Repetition, Random Sequences).

**Strengths:**

- The paper is clearly written and self-contained. It provides almost every background knowledge to understand the whole paper in its appendices, which is very satisfying.
- The bilinear programming studied in the paper has a strong motivation in the hyperparameter tuning problem for an optimization algorithm with multiple workers (AGD).
- I think most of the proofs are correct. I have read most of the proofs in Appendices C, E, F, and G, but I couldn’t find significant errors (except for some typos or redundant sentences).
- The main contributions are sound, novel, and interesting:
    - The paper provides a sharper characterization of the optima of a subclass of bilinear programs. Given the renowned sufficiency, which describes that a linear functional attains its maximum over a nonempty compact set K at an extreme point of K (which can be easily proved using, e.g., Theorem 3.2.2 of Barvinok (2022)), this work offers a necessity under the particular bilinear programs of interest: every maximizer is an extreme point for these problems. This means that we (will) never ignore any of (possibly non-extremal) maximizers by searching for maximizers over a finite number of extreme points.
    - The implication in convergence rate analysis of (non-stochastic full-batch) AGD is interesting and significant. Using the paper’s main results, the paper shows that the step size choice of a recently proposed Ringmaster A(S)GD algorithm is very close (up to a constant factor 2) to the optimal step size choice for AGD in terms of minimizing a convergence upper bound (under nonconvex L-smoothness assumption).
    - Lastly, the paper offers a simple and efficient heuristic (an MMAHH-based solver) to solve the step size tuning problem on a discrete domain.

---

A. Barvinok, *A course in convexity*, 2nd edition, Grad. Stud. Math. 54, American Mathematical Society, New York, 2002.

**Weaknesses:**

1. In my view, the very first paragraph of the introduction is redundant. It is dedicated to summarizing the well-known history of modern developments in AI/ML/DL. Being almost irrelevant to the paper’s topic, it does not help readers get motivated with the problem setting. The authors had better open the first section with some background on (and importance of) asynchronous GD and/or difficulties in choosing an appropriate sequence of step sizes.
    - Provided that some additional spaces after omitting/shrinking the historical paragraph, I would suggest to add some more intuitions on why the particular bilinear programming satisfies the proven necessity condition, e.g., describing some geometric properties (proved in Appendix D.2 and D.3, but in plain words!) of the feasibility set that enforces the bilinear program to have no non-extremal maximizers.
2. It is worth mentioning that Theorem 3.2 (sufficiency) is quite a well-known result since it is easy to prove it using the compactness of the feasibility set and a theorem in a math textbook, for example, Barvinok (2022, Theorem 3.2.2) (mentioned in ‘Strengths’).
3. A single sentence in Lines 326-330 is too long; it needs rewriting.
4. Overall, there is room for improvement in the organization of the proofs in the appendices, in particular, E, F, and G. For instance, there are proofs of the main/important theorems (e.g., Theorems 4.3, 4.4, 4.6, …) in the section with a name “Omitted Proofs”! Also, although Theorems 4.3 and 4.4 are relevant to each other (and in fact the proof of Theorem 4.4 seems to depend on Theorem 4.3), their proofs are far apart (one is in Appendix E.3 and another is in Appendix F.1).
5. Notation and font styles for scalars, vectors, and matrices are very disorganized throughout the paper; please use a consistent set of symbols. Have a look at the usual math notation in machine learning literature: https://github.com/goodfeli/dlbook_notation/blob/master/math_commands.tex
6. Lemma E.3: I believe this lemma can be proved more cleanly by using zero derivative at the minimum/maximum.
    - (Same proof until Equation (76).) Let $h_i(x) = x_i (1 + (Mx)_i)$. Note that $h_i\in C^\infty$ because it’s a polynomial of $x$. Define $\tilde{h}_i : (-\epsilon_0, \epsilon_0) \to \mathbb{R}$ as $\tilde{h}_i(t) = h_i (p + td)$, which is also $C^\infty$. Since $p+td \in \mathscr{F}$, the range of $\tilde{h}_i$ is a subset of $[0, 1]$, i.e., $\tilde{h}_i (t) \in [0,1]$ for all $t\in (-\epsilon_0, \epsilon_0)$. Observe that $\tilde{h}_i$ has either a minimum or a maximum at $t=0$: $\tilde{h}_i (0) = 0$ if $i \in Z$ and $\tilde h_i (0) = 1$ if $i\in S$. Thus, $\tilde{h}^\prime _i (0) = \nabla \tilde h_i (p) \cdot d = 0$. It means that the jacobian of $h(x) = (h_1(d), \cdots, h_d(x)) = x+ x \odot (Mx)$ satisfies $\nabla h (p) \cdot d = 0$. Since $\nabla h(p)$ is a $P$-matrix and hence invertible, we have $d=0$. (Proceed with the remaining proof to yield a contradiction.)
7. Here is a list of typos or misleading reasons that I find crucial. (Some of them might be wrong.)
    - Equations (1) and (3): the subscript $i$ —> $k$
    - Line 173 “where $\mathbf{a} \in \mathbb{R}^d \setminus \\{0\\}$ …”: $\mathbf{a}$ —> $\mathbf{c}$
    - Equation (13): factor 2 is missing in the middle term of LHS (accordingly, I guess the definition of $M_{ij}$ in Line 383 must be $2\delta^j \mathbb{I} \\{j\in M_i\\}$)
    - Line 384: “left-hand side” —> “right-hand side”
    - Definition D.8 versus Lemma D.11: Is  $\deg (I)$ a set of indices? Or its size?
    - Below Equation (48): The inequality (a) is only due to the non-negativity of $t$ and $1-t$. The inequalities $f(p)\ge f(x)$ and $f(p) \ge f(y)$ imply the inequality (b), not (a).
    - Equation (67): wrong beginning index of the sum in the middle ($k+1$ —> $1$) unless $M$ is strictly upper triangular
    - Lemma E.3: A typo in line 2665: “for every $i\in [d]$” —> “for every $i\in Z$”.
    - Equations (93) and (94): $\pm tv$ terms are missing in $\phi_k^\square(\Lambda)$.
    - Line 7 of Algorithm 3: “Run Procedure 1 in all workers” —> “Run Procedure 2 in all workers”
    - Equation (196): In the last term, the sum in the parentheses must be $\sum_{j\in M_k} \gamma_j$

**Questions:**

- As far as I understood, in Section 6, the MMAHH-based heuristic is compared with Gurobi 11 software for solving discrete optimization benchmarks (over $\\{0,1\\}^d$). Have you numerically compared these two solvers for solving the main bilinear programming $(\mathscr{P}_d)$ or a mixed-integer variant $(\mathscr{P}^{\rm mi}_d)$ as below?:
    - MMAHH solver: Solve it as a discrete optimization over the vertices of a hypercube (as already done in the paper).
    - Gurobi 11: Solve it directly, considering the compact feasibility set (which has infinitely many elements) as it is, (possibly) allowing practical implementations like the Big-M Method (mentioned in Appendix G.10).
- How should we compute $\Psi(\cdot)$ efficiently and practically if it is an implicit function? Or, equivalently, is the set of extreme points of a feasibility set guaranteed to be easy to obtain? If so, please explain a general way to do so (as a rebuttal and in the paper).

---

> ### Author Response · Authors · 2025-11-18
>
> Thank you for your review and insightful remarks.
>
> **Regarding the Benchmarks.** In our experiments, we directly compare the two approaches on the bilinear programs $(P_d)$. In particular, Gurobi 11 is applied to the original continuous formulation, whereas MMAHH is applied only to the corresponding discrete optimization problem obtained via the mapping $\Psi$ introduced in the manuscript. This ensures that both solvers are evaluated on equivalent instances of the underlying optimization task.
>
> **Regarding the Computation of $\Psi$.** The mapping $\Psi$ is defined as the (unique) solution of a system of nonlinear equations. In general, a closed-form expression for $\Psi$ is not available, but it can be computed efficiently using standard numerical methods for solving nonlinear systems (e.g. with fixed-point methods). Importantly, for the specific class of bilinear programs arising from asynchronous GD -which is the one used in all of our experiments- the resulting system is upper triangular, which allows us to obtain $\Psi$ in closed form (eq. ($158$) and ($159$)) and compute it directly. This structure ensures that, in the settings we consider, evaluating $\Psi$ is both practical and computationally cheap.
>
> **Regarding the Typos.** Thank you for carefully pointing these out. We will correct all noted typos and minor issues in the camera-ready version.

---

> > ### Comment · Reviewer_AT7b · 2025-11-22
> >
> > I thank the authors for their rebuttal. The response resolved most of my questions.
> >
> > Unfortunately, I am wondering why the authors do not leave any comments about the weaknesses I pointed out.
> > Although most of the points in the weaknesses I raised are issues about the paper's presentation, which I think are quite easy to fix, I would say they are not all about typos.
> > However, the authors answered that they will only correct the typos that I mentioned in my review.
> > Not only did they remain silent about my suggestions about further modifications in the paper, but they also did not reflect any of them in their revised manuscript.
> >
> > My current score (6) is already high, for which I assumed that the authors would either modify all the problems that I have mentioned or explain what I had misunderstood.
> > I find that the other reviewers' assessments are mixed (though I have not checked them all yet).
> > Hence, I want to let the authors know that, if they have no intention to improve their manuscript further as per my comments, and the other reviewers' points are indeed correct, I am willing to decrease my score (to 4, probably).
> >
> > I am curious whether the authors are planning to submit another revised version of their manuscript.

---

> > > ### Author Response · Authors · 2025-11-27
> > >
> > > We thank the reviewer for the careful follow-up and we are sorry for the time taken to address most of the points raised above.
> > >
> > > First, we agree that the opening paragraph of the introduction may have felt somewhat disconnected from the core of the paper. In the revised manuscript, we have shortened and refactored this paragraph to better align it with our main contributions.
> > >
> > > Second, we have added a brief subsection (Subsection 3.2) providing geometric intuitions for our main result, along with references to the lemmas that analyze the feasible region $\mathscr{F}$. We also clarify an assumption on the cost vector $\mathbf{a}$, whose entries are required to be non-zero; this condition is essential for the necessity result to hold.
> > >
> > > Third, regarding the sufficiency result: we already explicitly note in the manuscript that this result is not new (_Theorem 3.4 above is in fact a special case of Theorem 3.1 from Chen et al. (2021)_), and we have kept this clear attribution in the revision. In addition, we have added Remark E.1 following the proof of Theorem 3.4 to insists on this point, adding the reference suggested by the reviewer (i.e., A. Barvinok, A course in convexity, 2nd edition, Grad. Stud. Math. 54, American Mathematical Society, New York, 2002).
> > >
> > > We also thank the reviewer for pointing out the overly long sentence; we have now rewritten it for clarity.
> > >
> > > We appreciate the suggested simplification of one of our proofs. We agree that the reviewer’s argument is cleaner, and we have incorporated it into the proof of Lemma E.3 in the revised version. We will also give credit to the reviewer for this contribution in the acknowledgments section of the camera-ready version.
> > >
> > > We have corrected all typographical errors that have been identified by the reviewer in the _main part_ of the paper.
> > >
> > > Finally, concerning the remaining comments on the organization of the appendices: we agree that the structure can indeed be improved. Due to space and time constraints during the rebuttal period, we were not able to fully reorganize the material, but we plan to address these presentation issues in the camera-ready version, where we will reorganize the appendix for better readability.
> > >
> > > We hope that these clarifications address the reviewer’s concerns, and we thank the reviewer again for the detailed and constructive feedback.

---

> > > > ### Comment · Reviewer_AT7b · 2025-11-27
> > > >
> > > > Thank you for your further response. I have checked the updated manuscript and recognized that it reflects most of the concerns I raised. Also, I understand there is not much time to fix all the typos and the organization of appendices. Hence, I retain my score to 6.

---

### Official Review · Reviewer_i93o · 2025-11-07

**Soundness:** 2
**Presentation:** 2
**Contribution:** 2
**Rating:** 2
**Confidence:** 3

**Summary:**

The paper is dedicated to solving a nonconvex quadratically constrained linear program (QCLP) of special structure inspired by the problem of stepsize choice in asynchronous gradient descent (AGD). The authors show that the set of maximizers belongs to an inverse image of hypercube vertices under a smooth invertible map, thus reducing the nonconvex optimization problem to a search over a finite set. Consequently, the authors propose a solver for the studied class of problems based on an evolutionary heuristic recently proposed in the literature. Additionally, the paper refines the analysis of AGD and explains how the results in quadratically constrained linear programming may help in choosing its stepsizes.

**Strengths:**

The authors conduct an in-depth analysis of a class of nonconvex QCLPs. This analysis might be a valuable contribution as there is a potential to apply techniques and lemmas to other QCLP arising in various applications.

**Weaknesses:**

- The term *bilinear program (BLP)* used in the title and throughout the text is confusing. The authors write: "BLPs are a class of nonlinear optimization problems in which the objective function or constraints involve products of pairs of variables from two distinct sets". This definition is correct, but it does not apply to Problem (1) the paper is solving. To the best of my knowledge, Problem (1) is a nonconvex quadratically constrained linear program.
- One of the paper's main motivations is selecting the step size for AGD. However, it's unclear whether the proposed approach to this problem is practical. First, the problem is reduced to a search over $2^{K+1}$ points, where $K$ is the number of steps. If my understanding is correct, the largest $K$ considered in experiments is $48$, whereas in practice, first-order methods often perform thousands of iterations, which may lead to prohibitively high execution time of the approach. Furthermore, the experiments do not demonstrate the actual performance of AGD with the computed stepsizes, only the time it took to compute the stepsizes.
- The proof of Theorem 3.2 is almost identical to the proof in the post https://math.stackexchange.com/a/3952098/1134792, but no attribution is given. This point requires clarification.

**Questions:**

- How do you compute the map $\Psi$ (and its inverse) in your solver?
- Could you add a caption to the first figure?
- I suggest moving the definition of extreme point to subsection 3.2 to improve readability.
- Please consider giving the definition of diffeomorphism in appendix to remind it to a reader.

I am open to reconsidering the score.

---

> ### Author Response · Authors · 2025-11-18
>
> Thank you for your review and thoughtful comments.
>
> **Regarding the BLP term.** We agree that this point may be confusing. While Problem $(1)$ is not, _strictly speaking_, a BLP, the special case that arises in the analysis of AGD -where the matrix $M$ is strictly upper triangular- is indeed a BLP. We have clarified this in the manuscript by referring to Problem $(1)$ as a “quadratic program”, while retaining the “BLP” term for the specific case that arises in asynchronous gradient descent (AGD), as referred notably in the title and abstract of the paper.
>
> **Regarding the Attribution to the Post.** We thank the reviewer for pointing this out. We inadvertently omitted the appropriate attribution when preparing the manuscript. This has now been corrected: the revised version explicitly acknowledges the Math StackExchange post in a footnote and cites it as the source on which our argument is based.
>
> **Regarding the Computation of the Map $\Psi$.** The map $\Psi$ is defined as the (unique) solution to a system of nonlinear equations. In general, there is no closed form for $\Psi$, however it can be computed numerically using a standard nonlinear solver (e.g., Newton's method or a fixed-point iteration). In some particular cases (including the class of BLPs that we fully study in our manuscript, and for which we conducted experiments), the system is upper triangular and we have a closed-form for $\Psi$.
>
> Concerning its inverse, the computation of $\Psi^{-1}$ is straightforward from the definition: $\Psi^{-1}(\Lambda) = \Lambda \odot ( 1 + M \Lambda)$, which is just a quadratic form.
>
> **Regarding the Presentation Comments.** Thank you for these remarks. In the revised PDF, a descriptive caption has now been added to the first figure, the two definitions have been moved accordingly, and the definition of diffeomorphism has been added to the appendix, with a footnote  included in the main text to guide the reader to it.

---

### Meta-Review · Area_Chair_s4wW · 2026-01-05

**Summary:**

This paper studies a class of bilinear programming. The authors characterize the optimal solution of this problem and design tailored heuristics to accelerate the solution. The practicality of this method is questionable due to the procedure of searching over $2^{K+1}$ points. Additionally, the presentation should be significantly improved. Therefore, I recommend rejection.

**Reviewer Concerns:**

There are several main concerns have not been addressed, such as the presentation and the expensive computational cost of proposed methods.

**Reviewer Scores:**

I think the reviewer will maintain the scores.

---

### Decision · Program_Chairs · 2026-01-26

Reject